# Stage-specific TRIM10 expression regulates erythroid maturation

Heesoo Kim[1], Wonji Shin[2], Dongeun Lee[1], Byunghoon Jeon[1], Yongbo Kim[3], Donghyuk Shin[1], Hyobin Jeong[1], Jun Young Hong[1], Sungwook Lee [ID][3 ✉] & Boyoun Park [ID][1,2 ✉]

## Abstract

**Mammalian erythroid cells undergo extensive organelle and protein remodeling during erythropoiesis. The transcriptome and proteome of ubiquitin E3 ligases change dynamically during erythroid differentiation, yet mechanisms beyond E3 activity remain unclear. Here, we identify that tripartite motif-containing protein 10α (TRIM10α), an erythroid- and stage-specific E3 ligase, as crucial for stepwise erythroid maturation. TRIM10α self-association to localize on erythroblast surfaces, binding extracellular complement C1q, which facilitates pyrenocyte encapsulation and macrophage recognition. Surface C1q interacts with EpoR to promote lysosomal degradation, and its depletion prolongs Epo signaling. Notably, cytosolic TRIM10α enhances hemoglobin (Hb) maturation and sequesters Hb aggregates under oxidative conditions. Ultimately, TRIM10α self-ubiquitination and its binding to p62 are anticipated to lead to TRIM10α degradation, promoting the removal of Hb aggregates via autophagy. In contrast to TRIM10α, an alternatively spliced TRIM10β, which is barely expressed in human tissues and cells, forms deleterious aggregates, suggesting that evolutionary suppression of TRIM10β supports erythroid homeostasis. Our findings propose that aberrant TRIM10 expression drives erythroid-related diseases and highlight TRIM10 as a potential biomarker or therapeutic target.**

**Keywords** TRIM10; C1q; Erythropoiesis; Hemoglobin Maturation
**Subject Categories** Development; Post-translational Modifications & Proteolysis; Signal Transduction

## Introduction

Mammalian terminal erythropoiesis is a finely tuned process involving organelle clearance and proteome remodeling to optimize function and maintain homeostasis (Chen et al, 2009; Liu et al, 2010; Mohandas and Evans, 1994; Mohandas and Gallagher, 2008; Salomao et al, 2008). This process progresses from progenitor commitment to erythroid maturation, characterized by selective hemoglobin (Hb) preservation, reduced global protein synthesis, and nuclear expulsion (Keerthivasan et al, 2011; Khandros and Weiss, 2010). Pyrenocytes, membrane-coated nuclei formed post-enucleation, are cleared by macrophages via phosphatidylserine-dependent engulfment (Chasis and Mohandas, 2008; McGrath et al, 2008; Yoshida et al, 2005). Erythroid maturation is thus tightly regulated by stage-specific gene expression and epigenetic changes, with disruptions linked to anemia and hematological disorders (Cheng et al, 2009; Kingsley et al, 2013; Schulz et al, 2019); however, the molecular regulation of this process is not fully understood.

During terminal erythroid differentiation, the ubiquitin-proteasome system facilitates extensive protein turnover, supplying amino acids for globin synthesis (Benderoff et al, 1978; Braten et al, 2016; Shabek et al, 2012; Wefes et al, 1995). Correspondingly, the expression of various E3 ubiquitin enzymes, including HECT domain E3, *SIAH*, *FBXO30*, *MKRN1*, and various tripartite motif (*TRIM*) genes, are upregulated during this process (An et al, 2014b; Nguyen et al, 2017; Schulz et al, 2019). Specifically, TRIM proteins, the largest superfamily of RING-type E3 ligases, are characterized by diverse domains and expression patterns, playing roles in immune response, apoptosis, autophagy, and differentiation (James et al, 2007; Vunjak and Versteeg, 2019). They are thus implicated in human diseases such as tumorigenesis and autoimmune disorders (Watanabe and Hatakeyama, 2017; Zhu et al, 2022). While their functions in erythroid cells remain poorly understood, TRIM58 is known to promote enucleation during erythroid maturation by degrading dynein (Thom et al, 2014). Recent findings suggest TRIM proteins also form oligomers through domains beyond the RING domain (Esposito et al, 2017; Hatakeyama, 2017; Perfetto et al, 2013), hinting at broader roles beyond E3 ligase activity.

Here, we identified TRIM10α as crucial in erythroid maturation, capturing extracellular C1q to coat pyrenocytes for clearance. TRIM10α-mediated surface C1q drives lysosomal targeting of EpoR, disrupting global protein synthesis. Cytosolic TRIM10α has dual roles: facilitating Hb maturation in mid stages and sequestering oxidized Hb aggregates through self-oligomerization in late stages, targeting them for autophagy via p62 binding. Conversely, the spliced variant TRIM10β, absent in mice, forms harmful aggregates, highlighting the need to inhibit its expression for

[1]Department of Systems Biology, College of Life Science and Biotechnology, Yonsei University, Seoul 03722, South Korea. [2]Interdisciplinary Program of Integrated OMICS for Biomedical Science, Yonsei University, Seoul 03722, South Korea. [3]Immuno-oncology Branch, Research Institute, National Cancer Center, Goyang 10408, South Korea. ✉E-mail: swlee1905@ncc.re.kr; bypark@yonsei.ac.kr

erythroid homeostasis. Thus, *TRIM10* expression must be tightly regulated during erythroid maturation, as its dysregulation can lead to erythroid-related diseases.

## Results

### Aberrant stage-specific *TRIM10a* expression in erythroblasts is linked to anemias

Given that the expression patterns of ubiquitin-related enzymes exhibit stage-specific dynamics during erythroid maturation (An et al, 2014b; Schulz et al, 2019; Wefes et al, 1995), we utilized public gene expression datasets to investigate the role of TRIM proteins. Unlike *TRIM58*, *TRIM10a* exhibited erythroid lineage-restricted expression, with marked upregulation in human erythroid tissues and precursor cells, but complete absence in mature red blood cells, similar to mouse *Trim10* (Fig. EV1A,B). A recent study also confirms *TRIM10* is exclusive to erythroblasts, with little to no expression in other cells (Yang et al, 2024). Reanalysis of RNA sequencing (RNA-seq) and microarray datasets (An et al, 2014b; Merryweather-Clarke et al, 2011b) revealed a stage-specific pattern of TRIM10α expression during erythroid maturation: barely detectable in pro-erythroblasts (Pro-E), upregulated in early and late basophilic erythroblasts (EB and LB), maximal in polychromatic erythroblasts (Poly), and reduced in orthochromatic erythroblasts (Ortho) (Fig. 1A,B). This pattern was recapitulated during ex vivo differentiation of mouse colony-forming unit–erythroid (CFU-E) cells (Fig. EV1C). Based on independent public datasets (Plaza-Florido et al, 2024b; Raghavachari et al, 2009b), we observed that *TRIM10a*, absent in normal red blood cells, was significantly elevated in sickle cell anemia (SCA) patients (Figs. 1C and EV1D). Moreover, reanalysis of single-cell RNA-seq datasets (Doty et al, 2023b) showed that patients with Diamond–Blackfan anemia (DBA), characterized by erythropoiesis defects, exhibit complete suppression of *TRIM10a* expression throughout all stages (Fig. 1D). To explore how *TRIM10a* expression dynamics influence erythroid maturation, we utilized HUDEP-2 cells, widely used for studying terminal erythroid differentiation (Daniels et al, 2020; Doerfler et al, 2021; Feng et al, 2022; Kurita et al, 2013). Culturing the cells in differentiation medium for 4–6, 8, and 12 days yielded EB/LB, Poly, and Ortho/reticulocyte stages, respectively (Fig. 1E). Consistent with previous reports (Daniels et al, 2020), differentiation was validated by stage-specific marker expression: at the EB/LB stage (Day 4–6), cells showed elevated CD36 (thrombospondin receptor) and CD49d (α4-integrin) with minimal CD235a (glycophorin A); at the Poly stage (Day 8), CD36 and CD49d decreased while CD235a increased markedly; and at the Ortho/reticulocyte stage (Day 12), CD36 and CD49d further declined whereas CD235a remained strongly upregulated (Fig. EV1E). Morphological changes, including reduced cell size, nuclear condensation, and increased hemoglobin expression, further confirmed proper differentiation (Fig. EV1F,G). Consistently, RNA-seq analysis of HUDEP-2 cells showed that *TRIM10a* expression was barely detectable from days 0–3, started on day 4, peaked on day 8, and declined by day 12, corresponding to the Pro-E, EB/LB, Poly, and Ortho stages (Fig. 1F). Other *TRIM* genes, including *TRIM58*, showed distinct expression patterns, with significant expression from days 0–8 and a marked increase on day

12, corresponding to the Pro-E to Poly and Ortho stages (Fig. 1F). Stage-dependent *TRIM10a* expression was further validated by volcano plot analysis, comparing days 0 and 8, highlighting highly inducible Poly stage genes, including those related to Hb (Fig. 1G). These findings suggest that the dynamic expression pattern of *TRIM10a* across different stages may play specific roles during each stage of erythroid maturation.

### Erythroid stage-dependent subcellular localization of TRIM10α

TRIM proteins are known to distribute across various cellular compartments, serving diverse biological functions (James et al, 2007; Vunjak and Versteeg, 2019). To elucidate the stage-dependent roles of TRIM10α, we examined its subcellular localization during erythroid differentiation in HUDEP-2 cells. Consistent with the RNA-seq and qPCR results, TRIM10α expression was barely detectable at the Pro-E stage, increased at the EB/LB stage, and subsequently declined during the later stages of differentiation (Fig. 2A,B). Intriguingly, TRIM10α predominantly displayed a cell surface localization pattern in EB/LB-stage cells, with ~30% showing both surface and intracellular localization (enlarged view #1 in Fig. 2A and Fig. 2C). The cell surface localization was also confirmed by flow cytometry and confocal microscopy performed without cell permeabilization (Figs. 2D and EV2A). At the Poly stage, TRIM10α was retained at the cell surface in ~60% of cells, while all cells displayed punctate cytosolic structures distal to the nucleus, suggesting endocytosis-mediated trafficking to endosomal or lysosomal compartments (Fig. 2A,C). Although orthochromatic erythroblasts with nuclei were predominant beyond day 12 in the HUDEP system, both pyrenocytes and enucleated cells remained detectable (Fig. EV2B). Notably, at the Ortho stage, TRIM10α encapsulated pyrenocytes, membrane-wrapped nuclei expelled from erythroblast, and formed punctate structures that partially colocalized with Hb subunit alpha (HbA) in the cytosol of reticulocytes (enlarged view #2-3 of Fig. 2A; pink, pyrenocyte; yellow, reticulocyte). Consistent with these results, TRIM10α underwent extensive oligomerization starting around day 12, potentially driven by the oxidative conditions that frequently occur during the late stages of erythroid maturation (Fig. EV2C). These findings reveal the dynamic, stage-specific subcellular localizations of TRIM10α during erythroid maturation.

Recent structural studies show that TRIM72 interacts with phosphatidylserine-enriched membranes through PRY/SPRY-dependent oligomeric assembly, crucial for membrane targeting (Ma et al, 2023; Park et al, 2023). This enabled us to investigate whether TRIM10α self-associates and targets the plasma membrane. Strong dimerization of TRIM10α was observed using green fluorescent protein (GFP)- and Myc-tagged TRIM10α constructs (Fig. EV2D). AlphaFold-predicted TRIM10α structure showed self-association in the CC domain and lipid interactions in the PRY/SPRY domain, similar to TRIM72 (Fig. 2E). To validate these predictions, we generated TRIM10α mutants lacking the RING (TRIM10αΔRING), B-box (TRIM10αΔBB), coiled-coil (TRIM10αΔCC), or PRY/SPRY (TRIM10αΔPRY/SPRY). The TRIM10αΔCC mutant failed to self-associate, while the TRIM10αΔBB exhibited a pattern similar to that of the wild-type TRIM10α (Fig. 2F). Moreover, both TRIM10αΔCC and TRIM10αΔPRY/SPRY mutants localized mainly to the cytosol instead

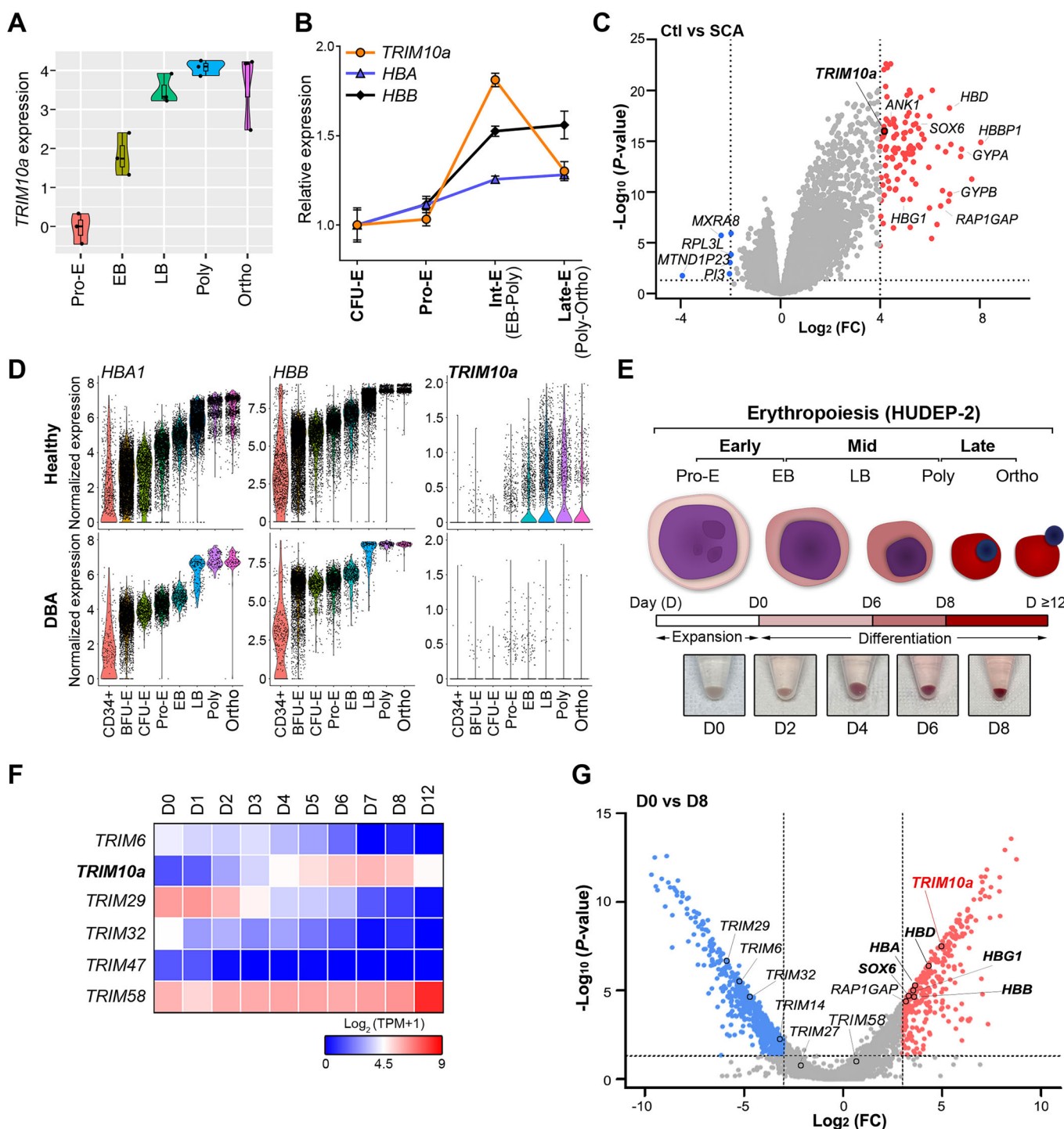

of the plasma membrane (Fig. 2G). Accordingly, surface expression was markedly reduced in the ΔPRY/SPRY mutant, whereas the ΔCC mutant also showed a reduction, though to a lesser extent than ΔPRY/SPRY (Fig. EV2E–G). Little effects on cell surface expression were seen for TRIM10αΔRING or TRIM10αΔBB, also observed in non-erythroid HEK293T cells (Fig. EV2H). Similar to TRIM72, the PRY/SPRY domain of TRIM10α contains several arginine, lysine, and histidine residues, suggesting their positive charges facilitate binding to negatively charged phospholipid

domains, enabling cell surface targeting. These findings highlight the stage-dependent subcellular dynamics of TRIM10α during erythropoiesis.

## Surface TRIM10α captures extracellular C1q to encapsulate pyrenocytes

To investigate the stage-and spatial-specific functions of TRIM10α during erythroid maturation, we identified its binding partners

◄ **Figure 1.  Aberrant expression of erythroblast- and stage-specific *TRIM10a* links anemias.**

(A) Violin plots showing *TRIM10a* expression patterns at different stages of erythropoiesis. Relative mRNA levels of *TRIM10a* were analyzed using public RNA-seq data from different stages of erythropoiesis (An et al, 2014b; Data ref: An et al, 2014a). $n = 3$ biological replicates for each developmental stages; boxplot was defined by minima = 25th percentile – 1.5× IQR, maxima = 75th percentile + 1.5× IQR, center = median and bounds of box = 25th and 75th percentile. (B) Relative mRNA levels of *TRIM10a*, *HBA*, and *HBB* expression in public microarray data across different stages of erythropoiesis (Merryweather-Clarke et al, 2011b; Data ref: Merryweather-Clarke et al, 2011a). CFU-E, colony-forming units erythroid; Int-E, intermediate erythroblast (EB to Poly); Late-E, late erythroblast (Poly to Ortho). Data were presented as mean ± SD ($n = 3$ biological replicates). (C) *TRIM10a* expression is increased in SCA patients. Volcano plots show differentially expressed genes in public RNA-seq data from whole blood of control (Ctl) and sickle cell anemia (SCA) patients (Plaza-Florido et al, 2024b; Data ref: Plaza-Florido et al, 2024a). In the case of Ctl, $n = 17$ healthy age and ethnicity/race-matched controls (14 ± 3 years, 29% girls); In the case of SCA, $n = 23$ patients with SCA (13 ± 3 years, 52% girls). Raw counts were normalized and transformed using limma-voom, and differential gene expression was analyzed using the limma package (linear models adjusted by sex; moderated $t$-test). Genes with $-\log_{10} P$ values > 1.3 and $\log_2$ fold-changes >4 (upregulated, red) or <−2 (downregulated, blue) are indicated. (D) *TRIM10a* expression is suppressed in Diamond–Blackfan anemia (DBA) patients. Violin plots showing single-cell expression of HBA1, HBB, and TRIM10a across distinct stages of human erythropoiesis in controls (top) and DBA patients (bottom), based on public scRNA-seq data (Doty et al, 2023b; Data ref: Doty et al, 2023a). In the case of Healthy, $n = 996, 8503, 2437, 3379, 2554, 4548, 1408,$ and 1388 single cells for CD34+, BFU-E, CFU-E, Pro-E, EB, LB, Poly, and Ortho were represented in the Violin plot, respectively; In the case of DBA, $n = 369, 4471, 855,$ 2290, 670, 208, 156, and 107 single cells for CD34+, BFU-E, CFU-E, Pro-E, EB, LB, Poly, and Ortho were represented in the Violin plot, respectively. CD34+, CD34$^+$ cell; BFU-E, burst-forming units erythroid. (E) Schematic representation of the HUDEP-2 culture protocol with representative images of pelleted HUDEP-2 cells cultured in differentiation medium for the indicated times. Early, early stage-erythroblasts (Pro-E to EB); Mid, mid stage-erythroblasts (EB to Poly); Late, late-stage-erythroblasts (Poly to Ortho). (F) *TRIM10a* exhibits stage-specific expression during erythropoiesis. Heatmap showing the relative expression of *TRIM* genes from RNA-seq data in HUDEP-2 cells before (day 0) and after differentiation (at the indicated time points). Red and blue colors represent high and low fold change, respectively. (G) Volcano plots representing genes with differential expression between HUDEP-2 cells before (day 0) and after (day 8) differentiation ($n = 2$ biological replicates). Differential expression analysis was performed using the edgeR package with TMM normalization (exact test). Genes with $-\log_{10} P$ values >1.3 and $|\log_2 \text{FC}| \geq 3$ are indicated by red dots (upregulated) and blue dots (downregulated). Data information: Unless otherwise stated, data shown are mean ± SD of three biological replicates. Source data are available online for this figure.

using liquid chromatography-tandem mass spectrometry (LC-MS/MS)-based proteomics on HUDEP-2 cells. We analyzed cells from days 0, 8, and 12, corresponding to the Pro-E, Poly, and Ortho/reticulocyte stages, which reflect low, high, and moderate TRIM10α expression, respectively (Fig. EV3A). Only proteins consistently identified as TRIM10α interactors across replicates were considered, yielding 136, 134, and 130 proteins at D0, D8, and D12. Interaction strength was quantified by spectral coverage, and the data were visualized with refined color coding to distinguish interaction categories (Fig. EV3B). Proteomics analysis revealed that TRIM10α binds to cytoskeletal and erythroid-related proteins, indicating roles in enucleation and other cellular functions during erythroid maturation (Fig. EV3B). To further minimize non-specific interactions, proteins identified in undifferentiated cells (D0), where TRIM10α is barely detectable, were excluded from the final data set. Notably, TRIM10α strongly bound to soluble complement component 1q proteins (C1qB and C1qC) at days 8 and 12, but not at day 0 (Figs. 3A and EV3B). C1q, the first complement component that mediates the engulfment of apoptotic cells (Ogden et al, 2001; Paidassi et al, 2008), suggests TRIM10α may be involved in erythroid homeostasis. Co-immunoprecipitation (IP) experiments with HUDEP-2 cells at days 0, 4, 8, and 12 confirmed that TRIM10α binding to C1q began ay day 4 and significantly increased on days 8 and 12 (Fig. 3B). While C1q oligomerized after day 8, TRIM10α bound only to the monomeric form of C1q (Fig. 3B, arrow, oligomer; arrowhead, monomer). Of note, HUDEP-2 cells showed little to no expression of C1qA, C1qB, or C1qC across differentiation stages, suggesting that the C1q interacting with TRIM10α originated from extracellular sources such as serum rather than being synthesized intracellularly (Fig. EV3C). Since the C1q complex consists of C1qA, C1qB, and C1qC in a fixed stoichiometric ratio (Thielens et al, 2017), the robust detection of C1qB and C1qC in the proteomic analysis indicates that TRIM10α interacts with the C1q complex, even though C1qA was not detected above the threshold.

To investigate how extracellular C1q is present in cells, we examined its distribution on days 0, 4, 8, and 12 of differentiation. Notably, C1q was detected on the cell surface and partially localized within cytosolic punctate structures on days 4 and 8, with pyrenocyte encapsulation evident on day 12, which was similar to TRIM10α (Fig. 3C, compared with Fig. 2A). Flow cytometry analysis of days 0 and 8 confirmed that both surface and intracellular C1q were significantly detectable on day 8 (Figs. 3D,E and EV3D,E). In addition, C1q colocalized with TRIM10α across differentiation and was detected in punctate cytosolic structures of reticulocytes, where partial overlap with TRIM10α was observed (Fig. 3C, white arrow). This colocalization was further supported by quantitative analysis (right panel of Fig. 3C,F,G). These findings prompted further investigation into the effect of TRIM10α loss on C1q entry into HUDEP-2 cells. Intracellular C1q levels were nearly undetectable in TRIM10α-depleted cells, with a concomitant reduction in cellular redness and overall cell numbers (Figs. 3H and EV3F–H). These findings suggest that TRIM10α is critical for capturing extracellular C1q and facilitating its cellular entry or pyrenocyte encapsulation during late erythroid maturation.

## C1q-coated pyrenocytes serve as an "Eat-Me" signal to macrophages

C1q binds to the surface of apoptotic cells or harmful particles and interacts with C1q receptors on macrophages, promoting their engulfment and phagocytosis, thereby ensuring proper clearance and maintenance of cellular homeostasis (Bobak et al, 1987; Bohlson et al, 2007; Donat et al, 2019; Fraser et al, 2009). In erythropoiesis, macrophages are also involved in regulating erythroid cell maturation and removing pyrenocytes (Li et al, 2019; Toda et al, 2014). Along with previous reports, our findings led us to speculate that surface C1q-coated pyrenocytes enhance recognition and attachment by macrophages, thereby facilitating

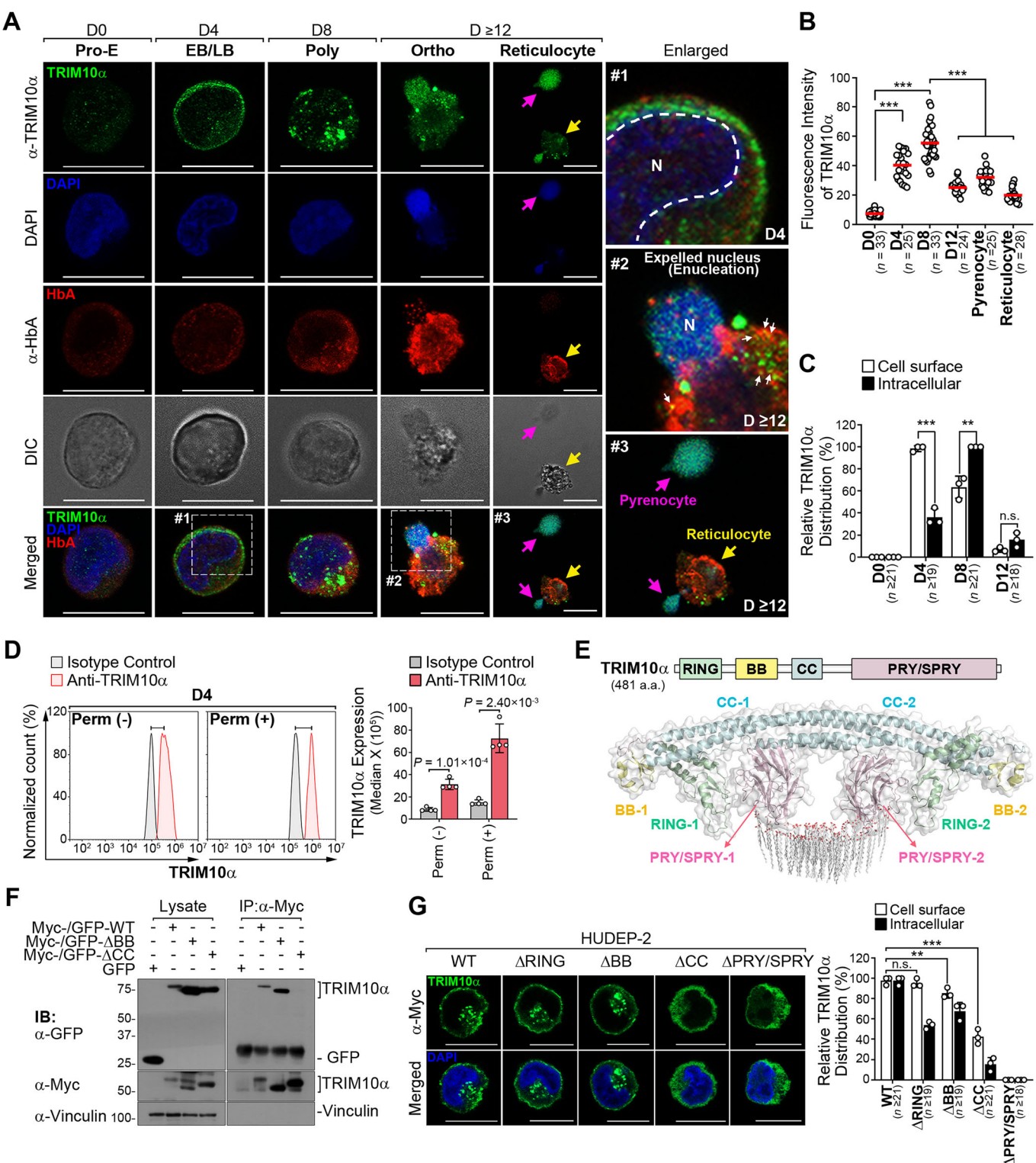

their engulfment and clearance. To test this, we differentiated U937 cells into macrophages using phorbol 12-myristate 13-acetate stimulation and labeled them with a macrophage-specific red tracker (Fig. 4A). HUDEP-2 cells were differentiated into orthochromatic erythroblasts undergoing nuclear extrusion in either normal or C1q-depleted medium and stained with Hoechst

to monitor maturation, as C1q depletion did not affect cell survival or Hb protein levels during erythroid differentiation (Figs. 4A and EV4A,B). After co-culturing the two cell types for 5 h, we assessed macrophage attachment to erythroblasts and subsequent engulfment. Although pyrenocytes were relatively scarce, all observed pyrenocytes were TRIM10α-positive, providing a

**Figure 2.  Erythroid stage-dependent subcellular localization of TRIM10α.**

(A) Stage-specific spatial changes in TRIM10α localization during erythropoiesis. Confocal fluorescence images of HUDEP-2 cells before (Pro-E) and after differentiation (EB/LB, Poly, and Ortho, Reticulocyte) at the indicated time points. Cells were stained with anti-TRIM10α antibody (TRIM10α, green), anti-HbA antibody (HbA, red), and nuclei (N) were stained with DAPI (blue). Enlarged views of the regions are indicated by the white dashed squares. White arrows in image #2 represent the colocalization of TRIM10α and HbA. Pink and yellow arrows in image #3 represent the pyrenocyte and reticulocyte, respectively. Images are orthogonal projections of Z-stacks. Scale bars, 10 μm. (B) Quantification of TRIM10α fluorescence intensity. The fluorescence intensity of TRIM10α in each cell shown in (A) was quantified. Each dot represents a single cell, and horizontal bars indicate the median ($n$ = number of analyzed cells). $P$ values were determined by an unpaired two-tailed $t$-test. ***$P < 0.001$. (C) Quantification of subcellular TRIM10α distribution during erythropoiesis. The bar graph shows the relative percentage of TRIM10α localized to the cell surface (white) or intracellular compartment (black) in HUDEP-2 cells before (D0) and after differentiation (D4, D8, and D12). Data were derived from the analysis of multiple cells ($n$ = number of analyzed cells). Data represent mean ± SD of three biological replicates. $P$ values were determined by an unpaired two-tailed $t$-test. ***$P < 0.001$. (D) TRIM10α localization on the cell surface. Flow cytometry analysis of TRIM10α expression in non-permeabilized (Perm(−)) and permeabilized (Perm(+)) HUDEP-2 cells cultured in differentiation medium on day 4. The graphs on the right depict the median fluorescence intensity of TRIM10α. Data were presented as mean ± SD ($n$ = 4 biological replicates). $P$ values were determined by an unpaired two-tailed $t$-test. (E) AlphaFold-predicted structure of TRIM10α. Domains are color-coded as follows: RING (green), B-box (yellow), coiled-coil (blue), and PRY/SPRY (pink). (F) The coiled-coil domain of TRIM10α is required for its dimer formation. Co-immunoprecipitation analysis showing the interaction between TRIM10α, each tagged with a different epitope. WT wildtype, ΔBB B-box deletion mutants, ΔCC coiled-coil deletion mutants. (G) TRIM10α is localized on the cell surface via the coiled-coil and PRY/SPRY domains. Confocal fluorescence images of HUDEP-2 cells expressing Myc-tagged WT-TRIM10α and TRIM10α deletion mutants on day 0. Cells were stained with anti-Myc antibody (TRIM10α, green), and nuclei were stained with DAPI (blue). Images are orthogonal projections of Z-stacks. Scale bars, 10 μm. ΔRING, RING deletion mutants; ΔPRY/SPRY, PRY/SPRY deletion mutants. The graph on the right depicts the relative distribution of TRIM10α on the cell surface (white) versus in the intracellular compartment (black) of HUDEP-2 cells at different stages, including before (D0) and after differentiation (D4, D8, and D12). Data were derived from the analysis of multiple cells ($n$ = number of analyzed cells). Data represent mean ± SD of three biological replicates. $P$ values were determined by an unpaired two-tailed $t$-test. n.s., $P > 0.05$; **$P < 0.01$; ***$P < 0.001$. Data information: Unless otherwise stated, data shown are mean ± SD of three biological replicates; statistical significance was determined by unpaired two-tailed $t$-test. (n.s., $P > 0.05$; **$P < 0.01$; ***$P < 0.001$). Source data are available online for this figure.

sufficient basis for evaluating macrophage–pyrenocyte interactions and engulfment (Figs. 2D and 4B). Consistently, quantitative analysis revealed that ~80% of macrophages attached to nucleated cells or pyrenocytes, whereas ~20% progressed to engulfment (Fig. 4B,C). Notably, C1q depletion significantly impaired macrophage attachment to erythroblasts, whereas efficient binding was observed under normal conditions (Fig. 4D, right graph). As TRIM10α is critical for capturing extracellular C1q and encapsulating C1q-coated pyrenocytes (Fig. 3), we next examined whether TRIM10α influences macrophage attachment. TRIM10α-depleted HUDEP-2 cells displayed a marked reduction in macrophage–erythroid association relative to control cells (Fig. 4E, right graph). Despite the knockdown efficiency not being high, this reduction was comparable to that observed with C1q depletion (Fig. 3H). Taken together, surface TRIM10α-bound extracellular C1q coats pyrenocytes, acting as an "eat-me" signal for macrophages to facilitate their recognition and clearance.

## Surface C1q reduces protein synthesis by accelerating lysosomal targeting of EpoR

To understand the intracellular roles of C1q-bound TRIM10α during mid-to-late erythroid differentiation, we revisited the mass-spectrometry datasets to gain further clues. TRIM10α binds to various protein synthesis-related proteins, including ribosomal subunits and translational elongation factors, with dynamic binding patterns observed across stages (Fig. EV3B). This suggests that the C1q complex may regulate protein synthesis during erythroid differentiation. Specifically, EpoR activation triggers a phosphorylation cascade involving key effectors, such as phosphatidylinositol 3-kinases (PI3K), protein kinase B (PKB, also known as AKT), mammalian target of rapamycin complex 1 with regulatory-associated protein of mTOR (mTORC1-Raptor), mTORC2 with rapamycin-insensitive companion of mTOR (mTORC2-Rictor), and ribosomal protein S6 kinase beta-1 (S6K), ultimately promoting protein synthesis (Elliott and Sinclair, 2012; Forester et al, 2022; Valent et al, 2018) (Fig. 5A). To investigate

the relationship between TRIM10α-C1q and the EpoR signaling pathway, we screened protein expression associated with EpoR and its downstream components. In HUDEP-2 cells differentiated to the Poly stage (day 8), protein levels of key factors, including PI3K, AKT, mTOR, Raptor, Rictor, and S6K, were significantly decreased; however, C1q depletion failed to inhibit the EpoR signaling cascade, as observed by the prolonged expression of these proteins (Fig. 5B, upper panels). Accordingly, EpoR-mediated phosphorylation of PI3K, AKT, mTOR, and S6K was significantly downregulated under normal conditions, but remained persistently active in C1q-depleted cells (Fig. 5B, middle panels). Importantly, in contrast to control cells, EpoR protein levels were significantly increased under C1q-depleted conditions (Fig. 5B, bottom panels). Heat-shock protein 70 (HSP70), known to remain stable during erythroid differentiation (Mathangasinghe et al, 2021), remained unchanged regardless of C1q (Fig. 5B). These findings led us to investigate whether C1q binds directly to EpoR and observed clear interaction and colocalization of EpoR with C1q and TRIM10α (Figs. 5C and EV4C). Our findings suggest that C1q interacts with the EpoR, reducing its protein levels and attenuating the signaling cascade.

High EpoR expression promotes erythroid progenitor proliferation, but its endocytosis-mediated signaling is downregulated after the Poly stage and absent in reticulocytes, crucial for optimizing erythroid maturation and homeostasis (Bulut et al, 2013; Elliott and Sinclair, 2012; Sulahian et al, 2009) (Fig. 5A). During mid-to-late HUDEP-2 differentiation, surface TRIM10α–C1q complexes relocated to intracellular punctate structures (Figs. 2 and 3), prompting us to investigate the role of C1q in the lysosomal targeting of EpoR via endocytosis. Notably, chloroquine-mediated lysosomal inhibition increased EpoR levels, supporting the notion that EpoR is targeted to lysosomes during erythropoiesis (Fig. 5D). Similarly, blockade of endocytosis with Dynasore increased both C1q and EpoR levels and reversed the C1q-mediated reduction of EpoR (Fig. 5E). Moreover, whereas EpoR colocalized with lysosomes under normal conditions, C1q depletion resulted in its retention at the cell surface and defective lysosomal targeting (Fig. 5F–H). These findings indicate that C1q promotes EpoR degradation via an

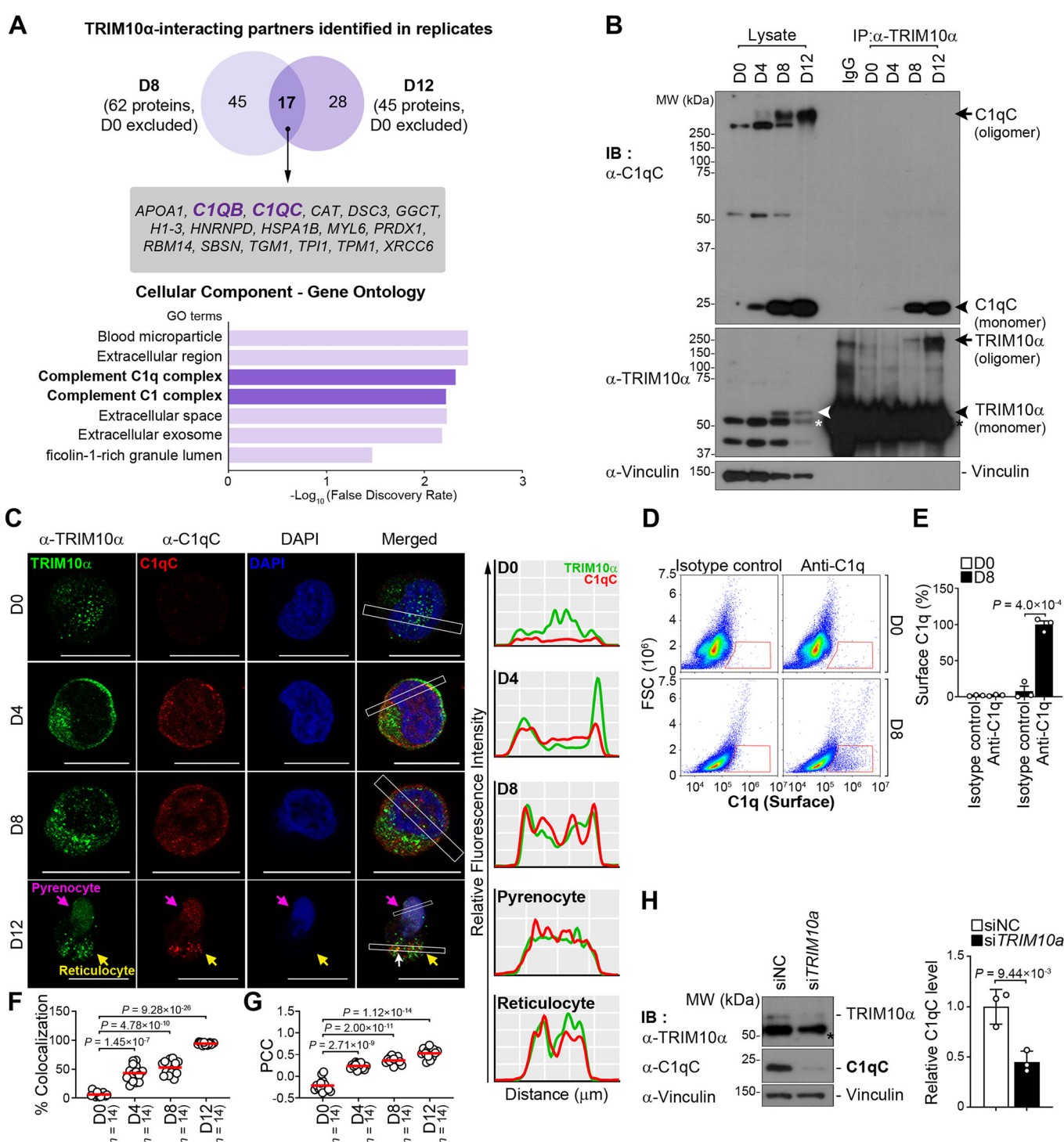

endocytic–lysosomal pathway. Intriguingly, TRIM10α levels increased in C1q-depleted cells, while HbA levels remained unchanged (Fig. 5I). This pattern closely resembled that of downstream EpoR signaling components, suggesting that impaired C1q-mediated endocytosis prevents surface TRIM10α from being targeted to lysosomes. Thus, surface C1q, rather than TRIM10α, likely drives EpoR endocytosis. Taken together, C1q-bound TRIM10α interacts with EpoR, directing it to lysosomal

degradation and maintaining protein synthesis homeostasis during erythroid maturation.

## Stage-dependent dual roles of TRIM10α in Hb formation and clearance

Cytosolic punctate structures of TRIM10α, which did not colocalize with C1q or EpoR (Figs. 3 and EV4C), suggested that TRIM10α

**Figure 3. Surface TRIM10α captures extracellular C1q to encapsulate pyrenocytes.**

(A) Venn diagram showing the overlap of TRIM10α-binding proteins identified in HUDEP-2 cells cultured in differentiation medium at the indicated time points across biological replicates. A total of 17 proteins were detected at both D8 and D12, and STRING-based gene ontology (GO) enrichment analysis revealed significant enrichment for the complement C1q complex. (B) TRIM10α binds to C1q during erythropoiesis. Co-immunoprecipitation analysis of the interaction between TRIM10α and C1qC in HUDEP-2 cells cultured in differentiation medium for the indicated times. Cell extracts were analyzed using the indicated antibodies. Arrows and arrowheads represent the oligomer and monomer, respectively. The asterisk (*) marks the non-specific bands. (C) C1q localization is similar to TRIM10 localization during erythropoiesis. Confocal fluorescence images of HUDEP-2 cells before (D0) and after (D4, D8, and D12) differentiation. Cells were stained with anti-TRIM10α antibody (TRIM10α, green), anti-C1qC antibody (C1qC, red), and nuclei were stained with DAPI (blue). Pink and yellow arrows in D12 represent the pyrenocytes and reticulocytes, respectively. White arrows in D12 represent the colocalization of C1qC with TRIM10α. Images are orthogonal projections of Z-stacks. Right graphs represent the quantification of colocalization between TRIM10α (green) and C1qC (red). Scale bars, 10 μm. (D) C1q is localized on the cell surface during erythropoiesis. Flow cytometry analysis of C1q in HUDEP-2 cells before (D0) and after (D8) differentiation. Plots show the counted number of surface C1q$^+$ cells. (E) Graphs showing the percentage of counted C1q$^+$ cell numbers of (D). Data were presented as mean ± SEM ($n = 3$ biological replicates). $P$ values are determined by an unpaired two-tailed $t$-test. (F) The graph shows the quantification of colocalization between TRIM10α and C1qC in (C). Colocalization was quantified using Manders' overlap coefficient (MOC) after background subtraction. Each dot represents the value from an individual cell, and horizontal bars indicate the median ($n$ = number of analyzed cells). $P$ values were determined by an unpaired two-tailed $t$-test. (G) The graph shows the Pearson's correlation coefficient (PCC) values for TRIM10α and C1qC in (C). Colocalization was quantified using PCC after background subtraction. Each dot represents the PCC value of an individual cell, and horizontal bars indicate the median ($n$ = number of analyzed cells). $P$ values were determined by an unpaired two-tailed $t$-test. (H) Extracellular C1q is directly captured by TRIM10α. Immunoblot analysis of TRIM10α or C1qC from whole cell lysates of HUDEP-2 cells transfected with non-targeting control siRNA (siNC) or siRNA against $TRIM10a$ (si$TRIM10a$). Cells were transfected on day 6 and differentiated for an additional 48 h before harvesting on day 8. Vinculin was used as a loading control. The asterisk (*) indicates non-specific bands. The graph on the right shows the relative C1qC protein levels from western blot analysis of siNC (white) and siTRIM10α (black). Data were presented as mean ± SD ($n = 3$ biological replicates). $P$ values were determined by an unpaired two-tailed $t$-test. Data information: Unless otherwise stated, data shown are mean ± SD of three biological replicates; statistical significance was determined by unpaired two-tailed $t$-test. Source data are available online for this figure.

may exist freely in the cytoplasm. Indeed, monomeric TRIM10α and C1q were detected in the cytoplasmic fraction of HUDEP-2 cells at the Poly stage (Fig. EV4D). This led us to revisit our mass spectrometry datasets to identify potential cytosolic binding partners of TRIM10α. We identified HbA and HbB as TRIM10α binding partners, along with several chaperones, including heat-shock protein 90 (HSP90), which facilitates heme insertion by promoting its association with globin partners (Ghosh et al, 2018; Stuehr et al, 2022) (Fig. EV3B). We previously observed partial colocalization of HbA with TRIM10α in cytosolic punctate structures during the Poly and Ortho stages, as well as in reticulocytes (Fig. 2A), speculating that TRIM10α may contribute to Hb maturation. We confirmed the colocalization of TRIM10α with HbA and HbB in Poly stage HUDEP-2 cells (day 8), where both globins were dispersed and partially colocalized with TRIM10α in the cytosolic regions opposite the nucleus (Fig. 6A–C). Terminal erythropoiesis involves tightly regulated Hb maturation through globin chain interactions and heme insertions (Ghosh et al, 2018; Stuehr et al, 2022; Tupta et al, 2022). To investigate the effect of TRIM10α on Hb-heme interactions, we examined PAGE-resistant Hb dimers, which are absent under heme-depleted conditions [53]. Despite similar monomeric levels, we observed a significant reduction in dimeric forms of both HbA and HbB in $TRIM10a$-depleted cells, suggesting that TRIM10α facilitates Hb-heme interactions during the Poly stage (Fig. 6D).

TRIM10α undergoes extensive oligomerization, resulting in the reduction of its monomeric form and the formation of larger cytosolic punctate structures, both of which appear in reticulocytes around day 12 (Figs. 2 and EV2C). As globin-enriched cytosolic puncta increased toward the late stages of erythropoiesis (Fig. 2), we examined whether HbA was ubiquitinated. When HUDEP-2 cell lysates from D0 and D12 were IP with an anti-HbA antibody and probed with an anti-ubiquitin antibody, no increase in HbA ubiquitination was detected at Day 12 compared with Day 0 (Fig. 6E). These findings suggest that the high-molecular-weight HbA species are more likely oligomeric forms of HbA rather than ubiquitinated species. Intriguingly, in reticulocytes, TRIM10α aggregates colocalized with globin aggregates and p62,

suggesting their potential targeting to lysosomes (Fig. 6F, arrows). We next examined whether oligomerized TRIM10α preferentially associates with monomeric/dimeric Hb or oxidized Hb, as oxidative stress often generates harmful aggregates during late erythroid maturation. Notably, TRIM10α selectively bound to high-molecular-weight aggregates of oxidized HbA on day 12, with no significant interaction with oxidized HbB (Fig. 6G). Consistent with these results, oligomeric TRIM10α strongly colocalized with larger HbA- or HbB enriched punctate structures only in reticulocytes on day 12 (Fig. 6H, arrows). Markedly, $TRIM10a$-depleted reticulocytes did not exhibit punctate structures of HbA or HbB nor did they display any oligomerized TRIM10α (Fig. 6H). To further understand the role of oligomeric TRIM10α in managing oxidized HbA during late erythroid maturation, we focused on its quality control mechanisms as its expression rapidly declines in reticulocytes (Figs. 1 and EV2C). We investigated whether oligomeric TRIM10α undergoes self-ubiquitination via its own E3 ligase activity. Notably, we observed significant accumulation of TRIM10α-ubiquitin conjugates, with confocal microscopy showing colocalization of TRIM10α and ubiquitin (Fig. 6I,J). Of note, the TRIM10α mutant lacking the RING domain (TRIM10αΔRING) failed to induce ubiquitination, indicating that TRIM10α self-ubiquitinates via its RING domain (Fig. 6I). To ascertain whether ubiquitinated TRIM10α undergoes degradation or functional enhancement, we examined its colocalization with p62, a selective autophagy receptor that recognizes and targets ubiquitinated substrates to autophagosomes for degradation. TRIM10α colocalized with p62 and ubiquitin (Fig. 6J), but this colocalization was abolished in ΔRING-expressing cells. These results suggest that oligomeric TRIM10α sequesters deleterious Hb aggregates under oxidative conditions, thereby promoting their p62-dependent autophagic clearance during late erythroid maturation.

## Suppression of alternatively spliced *TRIM10b* expression in erythroid homeostasis

Alternative splicing events are increasingly prominent in the later stages of erythropoiesis (Pimentel et al, 2014; Yamamoto et al, 2009).

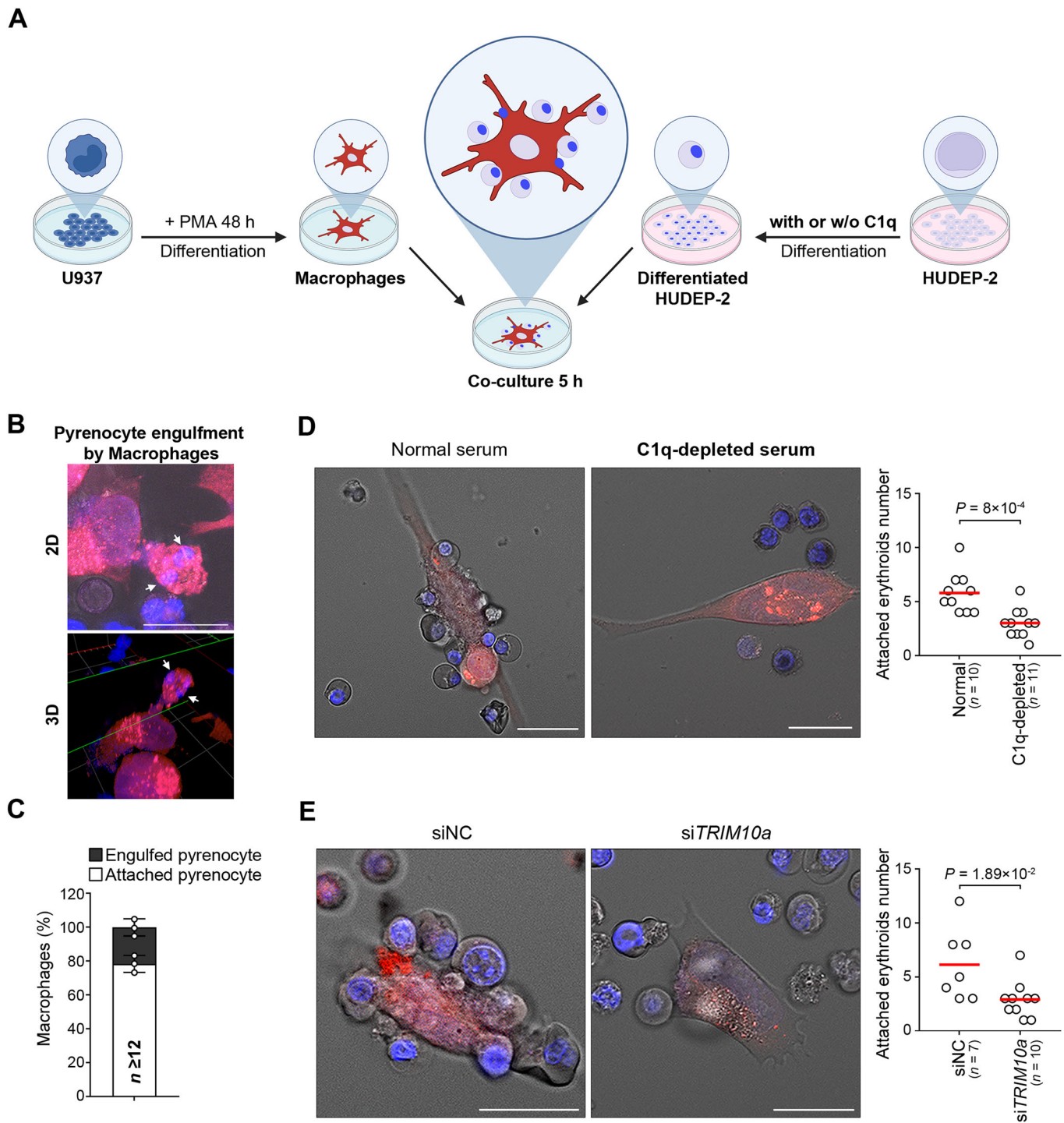

TRIM10b in humans, which is absent in mice, arises from alternative splicing that shortens the PRY/SPRY domain and inserts a 27-amino acid sequence with three cysteine residues (Fig. 7A, upper), though its functional role remains unexplored. Unlike TRIM10a, RNA-seq analysis showed that TRIM10b was negligible at all stages of HUDEP-2 differentiation (Fig. 7A, bottom). To understand why TRIM10b expression is suppressed during erythroid maturation, we ectopically expressed GFP-tagged TRIM10α or TRIM10β in HUDEP-

2 cells and examined their subcellular localization. Consistent with endogenous TRIM10α during the Poly stage, ectopic TRIM10α localized to the cell surface, cytosolic punctate structures opposite the nucleus, and near the microtubule-organizing center. In contrast, TRIM10β dispersed in the cytosol and formed inclusion bodies or aggregates when highly expressed in HUDEP-2 cells (Figs. 7B and EV5A). These aggregates were linked to loss of cell adhesion and reduced cell size (Fig. 7B; third image). This phenomenon was also

**Figure 4. C1q-coated pyrenocytes as an "Eat-Me" signal to macrophages.**

(A) A schematic representative image for the co-culture of differentiated HUDEP-2 cells and macrophages (image created with BioRender.com). Macrophages were stained with CellTracker Red CMTPX (red), and the nuclei of HUDEP-2 cells were stained with Hoechst 33258 (blue) before co-culture. A detailed protocol is provided in the Methods section. (B) Pyrenocyte engulfment by macrophages. Representative images show macrophages (red) co-cultured with HUDEP-2 cells (nuclei, blue) after 12 days of differentiation. Left, 2D image; right, 3D reconstruction. White arrows indicate engulfed pyrenocytes within macrophages. Scale bars, 10 μm. (C) Population of macrophages interacting with pyrenocytes of HUDEP-2 cells at day 12 of differentiation. A stacked bar graph shows the percentage of macrophages associated with pyrenocytes after co-culture with HUDEP-2 cells differentiated for 12 days. Macrophages with attached pyrenocytes (white) and macrophages with engulfed pyrenocytes (dark gray) are indicated. Data were derived from the analysis of multiple cells ($n$ = number of analyzed cells). Data represent mean ± SD of three biological replicates. (D) C1q is required for erythroid-macrophage attachment. Images show macrophages (red) co-cultured with HUDEP-2 cells (nuclei, blue) differentiated with either normal serum or C1q-depleted serum for day 12. Scale bars, 10 μm. The right graph shows the number of erythroid cells attached to a single macrophage, after differentiation with either normal serum or C1q-depleted serum for day 12. Each dot represents an individual cell, and horizontal bars indicate the median ($n$ = number of analyzed cells). $P$ values were determined by an unpaired two-tailed $t$-test. (E) TRIM10α-bound extracellular C1q is required for erythroid-macrophage attachment. Images show macrophages co-cultured with HUDEP-2 cells transfected with either siNC or siTRIM10α. HUDEP-2 cells were transfected on day 6 and differentiated for 72 h, followed by a second transfection on day 9 and further differentiation for an additional 72 h until day 12. Scale bars, 10 μm. The right graph shows the number of erythroid cells attached to a single macrophage in differentiated HUDEP-2 cells transfected with either siNC or siTRIM10α. Each dot represents the value from an individual cell, and horizontal bars indicate the median value ($n$ = number of analyzed cells). $P$ values are determined by an unpaired two-tailed $t$-test. Data information: Unless otherwise stated, data shown are mean ± SD of three biological replicates; statistical significance was determined by unpaired two-tailed $t$-test. Source data are available online for this figure.

observed in non-erythroid HEK293T cells and increased with higher ectopic TRIM10β expression levels (Fig. EV5B). Consistent with these findings, HUDEP-2 cells ectopically expressing TRIM10β showed reduced viability due to caspase-3 and PARP1 cleavage, indicating apoptosis activation (Fig. 7C,D). These results were consistent with those of confocal microscopy, which showed clear activation of caspase-3 (Fig. 7E). Taken together, TRIM10β forms deleterious aggregates, accelerating cell death.

We next investigated whether TRIM10β aggregates undergo self-ubiquitination and associate with p62 for autophagic targeting, and observed the colocalization of TRIM10β with ubiquitin, p62, and microtubule-associated protein 1 light chain 3 (LC3), a marker for autophagosomes (Fig. 7F). To determine how TRIM10β forms harmful aggregates, unlike TRIM10α, we investigated the role of the PRY/SPRY domain in TRIM10β aggregate formation as this domain differs from that of TRIM10α. To test this hypothesis, we generated TRIM10β mutants lacking the RING (TRIM10βΔRING), B-box (TRIM10βΔBB), coiled-coil (TRIM10βΔCC), and PRY/SPRY (TRIM10βΔP/S) domains (Fig. 7G). Consistent with confocal microscopy analysis, high-molecular weight bands of wild-type TRIM10β were strongly detected in non-reducing SDS-PAGE gels (Fig. 7H), suggesting the presence of a larger complex formed through cysteine-dependent inter-disulfide bonds in its unique C-terminal region, likely aggregates or inclusion bodies (Fig. 7A; red arrows, cysteine residues). In contrast, TRIM10βΔPRY/SPRY protein failed to form its aggregates, whereas the other mutants retained aggregate capabilities (Fig. 7H). Consistent with this, TRIM10βΔPRY/SPRY showed a markedly reduced ability to form aggregates, while other mutants induced larger cytosolic aggregates (Figs. 7I and EV5C). In addition, the TRIM10βΔPRY/SPRY protein failed to bind to p62 and lost its capability to self-ubiquitinate via its RING domain (Fig. EV5D,E). Taken together, TRIM10β forms deleterious aggregates in both erythroid and non-erythroid cells, indicating that its evolutionary suppression protects cells from proteotoxic stress.

## Discussion

Stage-specific gene expression is crucial for efficient resource use, energy conservation, and optimal physiological function. When these genes are expressed at improper stages or inappropriately, they can contribute to various diseases. A key example is the TRIM gene family, which is expressed only when necessary, helping organisms adapt to changing environments. For instance, the stage-specific expression of TRIM5a blocks viral entry during infection and is then downregulated to prevent unnecessary immune responses (Stremlau et al, 2004; Zhao et al, 2023). Similarly, inappropriate expression of epigenetically silenced TRIM40 in the colonic epithelium has been linked to inflammatory bowel disease (Kang et al, 2023). While TRIM28 and TRIM58 are proposed to regulate erythroid maturation, the stage-specific expression of TRIM genes during erythropoiesis remains poorly understood (Hosoya et al, 2013; Thom et al, 2014). Our study identifies stage-specific TRIM10α as a key regulator in optimizing erythroid maturation. TRIM10α self-associates for surface targeting, capturing extracellular C1q and encapsulating pyrenocytes for macrophage-mediated clearance. C1q-bound TRIM10α blocks the EpoR signaling cascade by inducing lysosomal degradation, disrupting protein synthesis in late erythroid maturation. In the cytosol, TRIM10α drives Hb dimer formation and sequesters harmful Hb aggregates. Unlike TRIM10α, the alternatively spliced TRIM10β forms harmful aggregates, suggesting its evolutionary suppression protects erythroid homeostasis. Thus, precise regulation of TRIM10 expression is crucial to preventing erythroid-related diseases like SCA or DBA.

In addition to interactions with C1q and Hb, our mass spectrometry analysis identified tubulins, actins, intermediate filaments, and myosin family members as TRIM10α binding partners during the Pro-E stage. However, many of these interactions diminish during the late stages (Fig. EV3B). Given that cell cycle arrest and enucleation occur predominantly in the later stages of erythroid maturation, these findings suggest that TRIM10α regulates the cell cycle or enucleation through stage-specific interactions with cytoskeletal components during erythropoiesis. In particular, we identified KIF20B, a kinesin-6 family member, as a TRIM10α binding partner. KIF20B regulates cytokinesis during the M phase by localizing to the cleavage furrow and midbody (Janisch et al, 2018), suggesting that TRIM10α may collaborate with KIF20B to support cytokinesis in undifferentiated cells. TRIM10α also interacts with chaperones and heterogeneous

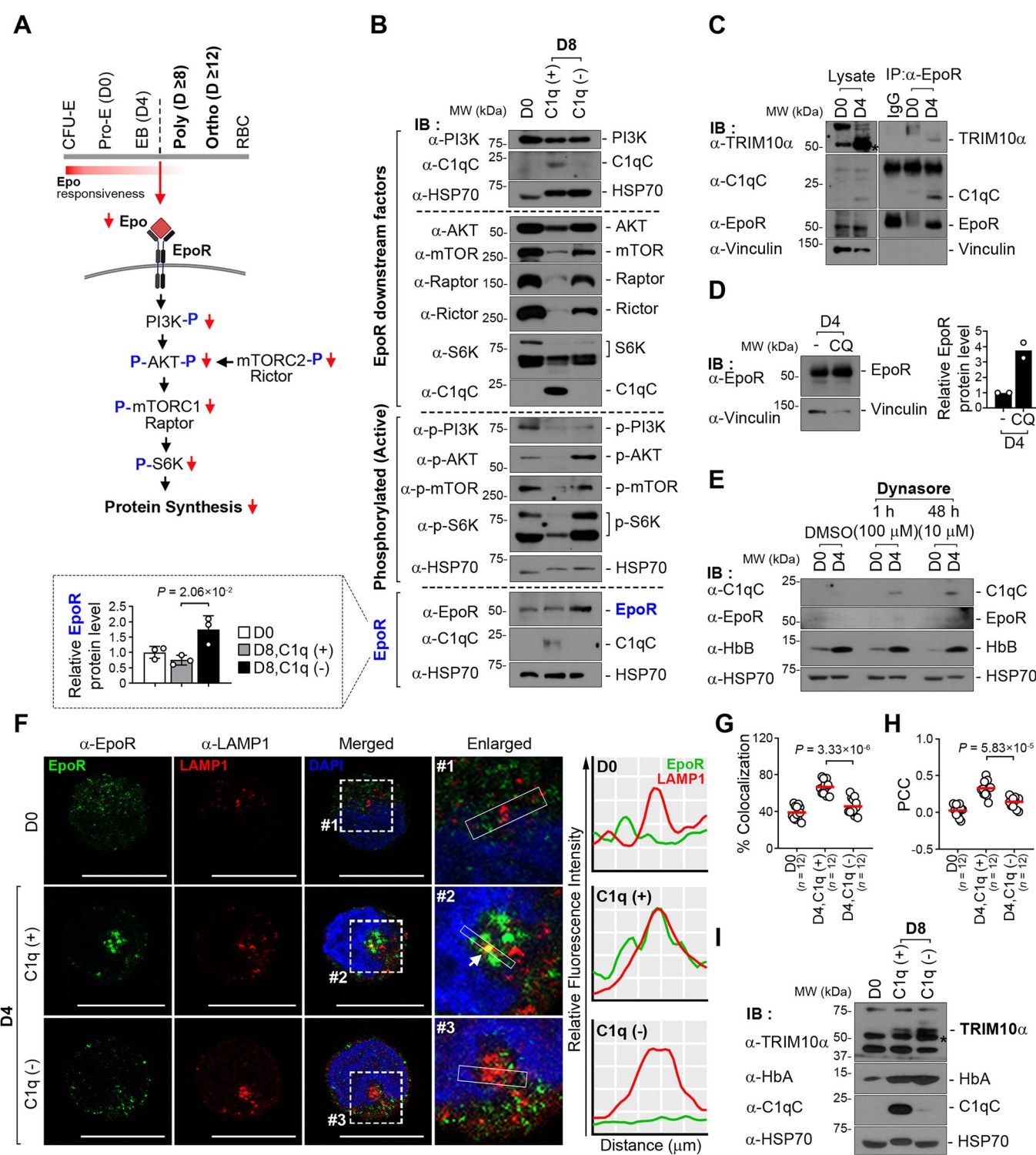

nuclear ribonucleoproteins (HNRNPs) during erythroid maturation, suggesting diverse functional roles through its E3 ligase activity or other mechanisms across different subcellular locations.

C1q, an interacting partner of pleckstrin 2 in erythroid cells, is significantly upregulated in macrophages of erythroblastic islands (Li et al, 2019; Zhao et al, 2014); however, its role in erythropoiesis has not been elucidated. Our study provides the first evidence for

stage-dependent functional roles of C1q during erythroid maturation. In addition to its classical complement functions, recent studies have described non-canonical roles of C1q, including macrophage-mediated engulfment, modulation of signaling pathways, and neuronal elimination (Benavente et al, 2020; Dejanovic et al, 2022; Donat et al, 2019; Fraser et al, 2009; Holden et al, 2021). Furthermore, C1q associates with RNPs to regulate mitochondrial

Figure 5.   **Surface C1q attenuates protein synthesis by accelerating lysosomal targeting of EpoR.**

(A) Schematic representation of EpoR-responsive signaling events that regulate protein synthesis during terminal erythroid differentiation. (B) Protein levels of key factors in EpoR signaling under normal serum or C1q-depleted serum differentiation conditions. Immunoblot analysis of EpoR downstream factors (upper panels), their phosphorylated active forms (middle panels), and EpoR (bottom panels) in whole cell lysates of HUDEP-2 cells before (D0) or after (D8) differentiation with normal serum or C1q-depleted serum. HSP70 was used as a loading control. The left graph shows the relative EpoR protein levels from western blot analysis. Data were represented as mean ± SD ($n = 3$ biological replicates). P values are determined by an unpaired two-tailed t-test. (C) EpoR binds to TRIM10α and C1q. Co-immunoprecipitation analysis of the interaction of EpoR to TRIM10α and to C1qC in HUDEP-2 cells cultured in differentiation medium for the indicated times. Cell extracts were analyzed using the indicated antibodies. The asterisk (*) marks the non-specific bands. (D) EpoR accumulates upon lysosomal inhibition. Immunoblot analysis of EpoR in whole cell lysates of HUDEP-2 cells differentiated for 8 days (D8), treated with or without the lysosomal inhibitor chloroquine (CQ). HSP70 was used as a loading control. The right graph shows the relative EpoR protein levels from western blot analysis ($n = 2$ biological replicates). (E) EpoR and C1q undergo endocytosis during erythropoiesis. Immunoblot analysis of EpoR, C1qC, and HbB in whole cell lysates of HUDEP-2 cells before (D0) and after (D4) differentiation, treated with DMSO or the endocytosis inhibitor (Dynasore) at the indicated doses and times. HSP70 was used as a loading control. (F) EpoR undergoes lysosomal degradation in a C1q-dependent manner. Confocal fluorescence images of HUDEP-2 cells before (D0) and after (D4) differentiation with normal serum or C1q-depleted serum. Cells were stained with anti-EpoR antibody (EpoR, green), anti-LAMP1 antibody (LAMP1, red), and nuclei were stained with DAPI (blue). Enlarged views of the regions are indicated by the white dashed squares. White arrows in image #2 represent the colocalization of EpoR and LAMP1. Images are orthogonal projections of Z-stacks. Right graph represents quantification of colocalization of EpoR (green) with LAMP1 (red). Scale bars, 10 μm. (G) Graph represents the quantification of colocalization percentage of EpoR with LAMP1 in (F). Colocalization between EpoR and LAMP1 signals was quantified using Mander's overlap coefficient after background subtraction. Each dot represents the value from an individual cell, and horizontal bars indicate the median value ($n =$ number of analyzed cells). P values are determined by an unpaired two-tailed t-test. (H) Graph shows Pearson's correlation coefficient (PCC) values of EpoR with LAMP1 in (F). Colocalization between the two signals was quantified using PCC after background subtraction. Each dot represents the PCC value of an individual cell, and horizontal bars indicate the median value ($n =$ number of analyzed cells). P values are determined by an unpaired two-tailed t-test. (I) The protein levels of TRIM10α and HbA are unaffected by C1q-depleted conditions. Immunoblot analysis of TRIM10α, HbA, and C1qC from whole cell lysates of HUDEP-2 cells before (D0) or after (D8) differentiated with normal serum or C1q-depleted serum. HSP70 was used as a loading control. The asterisk (*) marks the non-specific bands. Data information: Unless otherwise stated, data shown are mean ± SD of three biological replicates; statistical significance was determined by unpaired two-tailed t-test. Source data are available online for this figure.

metabolism and protein translation, and it can act as a functional ligand by inducing AKT signaling in neuronal stem cells and activating p38 signaling to promote cellular migration and invasion (Benavente et al, 2020; Lee et al, 2018; Ling et al, 2018; Scott-Hewitt et al, 2024). Based on these observations, we speculate that TRIM10α-mediated capture of extracellular C1q and its delivery into erythroid cells regulate global protein synthesis by modulating RNPs and intracellular signaling pathways. Supporting this view, we observed C1q-enriched nuclear speckles in erythroblasts and cytosolic puncta in reticulocytes (Fig. 3C), suggesting diverse subcellular roles for C1q during reticulocyte maturation.

Our findings suggest that TRIM10α is essential for hemoglobin maturation and erythroid survival. TRIM10α knockdown reduced both globin dimers and tetramers, consistent with its role in facilitating hemoglobin assembly. In late-stage erythropoiesis, TRIM10α colocalized with hemoglobin aggregates and p62, and loss of its RING domain impaired this association. These results indicate that TRIM10α self-ubiquitination enables p62 recruitment and promotes the organization of aggregates for autophagic clearance. Since globin oligomers were barely ubiquitinated, TRIM10α is unlikely to act as a direct ligase for globins but instead functions as a regulatory hub linking hemoglobin aggregation to degradative pathways. By coordinating these processes, TRIM10α helps maintain hemoglobin homeostasis and supports terminal erythroid maturation. Thus, further investigation using TRIM10α knockout mice will provide key insight into its physiological relevance and help determine whether TRIM10α serves as a mechanistic link between erythroid maturation and red cell pathologies.

Inclusion bodies play a crucial role in maintaining proteostasis by degrading misfolded proteins, thereby protecting cells from proteotoxic stress (Johnston and Samant, 2021). However, inclusion bodies composed of misfolded proteins or nucleic acids can impair erythroid function and morphology, contributing to anemia,

as observed in β-thalassemia (Arlet et al, 2014; Fairbanks et al, 1969; Fertman and Doan, 1948; Gasperini et al, 1992; Hitzig et al, 1960; Sheehy, 1964). Heinz bodies, insoluble inclusions formed by aggregated hemoglobin, are characteristic of reticulocytes and serve as transient sites for the sequestration of oxidized or misfolded globins during maturation (Weiss and dos Santos, 2009). These structures recruit chaperones such as p62/SQSTM1 and HSP proteins for subsequent clearance, all of which were also found to associate with TRIM10α in our study (Figs. 3A and EV3B). Thus, inclusion bodies can be a double-edged sword, as excessively large inclusions may compromise cell viability. Our results suggest that re-activation of dormant TRIM10β expression and the formation of large inclusion bodies are prerequisites for the transition to erythroid apoptosis (Fig. 7). Because TRIM10α self-associates via its CC domain, and TRIM10β contains the same domain, TRIM10β may interact with TRIM10α, thereby accelerating the formation of harmful inclusion bodies in anemia. It is plausible that the epigenetic silencing of TRIM10β in humans evolved as a protective mechanism against this risk. Supporting this notion, TRIM10α overexpression alone, though less pronounced, also reduced cell viability (Fig. 7C). This suggests that the detrimental effect is not strictly isoform-specific but depends on relative expression levels, which may explain why TRIM10α is confined to defined stages of erythropoiesis, silenced in non-erythroid cells, and potentially pathogenic when aberrantly expressed, as in reticulocytes of sickle cell anemia patients. Furthermore, TRIM10α oligomerized under oxidative conditions (Figs. 2 and EV2) and may form TRIM10α-enriched inclusion bodies if not properly downregulated during late erythroid maturation. Consequently, molecular mechanisms may have evolved to silence or stage-specifically regulate potentially harmful paralogous genes such as *TRIM10a* and *TRIM10b*.

In summary, our findings suggest that interventions targeting TRIM10α- or TRIM10β-mediated disruptions in erythroid maturation may provide novel therapeutic avenues for anemias such as

SCA and DBA. Regulating the expression or selectively inhibiting the pathological activities of TRIM10 paralogs could help prevent the onset and progression of anemia. Furthermore, TRIM10 expression itself may hold value as a diagnostic biomarker and as a therapeutic target for mitigating disease severity.

# Methods

## Cell cultures

Human umbilical cord-derived erythroid progenitor (HUDEP-2) cells were purchased from RIKEN BioResource Research Center,

Japan (Kurita et al, 2013). HUDEP-2 cells were expanded in StemSpan serum-free expansion medium (SFEM) (#09650, Stem Cell Technologies) supplemented with 100 ng/ml recombinant human stem cell factor (rhSCF) (#255-SC, R&D Systems), 3 IU/ml recombinant human erythropoietin (rhEPO) (#78007, Stem Cell Technologies), 1 μM dexamethasone (D4902, Sigma-Aldrich), 1 μg/ml doxycycline (D9891, Sigma-Aldrich), and 1% penicillin-streptomycin (SV30010, HyClone). To induce erythroid differentiation, HUDEP-2 cells were cultured in Iscove's modified Dulbecco's medium (IMDM) (SH30259, HyClone) supplemented with 330 μg/ml human holo-transferrin (T0665, Sigma-Aldrich), 5% AB human serum (H5667, Sigma-Aldrich), 2 IU/ml heparin (H3149, Sigma-Aldrich), 10 μg/ml human insulin (I9278,

**Reagents and tools table**

| Reagent/resource | Reference or source | Identifier or catalog number |
| --- | --- | --- |
| **Experimental models** | | |
| HUDEP-2 cells (*H. sapiens*) | Kurita et al, (2013) | RIKEN BioResource Research Center, Japan |
| HEK293T cells (*H. sapiens*) | ATCC | CRL-1573 |
| C57BL/6 N (*M. musculus*) | OrientBio | The Jackson Laboratory |
| **Recombinant DNA** | | |
| pMSCVpuro | Clontech | #634401 |
| pcDNA3.1(+) | Life Technologies | V79020 |
| eSpCas9-LentiCRISPR v2 | GenScript | N/A |
| Myc-TRIM10α | pMSCV | N/A |
| Myc-TRIM10α ΔRING | pMSCV | N/A |
| Myc-TRIM10α ΔB-box | pMSCV | N/A |
| Myc-TRIM10α ΔCoiled-coil | pMSCV | N/A |
| Myc-TRIM10 ΔPRY/SPRY | pMSCV | N/A |
| GFP-TRIM10β | pMSCV | N/A |
| GFP-TRIM10β ΔRING | pMSCV | N/A |
| GFP-TRIM10 ΔPRY/SPRY | pMSCV | N/A |
| GFP-TRIM10α | pcDNA3.1(+) | N/A |
| GFP-TRIM10α ΔRING | pcDNA3.1(+) | N/A |
| GFP-TRIM10α ΔB-box | pcDNA3.1(+) | N/A |
| GFP-TRIM10α ΔCoiled-coil | pcDNA3.1(+) | N/A |
| GFP-TRIM10 ΔPRY/SPRY | pcDNA3.1(+) | N/A |
| GFP-TRIM10β | pcDNA3.1(+) | N/A |
| GFP-TRIM10β ΔRING | pcDNA3.1(+) | N/A |
| GFP-TRIM10β ΔB-box | pcDNA3.1(+) | N/A |
| GFP-TRIM10β ΔCoiled-coil | pcDNA3.1(+) | N/A |
| sgControl | eSpCas9-LentiCRISPR v2 | N/A |
| sgTRIM10 | eSpCas9-LentiCRISPR v2 | N/A |
| **Antibodies** | | |
| Mouse anti-Myc | Cell Signaling Technology | #2276 |
| Rabbit anti-Myc | Cell Signaling Technology | #2278 |
| Mouse anti-Lamin A/C | Cell Signaling Technology | #4777 |
| Rabbit anti-PARP1 | Cell Signaling Technology | #9532 |
| Rabbit anti-cleaved Caspase-3 | Cell Signaling Technology | #9661 |

| Reagent/resource | Reference or source | Identifier or catalog number |
|---|---|---|
| Rabbit anti-phospho-PI3K | Cell Signaling Technology | #4228 |
| Rabbit anti-PI3K | Cell Signaling Technology | #4257 |
| Rabbit anti-phospho-Akt (Ser473) | Cell Signaling Technology | #4060 |
| Rabbit anti-Akt | Cell Signaling Technology | #4691 |
| Rabbit anti-phospho-mTOR (Ser2448) | Cell Signaling Technology | #5536 |
| Rabbit anti-mTOR | Cell Signaling Technology | #2983 |
| Rabbit anti-Raptor | Cell Signaling Technology | #2280 |
| Rabbit anti-Rictor | Cell Signaling Technology | #2114 |
| Rabbit anti-phospho-p70 S6 kinase (Thr389) | Cell Signaling Technology | #9234 |
| Rabbit anti-p70 S6 kinase | Cell Signaling Technology | #2708 |
| Goat anti-HbA | Novous Biologicals | NB110-41083 |
| Rabbit anti-Pericentrin | Abcam | ab220784 |
| Mouse anti-p62 | Abcam | ab56416 |
| Rabbit anti-ubiquitin | Abcam | ab7780 |
| Rabbit anti-LAMP1 | Abcam | ab24170 |
| Mouse anti-GFP | Santa Cruz Biotechnology | sc-9996 |
| Mouse anti-HbA | Santa Cruz Biotechnology | sc-514378 |
| Mouse anti-Vinculin | Santa Cruz Biotechnology | sc-73614 |
| Mouse anti-C1qC | Santa Cruz Biotechnology | sc-365301 |
| Mouse anti-EpoR | Santa Cruz Biotechnology | sc-365662 |
| Mouse anti-ubiquitin | Santa Cruz Biotechnology | sc-8017 |
| Mouse anti-β-actin | Santa Cruz Biotechnology | sc-47778 |
| Mouse anti-α-tubulin | Santa Cruz Biotechnology | sc-23948 |
| Rabbit anti-TRIM10α | Thermo Fisher Scientific | PA5-31291 |
| Mouse anti-GAPDH | Thermo Fisher Scientific | MA5-15738 |
| Mouse anti-GFP | Roche | #11814460001 |
| Mouse anti-TRIM10 | Abnova | H00010107-B01P |
| Rabbit anti-HbB | Proteintech | 16216-1-AP |
| Rabbit anti-HSP70 | Proteintech | 10995-1-AP |
| Alexa Fluor 488-conjugated secondary mouse antibody | Thermo Fisher Scientific | A-11029 |
| Alexa Fluor 488-conjugated secondary rabbit antibody | Thermo Fisher Scientific | A-11034 |
| Alexa Fluor 568-conjugated secondary mouse antibody | Thermo Fisher Scientific | A-11031 |
| Alexa Fluor 568-conjugated secondary rabbit antibody | Thermo Fisher Scientific | A-11036 |
| Alexa Fluor 568-conjugated secondary goat antibody | Thermo Fisher Scientific | A-11057 |
| Alexa Fluor 647-conjugated secondary rabbit antibody | Thermo Fisher Scientific | A-21245 |
| Goat anti-mouse HRP | Thermo Fisher Scientific | #31430 |
| Goat anti-rabbit HRP | Thermo Fisher Scientific | #31460 |
| Rabbit IgG-Isotype Control-FITC | Abcam | ab37406 |
| Rabbit anti-C1q-FITC | Abcam | ab4223 |

| Reagent/resource | Reference or source | Identifier or catalog number |
|---|---|---|
| Anti-human CD235ab-PE | Biolegend | 306603 |
| Anti-human CD49d-APC | Biolegend | 304307 |
| Anti-human CD36-APC | Biolegend | 336207 |
| Mouse IgG control | Santa Cruz Biotechnology | SC-2025 |
| Rabbit IgG control | Cell Signaling Technology | 2729S |
| **Oligonucleotides and other sequence-based reagents** | **Forward** | **Reverse** |
| TRIM10a | 5' CCAGCGAGCTCAGTT CTC3' | 5' CTTCCGCTGCAC ATCCTC 3' |
| TRIM10b | 5' GGAGATGAAGATGTTTCTGGA 3' | 5' CATCCACACCCACGTGTG 3' |
| HBA1 | 5' CTCCTAAGCCACTGCCTG 3' | 5' AGAAGCCAGGAACTTGTCCA 3' |
| HBB | 5' CACGTGGATCCTGAGAACTT 3' | 5' CACTGGTGGGGTGATTCTT 3' |
| GAPDH | 5' TGATGACATCAAGAAGGTGGTGAA 3' | 5' TCCTTGGAGGCCATGTGGGCCAT 3' |
| Trim10 | 5' CTGTGATGCCAGTGTGGG 3' | 5' CACCACATTGGCCAGCTG 3' |
| Gapdh | 5' ATGGTGAAGGTCGGTGTGAAC 3' | 5' CGTTGATGGCAACAATCTCCAC 3' |
| **Chemicals, Enzymes and other reagents** | | |
| StemSpan Serum-Free Expansion Medium (SFEM) | Stem Cell Technologies | #09650 |
| Iscove's Modified Dulbecco's Medium (IMDM) | HyClone | SH30259 |
| Dulbecco's modified Eagle medium (DMEM) | HyClone | SH30243 |
| Roswell Park Memorial Institute (RPMI)-1640 medium | HyClone | SH30027 |
| Heat inactivated fetal bovine serum (FBS) | HyClone | SH30919 |
| Dulbecco's Phosphate Buffered Saline (DPBS) | Welgene | LB 001-02 |
| Recombinant Human Stem Cell Factor (rhSCF) | R&D Systems | #255-SC |
| Recombinant Human Erythropoietin (rhEPO) | Stem Cell Technologies | #78007 |
| Dexamethasone | Sigma-Aldrich | D4902 |
| Doxycycline | Sigma-Aldrich | D9891 |
| Human holo-transferrin | Sigma-Aldrich | T0665 |
| AB Human Serum | Sigma-Aldrich | H5667 |
| Heparin | Sigma-Aldrich | H3149 |
| Human insulin | Sigma-Aldrich | I9278 |
| L-glutamine | Welgene | LS 002-01 |
| Penicillin-Streptomycin | HyClone | SV30010 |
| Puromycin, Dihydrochloride | AG Scientific | P-1033 |
| Phorbol 12-myristate 13-acetate (PMA) | Sigma-Aldrich | P8139 |
| Dynasore | Sigma-Aldrich | 324410 |
| N-ethylmaleimide (NEM) | Sigma-Aldrich | #E3876 |
| Omicsfect | Omics Biotechnology | #CP2101 |
| P3 Primary Cell 4D-Nucleofector X Kit | Lonza | V4XP-3024 |
| Protein G-Sepharose | GenScript | #L00209 |

| Reagent/resource | Reference or source | Identifier or catalog number |
| --- | --- | --- |
| cOmplete Mini, EDTA-free Protease Inhibitor Cocktail | Roche | 11836170001 |
| Bio-Rad Protein Assay Dye Reagent Concentrate | Bio-Rad | #5000006 |
| PVDF membrane - Immobilon-P | Millipore | #IPVH00010 |
| WesternBright ECL kit | Advansta | K-12045-D50 |
| RNA preparation kit | Enzynomics | EP301 |
| RNeasy Mini Kit | Qiagen | #74104 |
| D-Plus™ CCK cell viability assay kit | Donginbio | CCK-3000 |
| T4 Polynucleotide kinases | Thermo Fisher Scientific | #EK0031 |
| Reverse transcriptase | Enzynomics | RT001H |
| Poly-L-lysine | Sigma-Aldrich | P4707 |
| CellTracker Red CMTPX | Thermo Fisher Scientific | C34552 |
| Hoechst 33258 | Invitrogen | H3570 |
| DAPI (4′, 6-diamidino-2-phenylindole) | Sigma-Aldrich | D9542 |
| ProLong™ Gold Antifade Mountant | Life Technologies | #P10144 |
| **Software** | | |
| GraphPad Prism 8.0 software | https://www.graphpad.com | |
| Microsoft Excel software | https://www.microsoft.com/ | |
| ImageJ | https://imagej.nih.gov/ij/index.html | |
| Design and Analysis Software | https://www.thermofisher.com/kr/en/home/life-science/pcr/real-time-pcr/real-time-pcr-instruments/quantstudio-systems/models/quantstudio-3-5 | |
| ZEN blue software. | https://www.zeiss.com/ | |
| AlphaFold | https://deepmind.google/science/alphafold/ | |
| **Other** | | |
| 4D-Nucleofector X unit | Lonza | |
| Applied Biosystems QuantStudio3 Real-Time PCR System | Thermo Fisher Scientific | |
| LSM 900 confocal laser scanning microscope | Carl Zeiss | |
| Infinite® M Plex | Tecan | |
| GeneTouch | Bioer | |
| NovaSeq 6000 System Specifications | Illumina | |

Sigma-Aldrich), 2 mM L-glutamine, 3 IU/ml rhEPO, 100 ng/ml rhSCF, 1 µg/ml doxycycline, and 1% penicillin-streptomycin. rhSCF was excluded from the differentiation media on day 6, followed by the removal of doxycycline on day 8. The human embryonic kidney cell line (HEK) 293 T was obtained from ATCC (CRL-11268) and cultured in Dulbecco's modified Eagle medium (DMEM) (SH30243, HyClone) supplemented with 10% heat inactivated fetal bovine serum (FBS) (SH30919, HyClone) and 1% penicillin-streptomycin. U937 cell line was obtained from ATCC (CRL-1593.2) and cultured in Roswell Park Memorial Institute (RPMI)-1640 medium (SH30027, HyClone) supplemented with

10% heat inactivated FBS and 1% penicillin-streptomycin. All cells were grown in a humidified incubator at 37 °C with 5% $CO_2$.

## DNA constructs and gene expression

Two isoforms of TRIM10, TRIM10α, and TRIM10β, were obtained from the cDNA library. GFP- and Myc-tagged constructs are subcloned into pMSCV (Clontech) and pcDNA3.1(+) (Life Technologies) vectors. The deletion mutants of TRIM10α/β were prepared by the overlap extension PCR method. Single-guide RNA (sgRNA) targeting human TRIM10 was generated through CRISPR

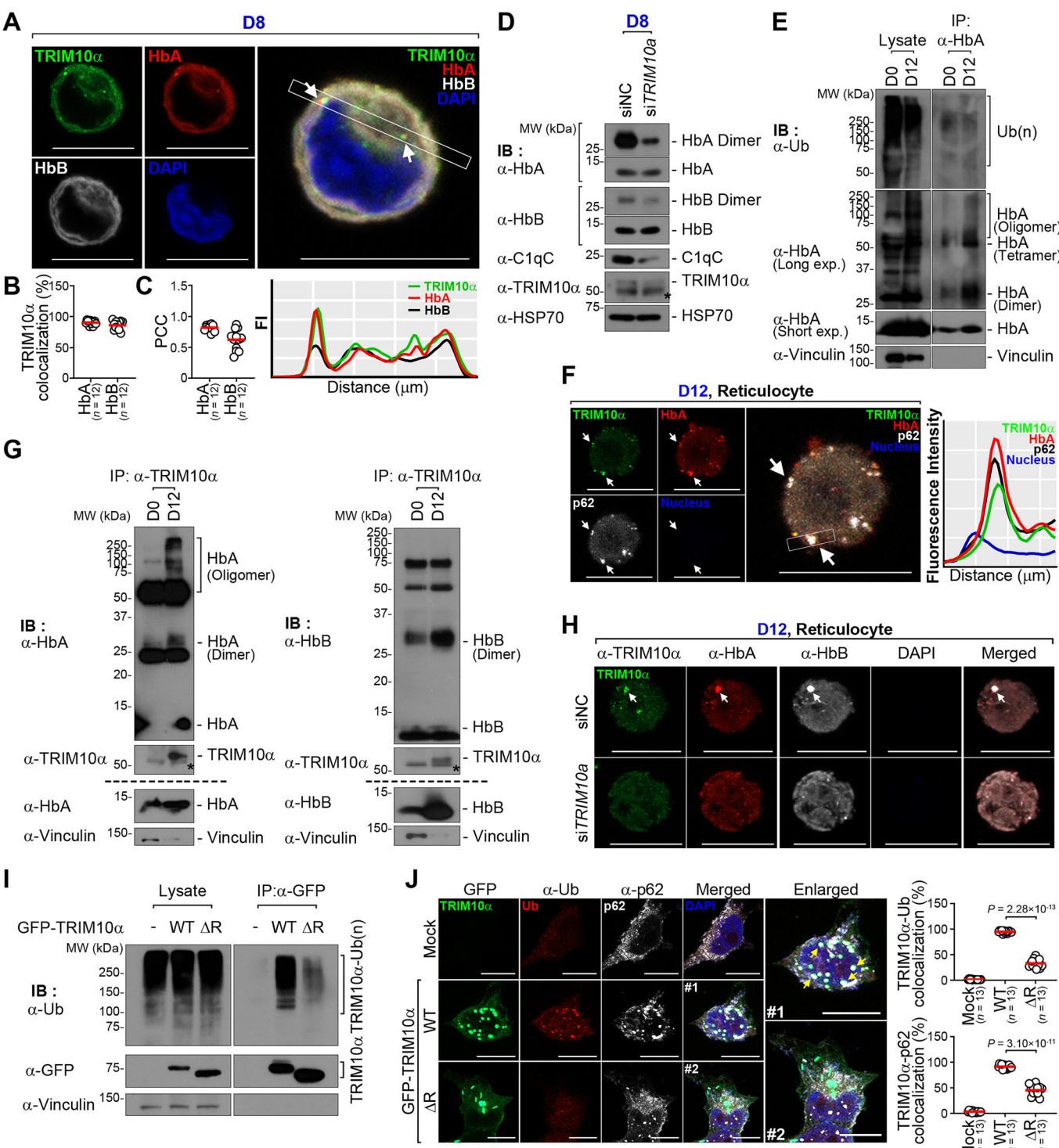

design tools and cloned into eSpCas9-LentiCRISPR v2 vector (GenScript). All coding constructs were verified by DNA sequencing. All primers and DNA constructs are listed in the Reagents & Tools Table. In HEK293T cells, transient expression of DNA constructs was performed using Omicsfect (OmicsBio) according to the manufacturer's instructions. Stable expression of TRIM10α/β was achieved by the retroviral infection method. The plasmid mixture contained a retroviral plasmid containing TRIM10α/β and

packaging plasmids (VSV-G and Gag-Pol) transfected into HEK293T cells to produce retroviral particles. Supernatants containing viral particles were harvested at 36–48 h after transfection and filtered through a 0.45 μm filter. For transduction, cells were treated with 8 μg/ml polybrene concurrently with retroviral particles, and centrifuged at 2200 rpm for 45 min. After 24 h, 2 μg/ml puromycin was added to the medium for the selection of TRIM10α/β-expressing cells.

◀  **Figure 6.   Stage-dependent dual roles of TRIM10α in Hb formation and clearance.**

(A) TRIM10α colocalizes with HbA and HbB. Confocal fluorescence images of HUDEP-2 cells after (D8) differentiation. Cells were stained with anti-TRIM10α antibody (TRIM10α, green), anti-HbA antibody (HbA, red), anti-HbB antibody (HbB, white), and the nuclei were stained with DAPI (blue). White arrows represent the colocalization of TRIM10α foci with HbA foci. Images are orthogonal projections of Z-stacks. The graph at the bottom right represents the quantification of colocalization of TRIM10α (green) with HbA (red) and HbB (Black). Scale bars, 10 μm. (B) Graphs represent the quantification of colocalization percentage of TRIM10α with HbA or HbB in (A). Colocalization between TRIM10α and each globin signals was quantified using Mander's overlap coefficient after background subtraction. Each dot represents the value from an individual cell, and horizontal bars indicate the median value ($n$ = number of analyzed cells). (C) Graphs show Pearson's correlation coefficient (PCC) values of TRIM10α with HbA or HbB in (A). Colocalization between the two signals was quantified using PCC after background subtraction. Each dot represents the PCC value of an individual cell, and horizontal bars indicate the median value ($n$ = number of analyzed cells). (D) TRIM10α facilitates Hb-heme association. Immunoblot analysis of HbA, HbB, C1qC, and TRIM10α from whole cell lysates of HUDEP-2 cells transfected with siNC or si*TRIM10α*. Cells were transfected on day 6 and differentiated for an additional 48 h before harvesting on day 8. HSP70 was used as a loading control. The asterisk (*) marks the non-specific bands. (E) HbA is barely ubiquitinated in late stages of erythropoiesis. Immunoblots showing HbA ubiquitination in HUDEP-2 cells cultured in differentiation medium on day 12. Lysates were immunoblotted with anti-Ubiquitin, anti-HbA, or anti-Vinculin. (F) Punctate structures of TRIM10α are colocalized with HbA and p62. Confocal fluorescence images of HUDEP-2 cells cultured in differentiation medium on day 12. Cells were stained with anti-TRIM10α antibody (TRIM10α, green), anti-HbA antibody (HbA, red), and anti-p62 antibody (p62, white), and nuclei were stained with DAPI (blue). White arrows represent the oligomeric TRIM10α colocalized with HbA and p62. Images are orthogonal projections of Z-stacks. Right graph represents the quantification of colocalization of TRIM10α (green) with HbA (red) and p62 (Black). DAPI, blue. Scale bars, 10 μm. (G) TRIM10α binds to Hb. Co-immunoprecipitation analysis of the interaction between TRIM10α and HbA (left)/HbB (right) in HUDEP-2 cells cultured in differentiation medium on day 12. Lysates were immunoblotted with anti-HbA, anti-HbB, anti-TRIM10α, or anti-Vinculin. Above the dashed lines are the immunoprecipitated proteins, and below the dashed lines are the whole cell lysates. The asterisk (*) marks the non-specific bands. (H) Punctate structures of Hb are absent in *TRIM10α*-depleted cells. Confocal fluorescence images of HUDEP-2 cells transfected with siNC or si*TRIM10α*. Cells were transfected on day 6 and differentiated for 72 h, followed by a second transfection on day 9 and further differentiation for 72 h until day 12. Cells were stained with anti-TRIM10α antibody (TRIM10α, green), anti-HbA antibody (HbA, red), and anti-HbB antibody (HbB, white), and nuclei were stained with DAPI (blue). White arrows represent the oligomeric TRIM10α colocalized with HbA and HbB. Images are orthogonal projections of Z-stacks. Scale bars, 10 μm. (I) TRIM10α self-ubiquitinates via its RING domain. Immunoblots showing TRIM10α ubiquitination in control vector-, GFP-tagged WT-TRIM10α-, or TRIM10α RING deletion (ΔR) mutants-expressing HEK293T cells. Vinculin was used as a loading control. (J) TRIM10α colocalized with ubiquitin and p62 in a RING-dependent manner. Confocal fluorescence images of HEK293T cells expressing control vector, GFP-tagged TRIM10α, and TRIM10α RING deletion (ΔR) mutants. Cells were stained with anti-Ub antibody (ubiquitin, red) or anti-p62 antibody (p62, white). Yellow arrows represent the oligomeric TRIM10α colocalized with ubiquitin and p62. Nuclei were stained with DAPI (blue). Images are orthogonal projections of Z-stacks. Right graphs represent the quantification of colocalization percentage of TRIM10α with Ub (top) or p62 (bottom). Colocalization of TRIM10α with each globin was quantified using Mander's overlap coefficient after background subtraction. Each dot represents the value from an individual cell, and horizontal bars indicate the median value ($n$ = number of analyzed cells). $P$ values are determined by unpaired two-tailed $t$-test. Scale bars, 10 μm. Data information: Data shown are from three independent biological replicates; statistical significance was determined by unpaired two-tailed $t$-test. Source data are available online for this figure.

## Antibodies

The following primary antibodies were used: anti-Myc (#2276), anti-Myc (#2278), anti-Lamin A/C (#4777), anti-PARP1 (#9532), anti-cleaved Caspase-3 (#9661), anti-phospho-PI3K (#4228), anti-PI3K (#4257), anti-phospho-Akt (Ser473) (#4060), anti-Akt (#4691), anti-phospho-mTOR (Ser2448) (#5536), anti-mTOR (#2983), anti-Raptor (#2280), anti-Rictor (#2114), anti-phospho-p70 S6 kinase (Thr389) (#9234), and anti-p70 S6 kinase (#2708) were obtained from Cell Signaling Technology. Anti-HbA (NB110-41083) was obtained from Novous Biologicals. Anti-Pericentrin (ab220784), anti-p62 (ab56416), anti-ubiquitin (ab7780), and anti-LAMP1 (ab24170) were obtained from Abcam. Anti-GFP (sc-9996), anti-HbA (sc-514378), anti-Vinculin (sc-73614), anti-C1qC (sc-365301), anti-EpoR (sc-365662), anti-ubiquitin (sc-8017), anti-β-actin (sc-47778), and anti-α-tubulin (sc-23948) were obtained from Santa Cruz Biotechnology. Anti-TRIM10α (PA5-31291) and anti-GAPDH (MA5-15738) were obtained from Thermo Fisher Scientific. Anti-GFP (#11814460001) was obtained from Roche. Anti-TRIM10 (H00010107-B01P) was obtained from Abnova. Anti-HbB (16216-1-AP), anti-Lamin B1 (12987-1-AP), and anti-HSP70 (10995-1-AP) were obtained from Proteintech. Antibodies and fluorescent dyes for immunofluorescence analysis and flow cytometry analysis were as follows: Alexa Fluor 488-, 568-, and 647-conjugated antibodies were obtained from Thermo Fisher Scientific. Anti-human CD235ab-PE (306603), anti-human CD49d-APC (304307), and anti-human CD36-APC (336207) were obtained from Biolegend. Anti-C1q-FITC (ab4223) and FITC-Rabbit IgG-Isotype Control (ab37406) were obtained from Abcam.

## Nucleofection of siRNA

For TRIM10α silencing, HUDEP-2 cells were nucleofected with 60 pmol of TRIM10α-specific or negative control siRNA (SN-1012) obtained from Bioneer Co. by using 4D-Nucleofector X unit (Lonza) and P3 Primary Cell 4D-Nucleofector X Kit (V4XP-3024) applying the CA-137 program. After nucleofection, cells were recovered in differentiation media for 10 min at 37 °C before transferring to a culture plate. HUDEP-2 cells differentiated for an additional 48-72 h before harvesting and analyzing gene-specific knockdown using immunoblotting.

## Immunofluorescence imaging microscopy

HUDEP-2 cells were plated on glass coverslips coated with poly-L-lysine (P4707, Sigma-Aldrich), followed by incubation at 37 °C in a humidified incubator for 10 min. Cells were fixed with 3.7% formaldehyde in PBS for 5 min at room temperature (RT) and permeabilized with 0.1% Triton X-100 in PBS for 5 min. After washing three times in PBS, the cells were blocked by 4% bovine serum albumin in PBS (PBA) for 30 min at RT. The cells were then incubated with the respective primary and secondary antibodies for 1 h, each followed by washing three times in PBS. After the final wash, the nuclei were stained with DAPI (4',6-diamidino-2-phenylindole, Sigma-Aldrich). Fluorescent slides were cover-slipped manually using Prolong® Gold (Life Technologies). Stained cells were visualized using an LSM 900 confocal laser scanning microscope (Carl Zeiss), and individual cell images were analyzed using ZEN Blue software. All confocal Z-stack images were

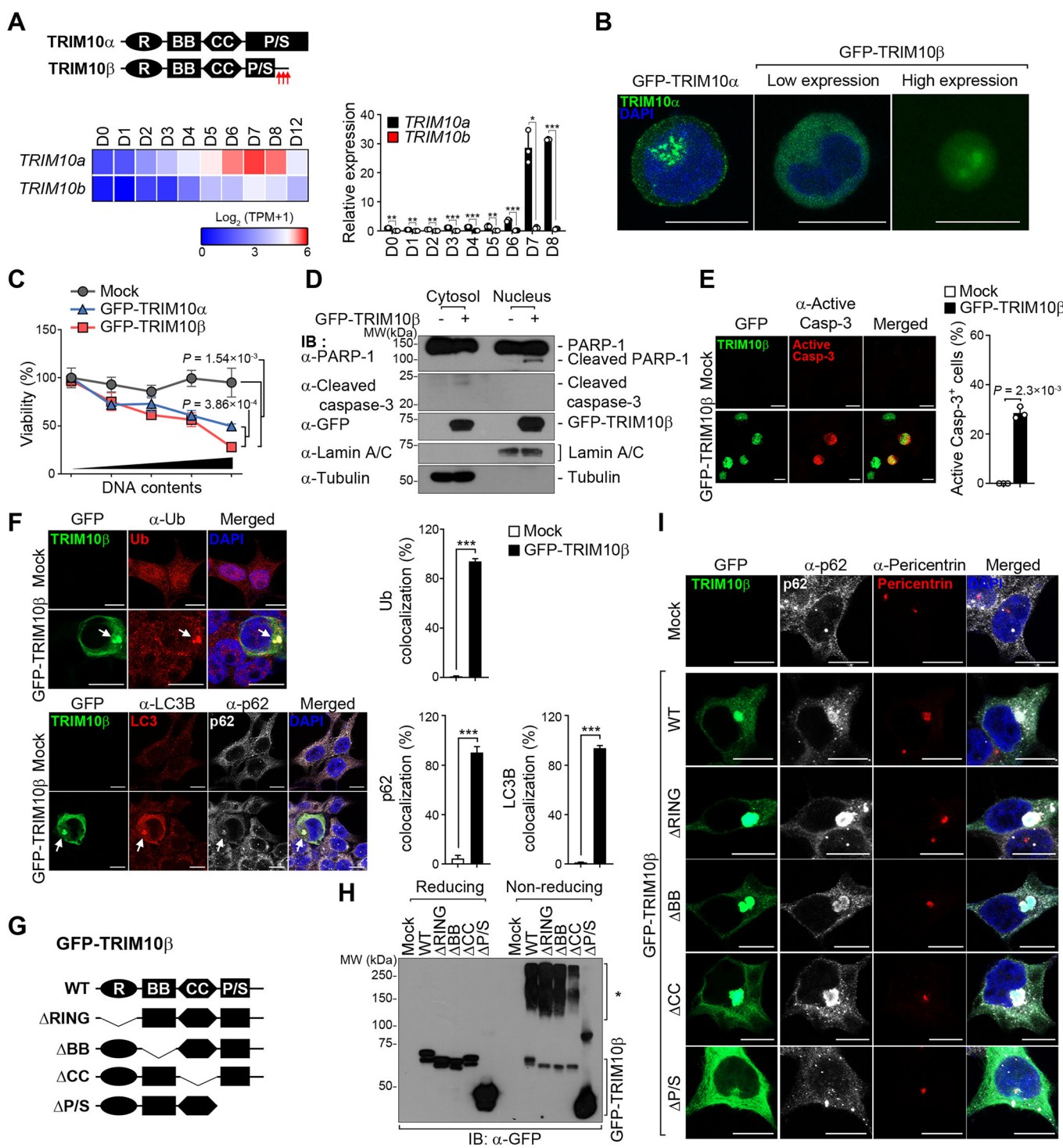

RNA-seq analysis

processed and analyzed with ZEN Blue software to quantify fluorescence intensity and determine colocalization percentages using threshold-based segmentation.

## RNA-seq analysis

The cultured HUDEP-2 cells in differentiation media were collected from day 0 to day 12 of the HUDEP-2 differentiation culture. Total

RNA was isolated from HUDEP-2 cells using the RNA preparation kit (Enzynomics) according to the manufacturer's protocol. To remove contaminated DNA, we cleaned the extracted RNAs using the RNeasy mini kit (Qiagen), and the integrity of RNA was evaluated on 1% agarose gels in Tris-Borate-EDTA (TBE) buffer. For RNA sequencing, the RNA-seq libraries for Illumina sequencing were generated using the TruSeq Stranded mRNA LT Sample Prep kit according to the manufacturer's protocols. Sequencing was

Figure 7.   Suppression of alternatively spliced *TRIM10b* expression in erythroid homeostasis.

(A) (upper) Schematic representation of the TRIM10α and TRIM10β domain structures. P/S, PRY/SPRY. (bottom, left) Heatmap showing the relative expression of two *TRIM10* isoforms by RNA-seq in HUDEP-2 cells before and after differentiation. Red and blue colors represent high and low fold change, respectively. (Bottom, right) qPCR showing relative mRNA levels of *TRIM10a* (black) and *TRIM10b* (red) in HUDEP-2 cells cultured in differentiation medium for the indicated times. Data were presented as mean ± SD ($n = 3$ biological replicates). $P$ values were determined by an unpaired two-tailed $t$-test. *$P < 0.05$; **$P < 0.01$; ***$P < 0.001$. (B) High expression of ectopic TRIM10β leads to aggregate formation. Fluorescent images of HUDEP-2 cells expressing GFP-TRIM10α or GFP-TRIM10β. Nuclei were stained with DAPI (blue). Scale bars, 10 μm. (C) TRIM10β decreases cell viability. Viability of HEK293T cells expressing a control vector (dark gray), GFP-TRIM10α (blue), or GFP-TRIM10β (red) with increased ectopic expression levels was determined using the CCK-8 assay. Data were presented as mean ± SD ($n = 3$ biological replicates). $P$ values are determined by unpaired two-tailed $t$-test. (D) Ectopic expression of TRIM10β activates apoptosis. Immunoblots showing protein levels of cleaved PARP1 and cleaved caspase-3 in cytosol and nuclear fractions from HEK293T cells expressing control vector or GFP-TRIM10β. Lamin A/C and α-Tubulin were used as the loading control. (E) TRIM10β induces cleavage of caspase-3. Confocal fluorescence images of HEK293T cells expressing control vector and GFP-tagged WT-TRIM10β. Cells were stained with anti-cleaved caspase-3 (Active Casp-3) antibody (red). Images are orthogonal projections of Z-stacks. Right graphs represent the percentage of active caspase-3⁺ cells from confocal fluorescence images. Data were presented as mean ± SD ($n = 3$ biological replicates). $P$ values are determined by unpaired two-tailed $t$-test. Scale bars, 10 μm. (F) TRIM10β aggregates colocalizes with ubiquitin, p62, and LC3. Confocal fluorescence images showing HEK293T cells expressing the control vector or GFP-TRIM10β. (Top) Cells were stained with anti-Ub antibody (Ubiquitin, red) and nuclei were stained with DAPI (blue). White arrows represent the TRIM10β aggregates colocalized with ubiquitin. (Bottom) Cells were stained with anti-LC3B antibody (LC3, red), anti-p62 antibody (p62, white), and nuclei were stained with DAPI (blue). White arrows represent the TRIM10β aggregates colocalized with LC3 and p62. Images are orthogonal projections of Z-stacks. Right graphs represent the quantification of colocalization percentage of TRIM10β with Ub (top, $n = 13$), p62 (bottom, left, $n = 13$), and LC3B (bottom, right, $n = 13$). Colocalization of TRIM10β with each marker was quantified using Mander's overlap coefficient after background subtraction. $n$ = number of analyzed cells. Data are presented as mean ± SD. $P$ values are determined by unpaired two-tailed $t$-test. ***$P < 0.001$. Scale bars, 10 μm. (G) Schematic representation of GFP-tagged full-length TRIM10β (WT) and TRIM10β mutants. ΔRING, ΔBB, ΔCC, or ΔP/S lacks RING, B-box, coiled-coil, or PRY/SPRY, respectively. (H) High-molecular weight bands of TRIM10β are absent in the ΔP/S mutant. Lysates from HEK293T cells expressing control vector, GFP-tagged WT-TRIM10β, or TRIM10β deletion mutants were separated by non-reducing or reducing SDS-PAGE, followed by immunoblotting with anti-GFP antibody. The asterisk (*) marks the high-molecular weight bands of oligomeric TRIM10β. (I) TRIM10β aggregates are absent in the ΔP/S mutant. Confocal fluorescence images of HEK293T cells expressing control vector, GFP-tagged WT-TRIM10β, and TRIM10β deletion mutants. Cells were stained with anti-p62 antibody (p62, white), anti-pericentrin antibody (pericentrin, red), and nuclei were stained with DAPI (blue). Images are orthogonal projections of Z-stacks. Scale bars, 10 μm. Data information: Unless otherwise stated, data shown are mean ± SD of three biological replicates; statistical significance was determined by unpaired two-tailed $t$-test. (*$P < 0.05$; **$P < 0.01$; ***$P < 0.001$). Source data are available online for this figure.

performed using NovaSeq 6000 System Specifications (Illumina). For analysis, raw reads were trimmed using the Trimmomatic program, aligned to the hg38 human reference genome by HISAT2. To estimate the transcript abundance and distribution of each differential samples, data were subjected to StringTie to calculate transcripts per kilobase million (TPM) values. A parallel differential expression gene (DEG) analysis was performed using edgeR. Read count data were performed on the trimmed mean of $M$-values (TMM) normalization method. Sample relatedness was evaluated through multidimensional scaling analysis using the plotMDS function in edgeR. For hierarchical clustering analysis (using Euclidean distance and complete linkage), samples and genes with similar expression levels were grouped based on the normalized expression values of each gene in at least one pairwise comparison of significant DEGs lists. RNA sequencing data have been deposited in the Gene Expression Omnibus (GEO).

## Single-cell RNA-seq analysis

Single-cell RNA-seq data set was downloaded from the GEO database (GSE222368) in h5ad format, and converted to a Seurat object using SeuratDisk software (version 0.0.0.9020). Good quality single-cell libraries were filtered based on the quality checking result from the original publication of this data set (Doty et al, 2023b; Data ref: Doty et al, 2023a). Normalization, selecting variable features, scaling, and dimension reduction were performed using Seurat (version 4.3.0) (Hao et al, 2021). Batch correction was performed with Harmony (version 0.1.0) (Korsunsky et al, 2019). Based on lineage annotation, the original publication provided, we extracted single-cells belonging to multipotent progenitors (MPP), and erythroid (Ery1, Ery2) categories to define them as erythroid lineage cells. To further annotate "erythroid lineage cells" into eight detailed differentiation

stages (CD34+, BFU-E, CFU-E, Pro-E, EB, LB, Poly, and Ortho), reference-based cell-type annotation was performed using the SingleR package (version 1.8.1) (Aran et al, 2019), with FACS-sorted bulk transcriptome profile for 8 differentiation stages (GSE107218) (Yan et al, 2018) as a reference data set.

## Structure prediction analysis

AlphaFold-based structures of TRIM10α in complex with lipid molecules were generated using the AlphaFold server (Abramson et al, 2024). To predict the dimeric structure of TRIM10α, two copies of the full-length *TRIM10a* sequence (Uniprot: Q9UDY6) were used. The prediction of membrane-bound TRIM10α was performed by including fifty molecules each of palmitic acid (PLM) and myristic acid (MYR), which were selected based on the limited range of predictable ligands available in the AlphaFold server.

## Quantitative RT-PCR (qPCR)

Total cellular RNA was extracted with the RNA prep kit, and 1 μg of total RNA was reverse transcribed with oligo-dT primers and random hexamer primers using Moloney Murine Leukemia Virus (M-MLV) reverse transcriptase (Enzynomics). The PCR reaction was generally performed with gene-specific primers, and the PCR products were detected in agarose gels containing ethidium bromide. Complementary DNA was analyzed by qPCR using Applied Biosystems QuantStudio3 Real-Time PCR System (Life Technologies). Values were normalized to the expression of glyceraldehyde 3-phosphate dehydrogenase (GAPDH) mRNA. All primer sequences used for qPCR are listed in the Reagents & tools table.

## Co-immunoprecipitation and immunoblot analysis

For co-immunoprecipitation (Co-IP), cells were washed once in ice-cold PBS and lysed in IP lysis buffer [50 mM Tris-HCl (pH 7.4), 150 mM NaCl, 1 mM EDTA, 1% Triton X-100 and protease inhibitor cocktail (Roche)] for 30 min at 4 °C. After centrifugation at 13,000 rpm for 10 min at 4 °C, cell lysates of supernatant were incubated with the indicated primary antibodies overnight at 4 °C and protein G-Sepharose beads (Sigma-Aldrich) for another 1 h. Then, the beads were washed three times with IP lysis buffer, and 1× denaturing buffer [50 mM Tris-HCl (pH 6.8), 5% β-mercaptoethanol, 2% sodium dodecylsulfate (SDS)] was added for elution by boiling. The immunoprecipitates or whole cell lysates were denatured with 5× sample buffer [50 mM Tris-HCl (pH 6.8), 5% β-mercaptoethanol, 2% sodium dodecylsulfate (SDS), 0.1% bromophenol blue, and 10% glycerol]. Protein samples were separated by SDS-PAGE and transferred to polyvinyl difluoride (PVDF) membrane (Millipore). After being blocked with 5% skim milk in PBS containing 0.1% Tween 20 (PBS-T) for 1 h at room temperature, the membrane was incubated with the indicated primary antibodies overnight at 4 °C. After being washed three times with PBS-T for 5 min, the membrane was incubated at room temperature for 1 h with horseradish peroxidase (HRP)-conjugated secondary antibodies. The bands were visualized using an enhanced chemiluminescence (ECL) detection reagent (Advansta).

## Ubiquitination assay

To analyze TRIM10 undergoing poly-ubiquitination, HEK293T cells were transfected with plasmids of a control vector, GFP-TRIM10α/β or its mutants. After the indicated time of transfection, cells were harvested and lysed in Ubiquitin lysis buffer [50 mM HEPES, 1% NP-40, 150 mM NaCl, 2.5 mM $MgCl_2$, 0.01% SDS, 0.1 mM EDTA, 0.05% sodium deoxycholate, 10 mM $N$-ethylmaleimide (NEM, Sigma-Aldrich), and protease inhibitor cocktail]. Cell lysates were immunoprecipitated with anti-GFP antibody, and immunoblotting analysis was performed using anti-Ubiquitin.

## Silver staining and mass spectrometry

After co-immunoprecipitation with anti-TRIM10α antibody in HUDEP-2 cells cultured on the indicated differentiation day, the eluted immunoprecipitates were separated by SDS-PAGE, and subsequently the gels were visualized by silver staining. The bands of interest were excised and subjected to tandem mass spectrometry identification.

## Cellular nuclear and cytoplasm fractionation

Cells were collected and resuspended in hypotonic buffer [50 mM HEPES, 7.5 mM $MgCl_2$, 50 mM KCl, 0.5 mM DTT, 0.25% NP-40, protease inhibitor cocktail] on ice. After centrifugation, the cytoplasmic lysate (supernatant) was separated from the nucleus (pellet). The pellets were washed twice with hypotonic buffer, resuspended in nuclear lysis buffer [50 mM Tris-HCl (pH 7.4), 0.05% NP-40, 100 mM NaCl, protease inhibitor cocktail], followed by sonication for five cycles of 1 s pulses with 3 s intervals at 20% amplitude on ice. Samples were then centrifuged, and the supernatant was harvested as nuclear lysate.

## Flow cytometry

For flow cytometry analysis of C1q or TRIM10α on HUDEP-2 cells, $5-10 \times 10^5$ cells were suspended in staining buffer (2% BSA in PBS). In the non-permeabilized condition, cells were washed, blocked with blocking reagent (human rabbit IgG at 1 mg/ml) for 15 min, and stained for 45 min in the dark either with fluorescein isothiocyanate (FITC)-conjugated anti-C1q antibody (Abcam) or TRIM10α primary antibodies followed by Alexa Fluor 488-conjugated secondary antibodies for 45 min. For C1q analysis, FITC-rabbit IgG-isotype control (ab37406, Abcam) was used as a control. Cells were then fixed in 1% formaldehyde in PBS. In the permeabilized condition, cells were first fixed in 1% formaldehyde in PBS and then permeabilized with 1% Triton X-100 in PBS. After blocking with blocking reagent, the cells were stained as described for the non-permeabilized condition for each analysis. After the final wash, samples were subjected to flow cytometry analysis (NovoCyte 3000 Flow Cytometry, Agilent Technologies).

## Erythroid and macrophage co-culture

For differentiation, U937 cells were grown on 60-mm dishes at a density of $2 \times 10^5$ cells per sample. Cells were then incubated with 100 ng/ml Phorbol 12-myristate 13-acetate (PMA) for 48 h and washed with PBS to remove non-adherent cells. After washing, attached cells were stained with 10 µM working CellTracker Red CMTPX (C34552, Thermo Fisher Scientific) diluted in prewarmed serum-free medium for 30 min at 37 °C. After 30 min, cells were washed two times with PBS and seeded into 35-mm glass-bottom culture dishes (SPL Life Science) for co-culture with HUDEP-2. Before co-culture, nuclei of differentiated HUDEP-2 cells were stained with 1 µg/ml Hoechst 33258 (H3570, Invitrogen) for 30 min at 37 °C. After 30 min, cells were washed three times with PBS and seeded into 35-mm glass-bottom culture dishes (SPL Life Science) for co-culture with adherent U937 cells for 5 h at 37 °C. After co-culturing, live images of cells were captured on the LSM 900 confocal laser scanning microscope with a microscope stage incubator at 37 °C in a humidified air with 5% $CO_2$.

## Mouse embryo organ collection

C57BL/6 N mice (4-8 weeks) were purchased from OrientBio. All mice were maintained in the specific pathogen-free facility of the Yonsei Laboratory Animal Research Center at Yonsei University, with the institutional guidelines for the care and use of laboratory animals in the Korean Food and Drug Administration. All animal experiments were reviewed and approved by the Institutional Animal Care and Use Committee of Yonsei University (YLARC, No. IACUC-A-202010-1158-02, IACUC-A-202111-1365-02). At E13.5, the pregnant mice were euthanized with Carbon dioxide ($CO_2$) inhalation for harvesting the mouse embryo organs.

## Cell counting kit 8 (CCK-8) assay

D-Plus™ CCK cell viability assay kit (CCK-3000) assay was performed to assess the effect of gradually ectopically expressed TRIM10α/β on cell viability. HEK293T cells were plated in a 100 µl culture medium containing $1 \times 10^3$ cells into 96-well culture plates. After 16 h, transfect cells with GFP-TRIM10β or control plasmid

for 48 h, followed by adding 10 μl of CCK-8 reagent to each well in the dark. After incubating for 2 h at 37 °C, the absorbance (OD values) at 450 nm was measured to analyze the number of viable cells.

## Statistical analysis

All experiments were repeated at least three times with consistent results, and data were presented as mean ± standard deviation (SD) and standard error of mean (SEM) as noted in the figure legends. All the above analyses were carried out using GraphPad Prism 8.0 software or Microsoft Excel software. The significance of differences between the two groups was assessed using a two-tailed unpaired $t$-test. Student's $t$-test was applied when variances were equal, whereas Welch's correction was used when unequal variances were assumed. $P$ values <0.05 were considered significant throughout. No data were excluded from the analyses.

# Data availability

RNA-seq data generated in this study have been deposited in the Gene Expression Omnibus (GEO) under accession number GSE280894 (https://www.ncbi.nlm.nih.gov/geo/query/acc.cgi?acc=GSE280894). The mass spectrometry proteomics data have been deposited to the ProteomeXchange Consortium via the jPOST partner repository under the accession number PXD057420 (https://repository.jpostdb.org/entry/JPST003451).

The source data of this paper are collected in the following database record: biostudies:S-SCDT-10_1038-S44319-025-00616-0.

# Peer review information

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

## Acknowledgements

This work was supported by the National Research Foundation of Korea (NRF) funded by the Ministry of Science, ICT, and Future Planning (RS-2025-00521704). SL was supported by NRF (RS-2023-00243823) and the National Cancer Center of Korea (NCC-2310480). HK and BJ were supported by the Brain Korea (BK21) FOUR Program, and WS by the Yonsei Interdisciplinary Program of Integrated OMICS for Biomedical Science. HJ was supported by NRF (RS-2024-00336518 and NRF-2018R1A6A1A03025607) and the Yonsei University Research Fund of 2024-22-0044.

## Author contributions

**Heesoo Kim**: Conceptualization; Data curation; Validation; Investigation; Visualization; Methodology; Writing—original draft. **Wonji Shin**: Investigation. **Dongeun Lee**: Investigation. **Byunghoon Jeon**: Resources. **Yongbo Kim**: Resources. **Donghyuk Shin**: Investigation; Visualization; Methodology; Project administration. **Hyobin Jeong**: Investigation; Visualization; Methodology; Project administration. **Jun Young Hong**: Investigation; Project administration. **Sungwook Lee**: Resources; Investigation; Project administration; Writing—review and editing. **Boyoun Park**: Conceptualization; Supervision; Funding acquisition; Investigation; Project administration; Writing—review and editing.

Source data underlying figure panels in this paper may have individual authorship assigned. Where available, figure panel/source data authorship is listed in the following database record: biostudies:S-SCDT-10_1038-S44319-025-00616-0.

## Disclosure and competing interests statement

The authors declare no competing interests.

# Expanded View Figures

**Figure EV1.  *TRIM10a* is an erythroid-specific gene with a dynamic expression pattern during erythropoiesis.**

(A) *TRIM10a* exhibits erythroid lineage-restricted expression across different cell types and tissues (BioGPS) (Wu et al, 2009b; Data ref: Wu et al, 2009a). Heatmap shows absolute mRNA expression, with red and blue representing high and low expression, respectively. (B) Violin plots show relative mRNA levels of *mTrim10* in organs during mouse embryonic development. Data were presented as mean ± SD ($n = 3$ biological replicates). *P* values are determined by unpaired two-tailed *t*-test. n.s., $P > 0.05$; *$P < 0.05$. (C) *mTrim10* expression is increased during mouse erythropoiesis. Violin plots show relative mRNA levels of *mTrim10* in bone marrow (BM)-derived colony-forming units erythroid (CFU-E) cells during ex vivo differentiation at the indicated times. Data are presented as mean ± SD ($n = 3$ biological replicates). *P* values are determined by unpaired two-tailed *t*-test. (D) *TRIM10a* expression is increased in SCA patients. Comparison of *TRIM10a* expression in public data from whole blood of normal (black) and SCA (red) patients (Raghavachari et al, 2009b; Data ref: Raghavachari et al, 2009a). Data were presented as mean ± SD (*n*, numbers of patients). *P* values are determined by unpaired two-tailed *t*-test. (E) Flow cytometry analysis of cell surface marker expression in HUDEP-2 cells cultured in differentiation medium at the indicated times. Cells were dual-stained with anti-CD36-PE or anti-CD49d-PE and anti-CD235a-APC conjugated antibodies. (F) HUDEP-2 cells undergo a reduction in size under differentiation conditions. Representative images of HUDEP-2 cells before and after differentiation (D ≥ 10) are shown. Scale bars, 10 μm. (G) The expression of *HBA1*, *HBB*, or *TRIM10a* increases during HUDEP-2 differentiation. qPCR analysis showing relative mRNA expression levels of *HBA1*, *HBB*, or *TRIM10a* in HUDEP-2 cells cultured in differentiation medium for the indicated times. Data were presented as mean ± SD ($n = 3$ biological replicates). *P* values are determined by unpaired two-tailed *t*-test. n.s., $P > 0.05$; *$P < 0.05$; **$P < 0.01$; ***$P < 0.001$. Data information: Unless otherwise stated, data shown are mean ± SD of three biological replicates; statistical significance was determined by unpaired two-tailed *t*-test. (n.s., $P > 0.05$; **$P < 0.01$; ***$P < 0.001$). Source data are available online for this figure.

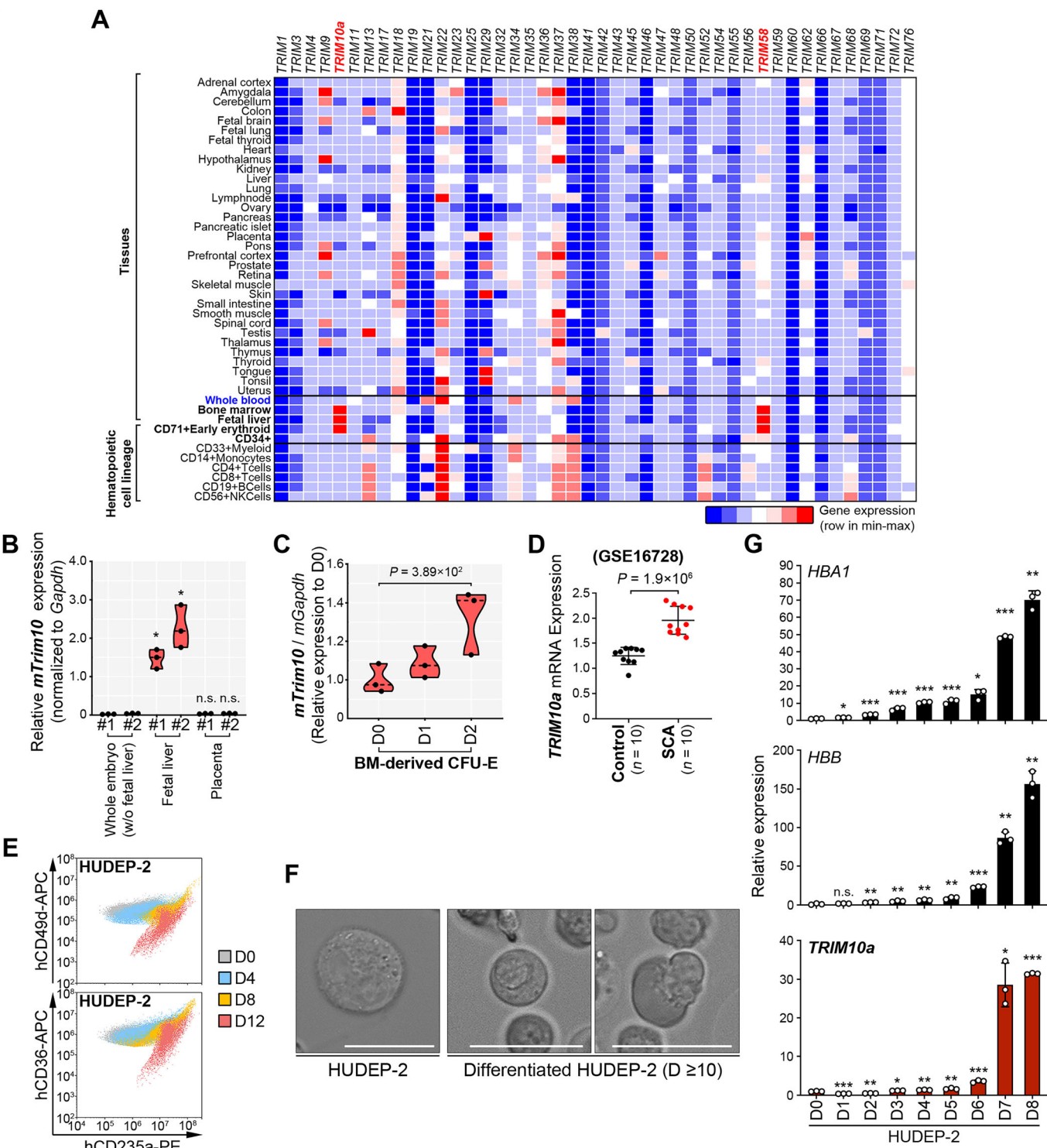

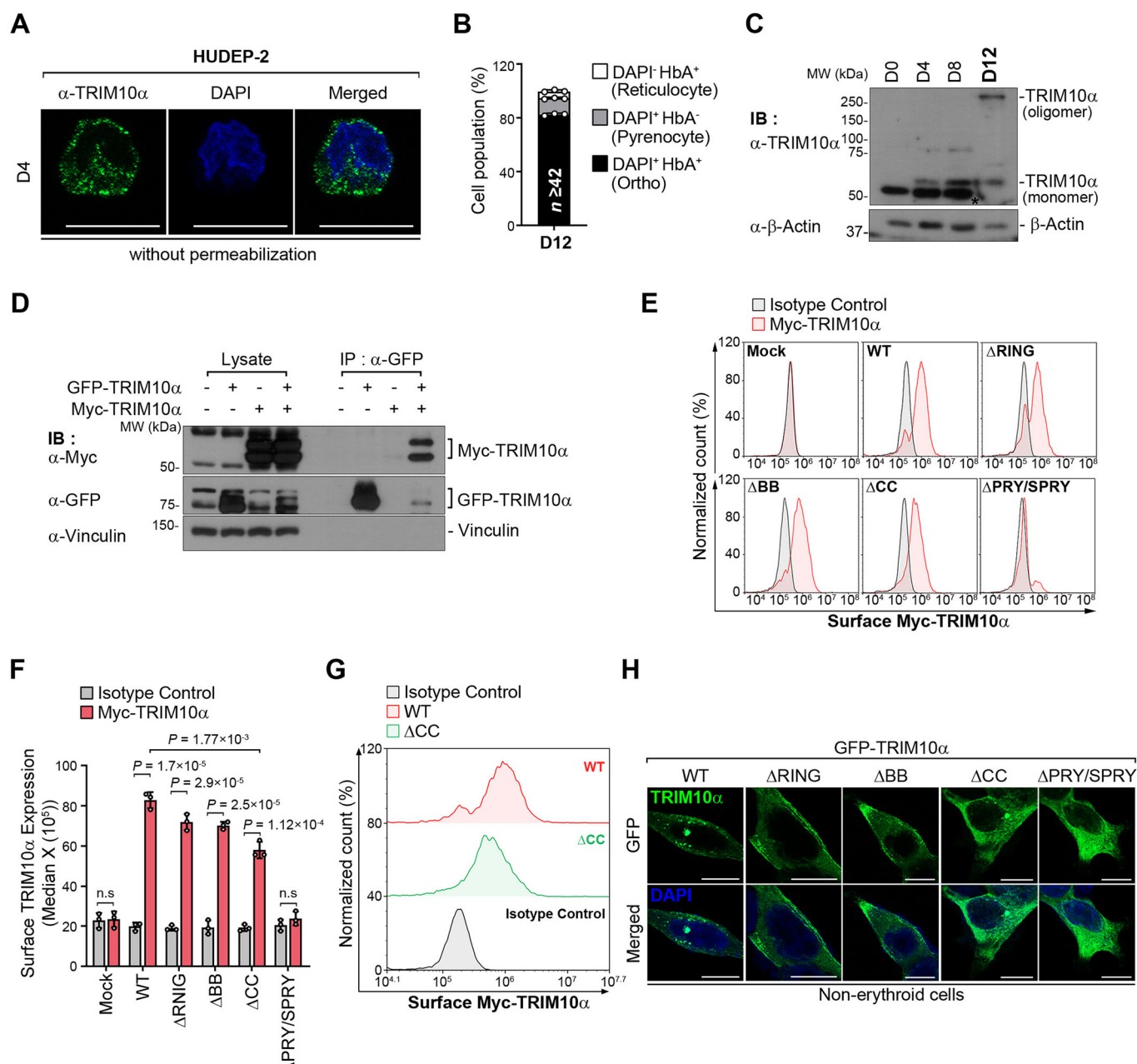

**Figure EV2.  Dynamics of protein levels and localization of TRIM10α.**

(A) TRIM10α is localized on the cell surface. Confocal fluorescence images of HUDEP-2 cells cultured in differentiation medium on day 4. Cells were stained with anti-TRIM10α antibody (TRIM10α, green), and nuclei were stained with DAPI (blue) without permeabilization. Images are orthogonal projections of Z-stacks. Scale bars, 10 μm. (B) Cell population of HUDEP-2 cells at day 12 of differentiation. Stacked bar graph shows the percentage of HUDEP-2 cell populations on day 12 of differentiation. Pyrenocyte, DAPI+ HbA- cells (dark gray); Reticulocyte, DAPI- HbA+ cells (gray); Ortho, DAPI+ HbA+ cells (white). Data were derived from the analysis of multiple cells ($n$ = number of analyzed cells). Data represent mean ± SD of three biological replicates. (C) TRIM10α forms oligomers at late stages. Immunoblots showing TRIM10α protein levels in HUDEP-2 cells cultured in differentiation medium for the indicated times. β-Actin was used as the loading control. The asterisk (*) marks the non-specific bands. (D) TRIM10α forms a dimer. Co-immunoprecipitation analysis of the interaction between GFP-TRIM10α and Myc-TRIM10α. Vinculin was used as the loading control. (E) TRIM10α is localized on the cell surface via its coiled-coil and PRY/SPRY domains. Flow cytometry analysis showing the expression of TRIM10α in HUDEP-2 cells expressing control vector, Myc-WT-TRIM10α, or TRIM10α deletion mutants cultured on expansion medium. (F) Bar graphs showing the median fluorescence intensity of TRIM10α expression in (D). Data were presented as mean ± SD ($n$ = 3 biological replicates). $P$ values are determined by unpaired two-tailed $t$-test. n.s., $P > 0.05$. (G) Coiled-coil domain is required for surface localization of TRIM10α. Flow cytometry analysis showing the expression of TRIM10α in HUDEP-2 cells expressing control vector (gray), Myc-WT-TRIM10α (red), or TRIM10α coiled-coil deletion mutants (green) in (D). (H) TRIM10α is localized on the cell surface via its coiled-coil and PRY/SPRY domains in non-erythroid cells. Confocal fluorescence images of HEK293T cells expressing control vector, GFP-WT-TRIM10α, or TRIM10α deletion mutants. Nuclei were stained with DAPI (blue). Images are orthogonal projections of Z-stacks. Scale bars, 10 μm. Data information: Unless otherwise stated, data shown are mean ± SD of three biological replicates; statistical significance was determined by unpaired two-tailed $t$-test. (n.s., $P > 0.05$). Source data are available online for this figure.

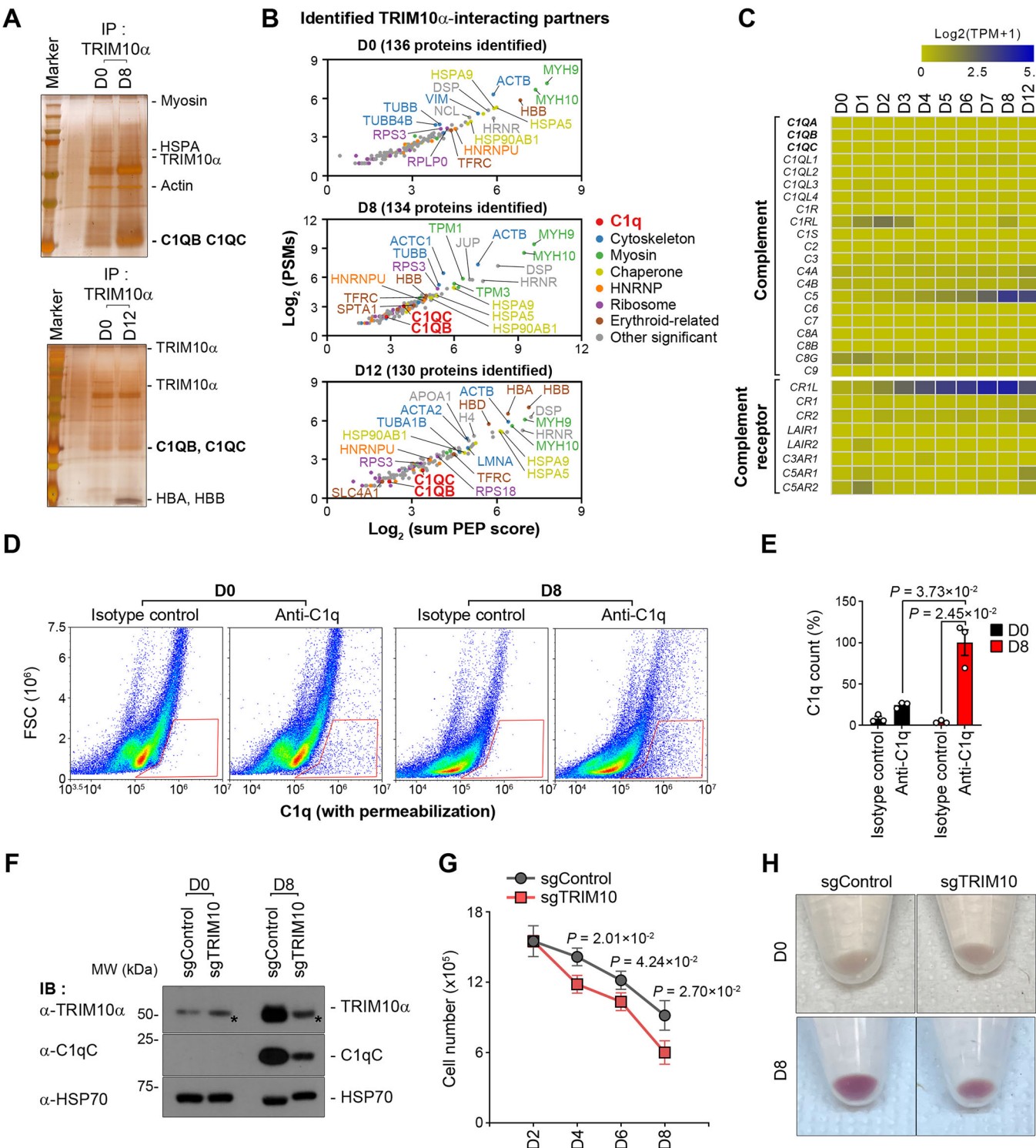

**Figure EV3. Mass spectrometry analysis of TRIM10α during erythropoiesis.**

(A) Silver staining and mass spectrometry analysis of TRIM10α-binding proteins purified from HUDEP-2 cells cultured in differentiation medium on day 8 (Top, D8) or day 12 (Bottom, D12). (B) Proteins identified by mass spectrometry of TRIM10α-binding partners. Graphs show TRIM10α-interacting proteins identified by mass spectrometry from HUDEP-2 cells cultured in differentiation medium at the indicated times across two biological replicates. A total of 136 proteins at D0, 134 proteins at D8, and 130 proteins at D12 were identified. (C) Heatmap showing the relative expression of complements or complement receptors by RNA-seq in HUDEP-2 cells before and after differentiation. Blue and yellow colors represent high and low fold change, respectively. (D) C1q is detected in differentiated cells. Flow cytometry analysis of C1q in HUDEP-2 cells before (D0) and after (D8) differentiation under permeabilization conditions. Plots show the number of intracellular C1q$^+$ cells. (E) Graphs showing the percentage of counted C1q$^+$ cell numbers in (D). Data were presented as mean ± SEM ($n = 3$ biological replicates). $P$ values are determined by unpaired two-tailed $t$-test. (F) Reduction of TRIM10α expression leads to decreased C1qC. Immunoblots showing C1qC and TRIM10α protein levels in HUDEP-2 cells expressing either a control vector or sgTRIM10 cultured in differentiation medium for the indicated times. HSP70 was used as the loading control. The asterisk (*) marks the non-specific bands. (G) Cell number decreases during differentiation under TRIM10-depleted conditions. Graph shows the number of HUDEP-2 cells expressing either a control vector (dark gray) or sgTRIM10 (red) cultured in differentiation medium for the indicated times. Data were presented as mean ± SD ($n = 3$ biological replicates). $P$ values are determined by unpaired two-tailed $t$-test. (H) Cell redness decreases during differentiation under TRIM10-depleted conditions. Images show the cell pellet redness of HUDEP-2 cells expressing either a control vector or sgTRIM10 cultured in differentiation medium for the indicated times. Data information: Unless otherwise stated, data shown are mean ± SD of three biological replicates; statistical significance was determined by unpaired two-tailed $t$-test. Source data are available online for this figure.

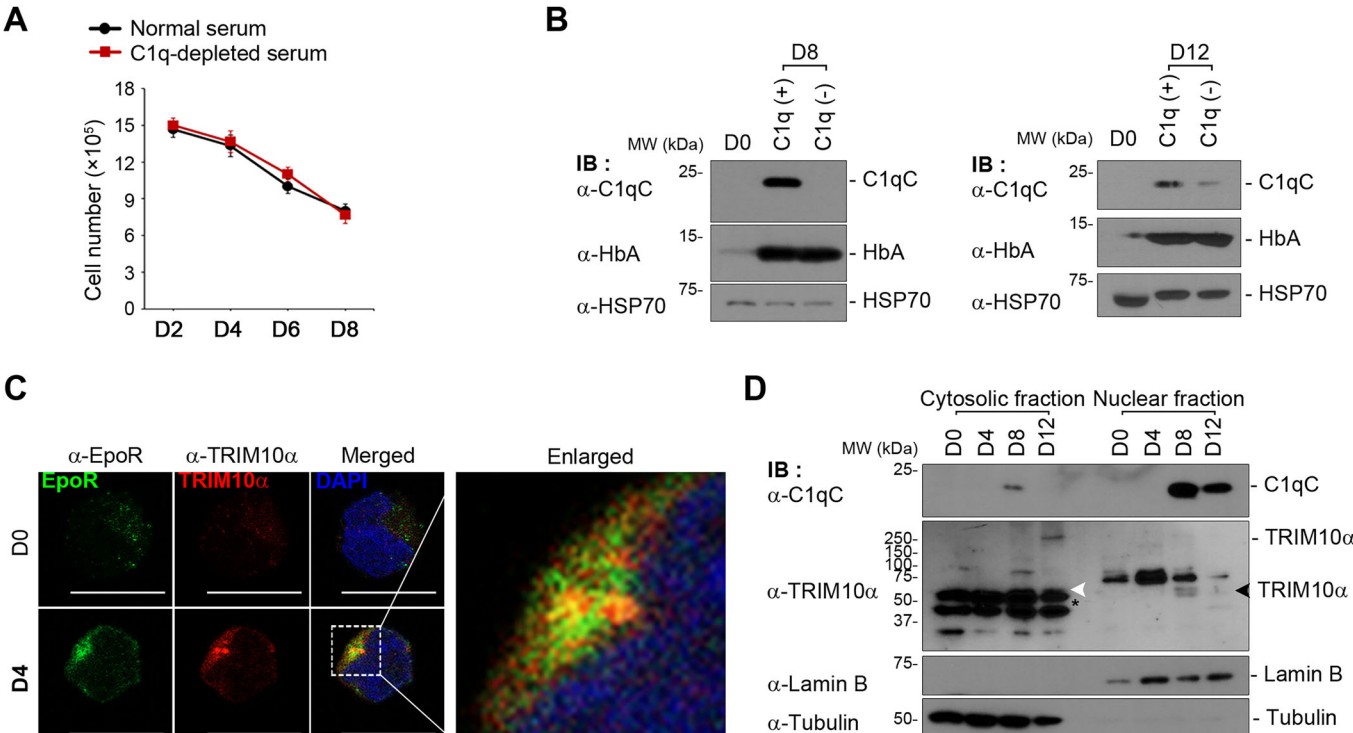

**Figure EV4. C1q and TRIM10α are present in both the cytosolic and nuclear fractions.**

(A) C1q depletion does not affect cell viability or hemoglobin synthesis. Cell numbers under normal or C1q-depleted conditions. Graph shows the number of HUDEP-2 cells counted after differentiation with normal serum (black) or C1q-depleted serum (red) at the indicated times. Data are presented as mean ± SD (n = 3 biological replicates). (B) Hemoglobin synthesis under normal or C1q-depleted conditions. Immunoblot analysis of C1qC and HbA from whole cell lysates of HUDEP-2 cells before (D0) or after differentiation (D8 or D12) with normal serum or C1q-depleted serum. HSP70 was used as a loading control. (C) EpoR colocalizes with TRIM10α. Confocal fluorescence images of HUDEP-2 cells before (D0) and after (D4) differentiation. Cells were stained with anti-EpoR antibody (EpoR, green), anti-TRIM10α antibody (TRIM10α, red), and nuclei were stained with DAPI (blue). Enlarged views of the regions are indicated by white dashed squares. Images are orthogonal projections of Z-stacks. Scale bars, 10 μm. (D) Immunoblots showing C1qC and TRIM10α in the cytosolic and nuclear fraction of HUDEP-2 cells cultured in differentiation medium at the indicated times. α-Tubulin was used as the loading control. Arrowheads represent the monomeric TRIM10α. The asterisk (*) marks the non-specific bands. Data information: Data shown are from three biological replicates. Source data are available online for this figure.

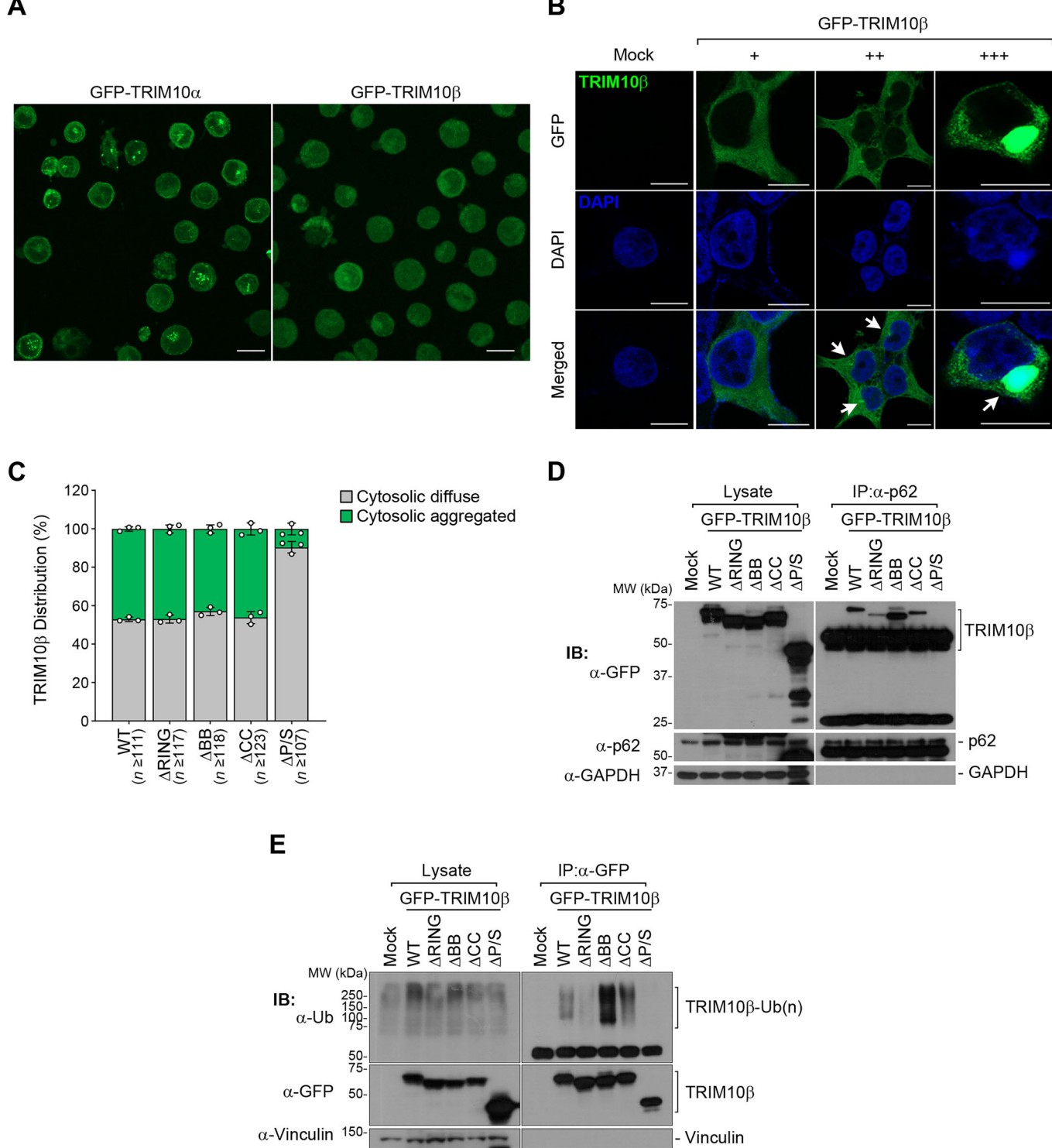

◀  **Figure EV5.   TRIM10β forms proteotoxic aggregates.**

(**A**) Wide-field fluorescent images of HUDEP-2 cells expressing GFP-TRIM10α or GFP-TRIM10β, showing multiple cells within the same field of view. Nuclei were stained with DAPI (blue). Scale bars, 10 μm. (**B**) The size of aggregates increases with higher levels of ectopic TRIM10β expression. Confocal fluorescence images of HEK293T cells expressing control vector or GFP-TRIM10β. The number of plus signs indicates an increase in ectopic GFP-TRIM10β expression levels. Nuclei were stained with DAPI (blue). White arrows represent the TRIM10β aggregates. Images are orthogonal projections of Z-stacks. Scale bars, 10 μm. (**C**) Distribution of TRIM10β in confocal microscopy images from Fig. 7I. Cells were classified as cytosolic diffuse (gray) or cytosolic aggregated (green). Data were derived from the analysis of multiple cells ($n =$ number of analyzed cells). Data represent mean ± SD of three biological replicates. (**D**) TRIM10β PRY/SPRY domain is required for binding to p62. Immunoblots showing the interaction between p62 and TRIM10β in HEK293T cells expressing control vector, GFP-WT-TRIM10β, or GFP-TRIM10β deletion mutants. GAPDH was used as a loading control. (**E**) TRIM10β self-ubiquitinates through its RING domain. Immunoblots showing TRIM10β ubiquitination in HEK293T cells expressing control vector, GFP-WT-TRIM10β, or GFP-TRIM10β deletion mutants. Vinculin was used as a loading control. Data information: Data shown are mean ± SD of three biological replicates. Source data are available online for this figure.

     