## [Peer Review File · EMBO Reports]

Stage-Specific TRIM10 Expression Regulates Erythroid Maturation

Boyoun Park, Hee soo Kim, dong hyuk Shin, Hyo bin Jeong, Junyoung Hong, Sungwook Lee, Wonji Shin, Byunghoon Jeon, Dongeun Lee, and Yongbo Kim

Corresponding author(s): Boyoun Park (bypark@yonsei.ac.kr) , Sungwook Lee (swlee1905@ncc.re.kr)

Review Timeline:

Submission Date:	24th Feb 25
Editorial Decision:	3rd Apr 25
Revision Received:	25th Aug 25
Editorial Decision:	29th Sep 25
Revision Received:	13th Oct 25
Accepted:	15th Oct 25

Editor: Achim Breiling

Transaction Report:

Dear Prof. Park,

Thank you for the submission of your manuscript to EMBO reports. I have now received the reports from the three referees that were asked to evaluate your study, which can be found at the end of this email.

As you will see, the referees think that these findings are of interest. However, they have several comments, concerns, and suggestions, indicating that a major revision of the manuscript is necessary to allow publication of the study in EMBO reports. As the reports are below, and all the referee concerns need to be addressed, I will not detail them here.

Given the constructive referee comments, I would like to invite you to revise your manuscript with the understanding that the concerns of the referees must be addressed in the revised manuscript and in a detailed point-by-point response. Acceptance of your manuscript will depend on a positive outcome of a second round of review. It is EMBO reports policy to allow a single round of revision only and acceptance of the manuscript will therefore depend on the completeness of your responses included in the next, final version of the manuscript.

- 1) a .docx formatted version of the final manuscript text (including legends for main figures, EV figures and tables), but without the figures included. Figure legends should be compiled at the end of the manuscript text.
- 2) individual production quality figure files as .eps, .tif, .jpg (one file per figure), of main figures and EV figures. Please upload these as separate, individual files upon re-submission.

- 4) a complete author checklist, which you can download from our author guidelines (<https://www.embopress.org/page/journal/14693178/authorguide>). Please insert page numbers in the checklist to indicate where the requested information can be found in the manuscript. The completed author checklist will also be part of the RPF.

- 5) that primary datasets produced in this study (e.g. RNA-seq, ChIP-seq, structural and array data) are deposited in an

appropriate public database. If no primary datasets have been deposited, please also state this in a dedicated section (e.g. 'No primary datasets have been generated and deposited'), see below.

The accession numbers and database should be listed in a formal "Data Availability" section that follows the model below. This is now mandatory (like the COI statement). Please note that the Data Availability Section is restricted to new primary data that are part of this study. This section is mandatory. As indicated above, if no primary datasets have been deposited, please state this in this section

Data availability

8) Regarding data quantification and statistics, please make sure that the number "n" for how many independent experiments were performed, their nature (biological versus technical replicates), the bars and error bars (e.g. SEM, SD) and the test used to calculate p-values is indicated in the respective figure legends (also for EV and Appendix figures). Please also check that all the p-values are explained in the legend, and that these fit to those shown in the figure. Please provide statistical testing where applicable. Please avoid the phrase 'independent experiment', but clearly state if these were biological or technical replicates. Please also indicate (e.g. with n.s.) if testing was performed, but the differences are not significant. In case n=2, please show the data as separate datapoints without error bars and statistics. See also: <http://www.embopress.org/page/journal/14693178/authorguide#statisticalanalysis>

9) Please add scale bars of similar style and thickness to microscopic images, using clearly visible black or white bars (depending on the background). Please place these in the lower right corner of the images themselves. Please do not write on or near the bars in the image but define the size in the respective figure legend.

10) Please also note our reference format:

12) We now use CRedit to specify the contributions of each author in the journal submission system. CRedit replaces the author contribution section. Please use the free text box to provide more detailed descriptions and do NOT provide your final manuscript text file with an author contributions section. See also our guide to authors: <https://www.embopress.org/page/journal/14693178/authorguide#authorshipguidelines>

13) All Materials and Methods need to be described in the main text using our 'Structured Methods' format, which is required for

all research articles. According to this format, the Methods section should include a Reagents and Tools Table (listing key reagents, experimental models, software, and relevant equipment and including their sources and relevant identifiers), uploaded as separate file, and a Methods section in which we encourage the authors to describe their methods using a step-by-step protocol format with bullet points, to facilitate the adoption of the methodologies across labs. More information on how to adhere to this format as well as downloadable templates (.doc) for the Reagents and Tools Table can be found in our author guidelines (section 'Structured Methods'):

14) Please add up to 5 keywords to the manuscript and order the sections like this, using (only) these names: Title page - Abstract - Keywords - Introduction - Results - Discussion - Methods - Data availability section - Acknowledgements (including the funding information) - Disclosure and Competing Interests Statement - References - Figure legends - Expanded View Figure legends

15) Please make sure that all the funding information is also entered into the online submission system and that it is complete and similar to the one in the acknowledgement section of the manuscript text file.

16) Please provide all the methods information in the main manuscript text files. We do not allow supplementary methods.

Please note that corresponding authors are required to supply an ORCID ID upon submission of a revised manuscript and an institutional e-mail address. Please make sure that all co-corresponding authors provide an ORCID and an institutional e-mail address in the submission system. Please find instructions on how to link the ORCID ID to the account in our manuscript tracking system in our Author guidelines: <http://www.embopress.org/page/journal/14693178/authorguide#authorshipguidelines>

I look forward to seeing a revised form of your manuscript when it is ready.

Yours sincerely,

Referee #1:

This study examines the role of TRIM10 α in erythroid maturation through stage-specific expression and subcellular localization patterns. The authors demonstrate that TRIM10 α exhibits dynamic expression during erythropoiesis, initially localizing to the cell surface in early erythroblasts where it captures extracellular C1q to facilitate pyrenocyte clearance by macrophages. In later stages, TRIM10 α partially relocates to cytosolic structures, where it reportedly enhances hemoglobin formation and sequesters harmful hemoglobin aggregates under oxidative conditions. The authors further suggest that surface C1q-TRIM10 α complexes promote EpoR lysosomal degradation, thereby attenuating protein synthesis during late erythroid maturation. Additionally, they identify an alternatively spliced variant, TRIM10 β , which forms deleterious aggregates when expressed. The study proposes that aberrant TRIM10 expression may contribute to erythroid-related diseases like sickle cell anemia and Diamond-Blackfan anemia.

While this manuscript presents an interesting hypothesis regarding TRIM10 α 's role in erythroid maturation, the data presented is insufficient to support the authors' mechanistic claims. The study relies heavily on descriptive observations from representative images without adequate quantification and statistical analysis. Additional experiments with quantitative analysis are required to support the proposed model.

Major Concerns ;

1. The manuscript relies heavily on immunofluorescence images and immunoblots, often using single representative cells to draw conclusions. While the images are informative, quantitative analysis is essential for scientific conclusions. This reviewer

recommends quantifying colocalization coefficients for all immunofluorescence experiments, providing multiple fields of view and statistical analysis of observed phenotypes, and presenting data on the percentage of cells showing the described localization patterns.

2. The HUDEP-2 differentiation system is central to this study, but its characterization is insufficient. Since HUDEP-2 cultures likely contain heterogeneous cell populations at each time point, please provide flow cytometry data showing the proportion of cells at each differentiation stage for each time point, including quantification of enucleation efficiency in late-stage cultures.
3. Many experiments rely on overexpression experiments, which can lead to artifacts, particularly regarding protein aggregation. To strengthen the physiological relevance of the authors' findings, validate key observations using endogenous TRIM10 α , compare expression levels between overexpressed and endogenous TRIM10 α , and consider using CRISPR/Cas9-mediated tagging of endogenous TRIM10 α for localization studies.
4. The flow cytometry data in Figure 2B is presented without adequate explanation of how it supports the conclusion that "TRIM10 α is localized on the cell surface." Please clarify the gating strategy and explain how surface versus intracellular staining was distinguished, include appropriate controls, and provide quantification across multiple experiments.
5. The proteomics data visualization in Figures 3A, 5A, and 6A lacks clarity, making it difficult to evaluate the interactions claimed. Please improve the color coding to make the interaction categories clearly distinguishable, provide quantitative information on interaction strengths, and include statistical analysis of the reproducibility of observed interactions.
6. The claim that C1q promotes lysosomal degradation of EpoR is based primarily on the colocalization of EpoR with LAMP1 in one representative image. This evidence is insufficient for such a significant mechanistic claim. This reviewer recommends performing endocytosis inhibition experiments to determine if C1q-induced EpoR decrease can be rescued, conducting lysosomal inhibition studies, quantifying EpoR protein turnover rates under normal and C1q-depleted conditions, and providing quantitative colocalization analysis across multiple cells.
7. The manuscript claims that TRIM10 α promotes the formation of HbA and HbB dimers, but several critical aspects are unclear. Please specify whether these are HbA/HbB heterodimers or HbA/HbA and HbB/HbB homodimers, explain the physiological significance of these dimers in erythroid maturation, assess whether normal hemoglobin tetramers are affected by TRIM10 α knockdown, and if claiming a role in hemoglobin quality control, evaluate the impact on functional tetrameric hemoglobin.
8. There appears to be a contradiction in the data regarding TRIM10 α 's role in hemoglobin aggregate clearance. If TRIM10 α contributes to aggregate clearance, one would expect increased aggregate accumulation in TRIM10 α knockdown cells, yet the opposite is observed. Please address this apparent contradiction and provide evidence for TRIM10 α 's role in clearance, consider alternative interpretations, such as TRIM10 α promoting aggregate formation rather than clearance.
9. The relationship between TRIM10 α self-ubiquitination and autophagy-mediated degradation needs further clarification. Explain the causal relationship between these processes, determine whether the Δ RING mutant undergoes autophagy-mediated degradation, investigate whether TRIM10 α , p62, and HbA colocalize in the same structures, and perform functional experiments to verify that autophagy is indeed responsible for degradation.
10. The analysis of TRIM10 β function lacks appropriate TRIM10 α controls at comparable expression levels. Please include TRIM10 α expressed at levels comparable to TRIM10 β in all Figure 7C-I experiments, clarify whether the TRIM10 α expression levels in previous figures are comparable to TRIM10 β levels shown in Figure 7, and consider titrating expression levels to determine if the observed differences are expression-level dependent or isoform-specific.

Referee #2:

In this study, Heesoo Kim et al. have identified the role of TRIM10, an erythroid-and stage-specific E3 ligase, in erythroid maturation. By analyzing the protein interactome via IP-mass spectrometry at different stages of erythroid maturation in the HUDEP-2 model, they uncovered several downstream targets of TRIM10, both in its extracellular and cytosolic forms. Specifically, they observed that TRIM10 self-associates on erythroblast surfaces and binds extracellular complement C1q, facilitating pyrenocyte encapsulation and macrophage recognition. Furthermore, they reported that surface C1q interacts with EpoR, promoting lysosomal degradation, while TRIM10 depletion prolongs Epo signaling. Additionally, they demonstrated that cytosolic TRIM10 enhances hemoglobin (Hb) maturation and sequesters Hb aggregates under oxidative conditions.

Although these findings are intriguing, the study lacks direct genetic evidence and relies solely on co-localization data between TRIM10 and its regulatory targets. Please see my major comments below:

1. Cellular Differentiation and Maturation Phenotype:

- What is the cellular differentiation and maturation phenotype following TRIM10 loss? Was enucleation severely disrupted after TRIM10 depletion? Given that the authors have generated a TRIM10-depleted HUDEP-2 cell line (Figure 3G using siRNAs), it is surprising that this phenotype was not included. It would be straightforward to knockout TRIM10 using the CRISPR/Cas9 system.

2. Primary Hematopoietic Stem or Progenitor Cells:

- More convincingly, what is the cellular phenotype when TRIM10 is depleted in mouse or human primary hematopoietic stem or progenitor cells? This would be a standard system to test erythroid differentiation or even enucleation. In Figure 1D, the authors showed TRIM10 expression data in DBA or healthy donors. It is necessary to determine whether TRIM10 is functionally relevant, rather than merely correlated.

3. Mutant Forms of TRIM10:

- The authors presented several TRIM10 mutant forms in Figure 2. Do these mutants exhibit normal biological function? In other words, if they have a TRIM10-depleted HUDEP-2 cell line, they could demonstrate cellular function by reintroducing these mutant forms and comparing them to the wild-type form.

4. Direct Ubiquitinated Targets of TRIM10:

- In Figure 7, the authors reported that TRIM10 self-ubiquitination and its binding to p62 lead to TRIM10 degradation, which promotes the removal of Hb aggregates via autophagy. This is an interesting finding. However, as an E3 ubiquitin ligase, what are the direct ubiquitinated targets of TRIM10? Is Hb aggregate a candidate? Does TRIM10a directly ubiquitinate HBA and degrade HBA dimers? In this regard, the authors have not fully explored the protein targets of TRIM10. Given that they used IP-mass spectrometry for the protein interactome, they could ideally compare the interactome between wild-type TRIM10 and a TRIM10 enzymatic mutant form. Alternatively, they could profile the total proteome after TRIM10 depletion and analyze potential substrates that might accumulate in the absence of TRIM10 (see the E3 ligase example in the reference method PMID:PMC7820877).

5. Mechanism of C1q Entry Mediated by TRIM10a:

- The authors propose that C1q entry into cells is mediated by TRIM10a, but TRIM10a expression in HUDEP2-D0 cells and C1q synthesis do not occur simultaneously. Therefore, why do undifferentiated HUDEP2 cells not absorb C1q? The authors' research found that TRIM10a suppression prevents C1q from entering cells. So, how does TRIM10a mediate the entry of C1q into cells? Do the proteins interacting with TRIM10a bind to any channel proteins? As an intermediate, does it facilitate C1q entry into the cytoplasm?

6. Inclusion of Proteome Dataset:

- It would be more comprehensive to include the proteome dataset in Figure 3. A total interactome network or Gene Ontology (GO) analysis is strongly recommended for TRIM10 binding partners in HUDEP-2 cells from days 0, 8, and 12, corresponding to the Pro-E, Poly, and Ortho/reticulocyte stages. Previous proteome studies could be referenced for comparison and discussion (e.g., HUDEP-2 PMID:PMC9718529 and primary CD34+ proteome study PMID:PMC7706838).

Referee #3:

The functions of tripartite motif (TRIM) proteins, the largest superfamily of RING-type E3 ligases, in erythroid cells remain poorly understood. Here, Heesoo Kim et al. found that TRIM10a is an erythroid- and stage-specific E3 ligase that plays a crucial role in stepwise erythroid maturation.

First, using published erythroid transcriptome data, clinical datasets, and in-house bulk RNA sequencing data, they found that aberrant TRIM10a expression is closely related to anemia. Then, they explored TRIM10a expression changes via immunofluorescence staining during human terminal erythroid differentiation and found that the localization of TRIM10a includes the erythroblast surface and cytoplasm. TRIM10a self-associates to localize on erythroblast surfaces and interacts with the extracellular complement C1q protein, which promotes pyrenocyte encapsulation.

Furthermore, the authors found that C1q interacts with EpoR and promotes its lysosomal degradation, while C1q depletion prolongs Epo signaling. In addition, cytosolic TRIM10a enhances hemoglobin maturation and sequesters its aggregates via p62-mediated autophagy under oxidative conditions. Finally, they also showed that TRIM10b, an alternatively spliced variant of TRIM10a, is barely expressed in human cells and suppresses erythropoiesis by forming deleterious aggregates. The authors concluded that aberrant TRIM10 could drive erythroid-related disorders and serve as a potential biomarker or therapeutic target.

The findings in this manuscript are not entirely convincing, as many key results are derived from subjective assays. The following concerns should be addressed to strengthen the conclusions.

Major points:

- 1, Given that the TRIM10a antibody can recognize several non-specific bands shown in Figure 3B, 3G and 5F, making the results of IP-Mass Spectrometry using this antibody unconvincing, and that ectopic GFP-TRIM10a shows similar colocalization with endogenous TRIM10a, could the authors perform IP-Mass Spectrometry with ectopic GFP-TRIM10a to strengthen the results?
- 2, Figure 4B and 4C are too subjective to be convincing. The authors should assess erythroid differentiation, enucleation, and pyrenocyte percentage changes using flow cytometry after co-culture, rather than relying solely on counting attached erythroid cells based on selectively chosen representative images.
- 3, In Figure 3D, could the authors quantify the immunofluorescence signals? Since both TRIM10a and C1qC signals consist of two forms-aggregates (bright spots) and smears (faint smears)-it is difficult to compare signal differences across different time points. Additionally, the colocalization between TRIM10a and C1qC does not appear to be significant. Could the authors provide comments on this observation?
- 4, Could the authors provide some in vivo results? What are the phenotypes of TRIM10a knockout mice and C1q knockout mice? Are there any defects in erythropoiesis in these mice?
- 5, The statement 'the TRIM10a Δ CC and TRIM10a Δ PRY/SPRY mutants mainly localized to the cytosol, losing their ability to target the plasma membrane' is not supported by the IF staining in Figure 2E, as these mutants also localize to the plasma

membrane, especially the TRIM10a Δ CC mutant. It would be better to perform flow cytometry without cell permeabilization to further confirm it.

6, Also, in Figure 2E, the nucleus size in the WT group appears smaller than in the other groups. Does this indicate that TRIM10a mutants affect nuclear condensation and terminal erythroid differentiation? If so, please quantify this figure and provide additional assays, such as flow cytometry, to further confirm the findings.

7, In Figure 3B, could the authors also detect C1qA and C1qB, since the secreted C1Q protein is a complex of C1qA, C1qB and C1qC in a constant ratio?

8, In Figure 3F and Supplemental Figure 3D, it is not appropriate to compare cell counts, as gated cell counts are positively correlated with the total recorded cell counts. Percentages would be a better measure to show the differences.

9, In Figure 5C, is HSP70 the loading control? If so, these HSP70 blots should be repeated, as the loading is unequal and the molecular weights of HSP70 vary between the different groups.

10, In Figures 3B, 3G, and 5F, there are several bands on the TRIM10a blot; however, in Figure 5D, there is only one band. This discrepancy is confusing and not convincing, and these assays should be repeated.

11, In Figure 3B, what cells were used in the IgG group for the IP assay? Additionally, why does the IgG control also bind a significant amount of TRIM10a proteins, including both the monomer and polymer?

12, Since phosphorylation of AKT, mTOR, and S6K, as well as EPOR protein levels, were increased under C1q-depleted conditions in Figure 5C, and these proteins are closely related to cell proliferation, yet no difference in cell number was observed between the normal and C1q-depleted conditions in Supplemental Figure 4A, could the authors comment on this discrepancy?

13, Also, in Supplemental Figure 4A, the cell number decreased with differentiation under both normal serum and C1q-depleted serum, suggesting that many cells died. Could the authors comment on this?

Minor points:

1, Since the authors identified TRIM10a oligomerization in Supplemental Figure 2B, could they provide evidence of similar oligomerization bands by displaying the full blot in Supplemental Figure 2C?

2, In Supplemental Figure 2D, the authors should include images of the WT group with the same scale as the other groups to facilitate comparison.

3, There are no indications provided in Figure 3A and 5A.

4, In Figure 5D, there is no IgG control for the IP assay.

5, In Supplemental Figure 6, the western blot results for the nuclear fraction should be shown.

Comments from the reviewers that required a response are in bold italics, with each reply appearing in normal font just below the comment.

Referee #1:

This study examines the role of TRIM10 α in erythroid maturation through stage-specific expression and subcellular localization patterns. The authors demonstrate that TRIM10 α exhibits dynamic expression during erythropoiesis, initially localizing to the cell surface in early erythroblasts where it captures extracellular C1q to facilitate pyrenocyte clearance by macrophages. In later stages, TRIM10 α partially relocates to cytosolic structures, where it reportedly enhances hemoglobin formation and sequesters harmful hemoglobin aggregates under oxidative conditions. The authors further suggest that surface C1q-TRIM10 α complexes promote EpoR lysosomal degradation, thereby attenuating protein synthesis during late erythroid maturation. Additionally, they identify an alternatively spliced variant, TRIM10 β , which forms deleterious aggregates when expressed. The study proposes that aberrant TRIM10 expression may contribute to erythroid-related diseases like sickle cell anemia and Diamond-Blackfan anemia.

While this manuscript presents an interesting hypothesis regarding TRIM10 α 's role in erythroid maturation, the data presented is insufficient to support the authors' mechanistic claims. The study relies heavily on descriptive observations from representative images without adequate quantification and statistical analysis. Additional experiments with quantitative analysis are required to support the proposed model.

We thank the reviewer for giving us valuable and important comments. We have carefully addressed all the reviewer's questions and issues on a point-by-point basis.

Major Concerns:

1. The manuscript relies heavily on immunofluorescence images and immunoblots, often using single representative cells to draw conclusions. While the images are informative, quantitative analysis is essential for scientific conclusions. This reviewer recommends quantifying colocalization coefficients for all immunofluorescence experiments, providing multiple fields of view and statistical analysis of observed phenotypes, and presenting data on the percentage of cells showing the described localization patterns.

We agree with the reviewer's comment and have therefore quantified colocalization in the majority of the immunofluorescence images using the ellipsoid formula in ZEN core software. In addition, we attempted to provide multiple fields of view; however, since erythroblasts are suspension cells with a small and rounded morphology, it was technically challenging to capture multiple fields of view during imaging. To address this issue, we

quantified the number of cells exhibiting the observed phenotype across multiple samples and performed statistical analysis to ensure the robustness of the data. All quantifications are now presented as graphs next to the corresponding confocal images, and their descriptions have been incorporated into the **Results** section of the revised manuscript.

2. The HUDEP-2 differentiation system is central to this study, but its characterization is insufficient. Since HUDEP-2 cultures likely contain heterogeneous cell populations at each time point, please provide flow cytometry data showing the proportion of cells at each differentiation stage for each time point, including quantification of enucleation efficiency in late-stage cultures.

We agree with the reviewer’s comment regarding the characterization of the HUDEP-2 differentiation system. As the reviewer noted, although HUDEP-2 cultures exhibit some heterogeneity, the majority of cells display characteristics corresponding to specific erythroid differentiation stages (Daniels et al., *Haematologica*, 2020; see Ref.1A below). As the reviewer correctly emphasized, confirming normal HUDEP-2 differentiation is a key component of our study; therefore, we performed flow cytometry using the erythroid surface markers CD36, CD49d, and CD235a, which are commonly used to delineate distinct maturation stages. Based on their expression patterns, cells were classified into four stages: undifferentiated (day 0: CD36^{high}/CD49d^{high}/CD235a^{low}), early (day 4: CD36^{high}/CD49d^{high}/CD235a^{low}), intermediate (day 8: CD36^{medium}/CD49d^{medium}/CD235a^{high}), and late (day 12: CD36^{medium/low}/CD49d^{medium/low}/CD235a^{high}) stages. The distribution of each population is shown in **new Fig. EV1E**, and these results are consistent with previously reported HUDEP-2 differentiation profiles (Ref 1B, Daniels et al., *Haematologica*, 2020).

To further confirm the increase in TRIM10 α expression during the mid-to-late stages of HUDEP-2 differentiation (**Figs. 1A** and **1B**), we sorted the heterogeneous CD235a^{low}, CD235a^{medium}, and CD235a^{high} populations observed at day 8 and measured TRIM10 α expression by qPCR. We clearly observed that TRIM10 α expression was selectively

upregulated in the CD235a^{high} population, which represents a definitive marker of the intermediate stage, supporting the notion that *TRIM10a* expression increases in a differentiation stage-dependent manner (Please see additional figure as below).

[Additional Figure 1] *TRIM10a* expression is high in CD235a^{high} populations of HUDEP-2 cells differentiated for 8 days. **[Left]**, flow cytometry sorting strategy based on two erythroid surface markers (CD235a and CD71). **[Middle and right]**, qPCR analysis of *TRIM10a* and *HBA1* relative to *GAPDH* in CD235a^{low}, CD235a^{medium}, and CD235a^{high} populations.

Although it recapitulates the early to late stages of erythropoiesis relatively well, we acknowledge that this model has limitations in representing the reticulocyte stage, as enucleated cells are not predominant over nucleated cells beyond day 12 (Ref 1A, Daniels *et al.*, *Haematologica*, 2020). Despite this limitation, enucleated cells (DAPI-negative and HbA-positive) and pyrenocytes (DAPI-positive and HbA-negative) were clearly detectable after day 12, and quantification revealed that all of pyrenocytes were TRIM10 α -positive (**Fig. 2A**, **New Figs. 2B** and **EV2B**).

These findings have been thus incorporated into the revised manuscript and additional statements have been included in the **Results** sections. We now believe that these additional datasets provide a clearer characterization of the HUDEP-2 differentiation stages and support the robustness of our experimental system.

3. Many experiments rely on overexpression experiments, which can lead to artifacts, particularly regarding protein aggregation. To strengthen the physiological relevance of the authors' findings, validate key observations using endogenous TRIM10 α , compare expression levels between overexpressed and endogenous TRIM10 α , and consider using CRISPR/Cas9-mediated tagging of endogenous TRIM10 α for localization studies.

We thank the reviewer for raising this important point regarding the potential limitations of overexpression-based analyses. We believe, however, that this concern may stem from a misunderstanding. As shown in **Fig. EV1A**, TRIM10 α is barely expressed in most cell types except erythroblasts; therefore, to investigate its physiological role, we focused on HUDEP-2 cells in which TRIM10 α is naturally upregulated during erythroid differentiation. We would like to emphasize that the majority of our experiments were conducted with endogenously expressed TRIM10 α in HUDEP-2 cells, as shown in Fig. 1 (all panels), Figs. 2A–D, Fig. 3 (all

panels), Fig. 4 (all panels), Fig. 5 (all panels), and Figs. 6A–H. Under these conditions, we conducted immunofluorescence, biochemical, and functional assays to assess TRIM10 α localization and function in a physiologically relevant system.

Furthermore, we utilized both siRNA- and CRISPR/Cas9-based sgRNA approaches to knock down endogenous TRIM10 α expression, as well as endogenous C1q-depleted conditions to assess the functional contribution of C1q. These approaches enabled us to investigate the respective roles of TRIM10 α and C1q in various cellular processes. We therefore hope this clarification addresses the reviewer's concern.

4. The flow cytometry data in Figure 2B is presented without adequate explanation of how it supports the conclusion that "TRIM10 α is localized on the cell surface." Please clarify the gating strategy and explain how surface versus intracellular staining was distinguished, include appropriate controls, and provide quantification across multiple experiments.

We thank the reviewer for pointing out the lack of detailed experimental explanation regarding the flow cytometry data in Fig. 2B (now presented as **Fig. 2D**). To assess surface TRIM10 α expression, we performed flow cytometry using an endogenous anti-TRIM10 α antibody under two conditions: with and without cell permeabilization with 1% Triton X-100. However, as the reviewer pointed out, staining under permeabilized conditions reflects both surface and intracellular TRIM10 α , and therefore the term "intracellular" may be misleading in this context. Accordingly, we have revised the figure to use the terms "Permeabilized" and "Non-permeabilized" to more accurately reflect the experimental conditions. These details and changes have been incorporated into the figure legend and are more thoroughly described in the revised **Methods** section.

5. The proteomics data visualization in Figures 3A, 5A, and 6A lacks clarity, making it difficult to evaluate the interactions claimed. Please improve the color coding to make the interaction categories clearly distinguishable, provide quantitative information on interaction strengths, and include statistical analysis of the reproducibility of observed interactions.

We agree with the reviewer's points. To address this, we repeated the proteomics experiments to improve the robustness and clarity of the data. For analysis, we selected only proteins that were consistently identified as TRIM10 α -interacting across replicates. A total of 136, 134, and 130 common TRIM10 α -interacting proteins were identified at D0, D8, and D12, respectively. Interaction strength was quantified by spectral coverage, and the data were visualized with refined color coding to distinguish interaction categories (**New Fig. EV3B**). Consistent with the previous results, proteomics analysis revealed that TRIM10 α

binds to cytoskeletal and erythroid-related proteins, indicating roles in enucleation and other cellular functions during erythroid maturation (**New Fig. EV3B**).

To minimize non-specific interactions, we excluded proteins detected at D0, when TRIM10 α expression was barely detectable, and subsequently identified 62 proteins at D8 and 45 at D12. Consistent with our previous results, C1q was the most prominent TRIM10 α -interacting protein at both D8 and D12, while no detectable interaction was observed at D0. Moreover, gene ontology analysis of the 17 proteins uniquely detected at both D8 and D12 revealed enrichment for the "complement C1q complex" within the cellular component category (**New Fig. 3A**). To further validate this finding, we performed immunoprecipitation using an endogenous anti-TRIM10 α antibody followed by immunoblotting with an anti-C1q antibody. This confirmed a clear interaction between endogenous TRIM10 α and C1q. These revised data, along with quantification and improved visualization, are now included in **new Figs. 3A and EV3B**, and the details are described in the revised **Figure legends** and **Results** section.

6. The claim that C1q promotes lysosomal degradation of EpoR is based primarily on the colocalization of EpoR with LAMP1 in one representative image. This evidence is insufficient for such a significant mechanistic claim. This reviewer recommends performing endocytosis inhibition experiments to determine if C1q-induced EpoR decrease can be rescued, conducting lysosomal inhibition studies, quantifying EpoR protein turnover rates under normal and C1q-depleted conditions, and providing quantitative colocalization analysis across multiple cells.

The reviewer raised a valuable point regarding whether endocytosis inhibition may rescue C1q-induced EpoR levels. Based on two previously published studies demonstrating the involvement of C1q in clathrin-mediated endocytosis (Ling, G.S. et al., *Science*, 2018; Nicole S. H. et al., *Cell*, 2024), we hypothesized that the EpoR–TRIM10 α –C1q complex undergoes clathrin-mediated internalization. To test this, we treated cells with the endocytosis inhibitor, Dynasore and observed that C1q-induced reduction of EpoR levels was rescued upon inhibition. This result supports the notion that extracellular C1q bound to TRIM10 α is internalized through clathrin-mediated endocytosis. Furthermore, as suggested by the reviewer, we also observed C1q accumulation in cells treated with the lysosomal inhibitor chloroquine. These findings have been incorporated into the revised manuscript as part of **new Figs. 5D and 5E**, and additional statements have been included in the **Results** sections.

In addition, as suggested by the reviewer, we quantified EpoR protein turnover rates under both normal and C1q-depleted conditions, and completed the quantitative colocalization

analysis of EpoR with LAMP1 across multiple cells (**Fig. 5B** and **New Figs. 5F–H**). These additional data are now included in the revised manuscript to further substantiate the proposed mechanism.

7. The manuscript claims that TRIM10 α promotes the formation of HbA and HbB dimers, but several critical aspects are unclear. Please specify whether these are HbA/HbB heterodimers or HbA/HbA and HbB/HbB homodimers, explain the physiological significance of these dimers in erythroid maturation, assess whether normal hemoglobin tetramers are affected by TRIM10 α knockdown, and if claiming a role in hemoglobin quality control, evaluate the impact on functional tetrameric hemoglobin.

The reviewer raised a valuable point by asking about the role of TRIM10 α in hemoglobin quality control. Heme-inserted globin dimers are generally considered as mature intermediates in hemoglobin assembly and are known to form predominantly as heterodimers rather than homodimers, as reported by Ghosh et al. (PNAS, 2017). Notably, when hemoglobin was immunoprecipitated with anti-HbA and blotted for anti-HbB, we did not detect monomeric β -globin but instead observed only dimeric and tetrameric forms (please see Additional Fig. 2A (left) below). This finding is consistent with the well-established property that β -globin does not exist as a free monomer under physiological conditions, but rather forms stable $\alpha\beta$ heterodimers that subsequently assemble into $\alpha_2\beta_2$ tetramers (Ghosh et al., PNAS 2017; Kihm et al., Nature 2002). The absence of monomeric species therefore supports the notion that the functional assembly unit of hemoglobin is the heterodimer.

[Additional Figure 2A] HbA and HbB forms stable $\alpha\beta$ heterodimers. Co-immunoprecipitation analysis of the interaction between HbA and HbB in HUDEP-2 cells differentiated for 12 days. Lysates were immunoblotted with anti-HbA, anti-HbB, or anti-Vinculin.

Consistent with this notion, our source data showed that TRIM10 α knockdown caused a concomitant decrease in both dimeric and tetrameric hemoglobin species (as indicated by the yellow arrows in Additional Fig. 2B), supporting the idea that TRIM10 α contributes to hemoglobin maturation by modulating multimeric assembly. To further support this observation, we depleted *TRIM10a* using sgRNAs in HUDEP-2 cells and observed a marked reduction in cellular redness and overall cell number (**New Figs. EV3F–H**). These findings indicate that TRIM10 α may contribute to hemoglobin maturation and potentially support erythroid cell survival by facilitating proper globin assembly.

Although the direct regulatory mechanisms remain to be established, our findings raise the possibility that TRIM10 α contributes to hemoglobin maturation and cell survival, thereby providing a basis for future investigation. We have accordingly included additional statements in the **Discussion** section of the revised manuscript to reflect these points.

8. There appears to be a contradiction in the data regarding TRIM10 α 's role in hemoglobin aggregate clearance. If TRIM10 α contributes to aggregate clearance, one would expect increased aggregate accumulation in TRIM10 α knockdown cells, yet the opposite is observed. Please address this apparent contradiction and provide evidence for TRIM10 α 's role in clearance, consider alternative interpretations, such as TRIM10 α promoting aggregate formation rather than clearance.

We thank the reviewer for the valuable and insightful comments. Originally, we hypothesized that TRIM10 α is involved in hemoglobin aggregate clearance based on the following observations: (1) TRIM10 α physically interacts with HbA and HbB (**Fig. 6G**); (2) HbA appears to form oligomer-like species that, unlike those of HbB, exhibit strong binding affinity to TRIM10 α (**Fig. 6G**); (3) TRIM10 α undergoes self-ubiquitination via its RING domain (**Fig. 6I**); and (4) ubiquitinated TRIM10 α strongly colocalizes with p62 (**New Fig. 6J**).

However, as the reviewer pointed out, there is no direct evidence that TRIM10 α directly ubiquitinates HbA or that such ubiquitination is the mechanism by which HbA is targeted for degradation. Therefore, we first examined whether the oligomer-like HbA band pattern observed at day 12 (D12) represents a ubiquitinated form, and whether it accumulates further under proteasome inhibition. When HUDEP-2 cell lysates from D0 and D12 were IP with an anti-HbA antibody and subsequently immunoblotted with an anti-Ubiquitin antibody, there was no detectable increase in HbA ubiquitination at Day 12 compared with Day 0 (**New Fig. 6E**). This suggests that high-molecular-weight HbA species are more likely to represent oligomeric forms of HbA rather than ubiquitinated HbA and that TRIM10 α is unlikely to function as an E3 ligase that directly ubiquitinates HbA.

In line with the reviewer's suggestion, we therefore considered the possibility that TRIM10 α may be involved in the formation of hemoglobin aggregates. Interestingly, several studies have reported that "Heinz bodies"—insoluble proteinaceous inclusion bodies formed by aggregated hemoglobin—are among the most representative inclusion bodies observed in reticulocytes (Weiss *et al.*, *Blood*, 2008; Hilbert *et al.*, *Haematologica*, 2020; Sagar *et al.*, *Ann Hum Biol*, 2015). It has been also well known that during reticulocyte maturation, mitochondria, ribosomes, and other unnecessary proteins are eliminated, and oxidized or misfolded globins transiently accumulate in Heinz bodies before being efficiently cleared (Weiss *et al.*, *Blood*, 2008). Indeed, these bodies have been shown to recruit chaperones such as p62/SQSTM1 and HSP proteins (Weiss *et al.*, *Blood*, 2008), all of which were also found to bind to TRIM10 α (**New Figs. 3A and EV3B**). Although further comparative studies are warranted, our findings support the idea that TRIM10 α may play a role in promoting the formation or regulation of such protein aggregates, as well as in the recruitment of degradative pathways such as autophagy during erythroid maturation. Based on the reviewer's insightful suggestion, we revised the manuscript to include the alternative interpretation and additional statements in the **Discussion** section, clarifying the potential implications of this perspective.

9. The relationship between TRIM10 α self-ubiquitination and autophagy-mediated degradation needs further clarification. Explain the causal relationship between these processes, determine whether the Δ RING mutant undergoes autophagy-mediated degradation, investigate whether TRIM10 α , p62, and HbA colocalize in the same structures, and perform functional experiments to verify that autophagy is indeed responsible for degradation.

We agree with the reviewer's comment and have therefore examined whether TRIM10 α , p62, and HbA are colocalized together within the same structures in HUDEP-2 cells at day 12 of differentiation. As expected, all three proteins were found to be localized within the same structures (**New Fig. 6F**). As the reviewer suggested, we further examined the effect of Δ RING mutant on p62 recruitment. In Δ RING mutant-expressing cells, p62 colocalization were markedly reduced compared to the wild-type TRIM10 α , and the number of p62-positive speckles also decreased (**New Fig. 6J**). These findings support the notion that TRIM10 α self-ubiquitination facilitates p62 recruitment. As described above (**in response to Comment #8**), it is plausible that TRIM10 α facilitates the formation of hemoglobin aggregates and their subsequent degradation through p62-mediated autophagy. Although the precise molecular mechanism was not fully elucidated in this study, we believe our findings offer mechanistic insights that will guide future investigations. We have incorporated this point into the revised **Discussion** section to reflect the reviewer's valuable

suggestions.

10. The analysis of TRIM10 β function lacks appropriate TRIM10 α controls at comparable expression levels. Please include TRIM10 α expressed at levels comparable to TRIM10 β in all Figure 7C-I experiments, clarify whether the TRIM10 α expression levels in previous figures are comparable to TRIM10 β levels shown in Figure 7, and consider titrating expression levels to determine if the observed differences are expression-level dependent or isoform-specific.

We agree with the reviewer's point regarding whether the proteotoxic effect of TRIM10 β is due to its expression level or is isoform-specific. Since both TRIM10 α and TRIM10 β are barely detectable in most non-erythroid cell types, we selected HEK-293T cells and introduced either GFP-tagged TRIM10 α or TRIM10 β , allowing us to specifically assess the effects of each exogenously expressed isoform. We evaluated cell viability in HEK-293T cells expressing GFP-TRIM10 α or GFP-TRIM10 β and confirmed that TRIM10 β expression resulted in a dose-dependent decrease in cell viability when equal amounts of DNA constructs were transfected (**New Fig. 7C**). Interestingly, although to a lesser extent, a similar reduction in viability was also observed with TRIM10 α expression. These findings may help explain why TRIM10 α is not broadly expressed throughout erythropoiesis but instead shows stringent expression at specific, limited stages; why it is silenced in non-erythroid cells; and why its aberrant expression in reticulocytes of patients with sickle cell anemia may contribute to disease pathology. Although the current study does not elucidate the detailed mechanisms underlying this association, we have revised the text accordingly and included a discussion in the revised **Discussion** section on the potential link between the aberrant expression of TRIM10 α/β and the development of diseases associated with cell death.

Comments from the reviewers that required a response are in bold italics, with each reply appearing in normal font just below the comment.

Referee #2:

In this study, Heesoo Kim et al. have identified the role of TRIM10, an erythroid-and stage-specific E3 ligase, in erythroid maturation. By analyzing the protein interactome via IP-mass spectrometry at different stages of erythroid maturation in the HUDEP-2 model, they uncovered several downstream targets of TRIM10, both in its extracellular and cytosolic forms. Specifically, they observed that TRIM10 self-associates on erythroblast surfaces and binds extracellular complement C1q, facilitating pyrenocyte encapsulation and macrophage recognition. Furthermore, they reported that surface C1q interacts with EpoR, promoting lysosomal degradation, while TRIM10 depletion prolongs Epo signaling. Additionally, they demonstrated that cytosolic TRIM10 enhances hemoglobin (Hb) maturation and sequesters Hb aggregates under oxidative conditions.

Although these findings are intriguing, the study lacks direct genetic evidence and relies solely on co-localization data between TRIM10 and its regulatory targets. Please see my major comments below:

We thank the reviewer for giving us valuable and important comments. We have carefully addressed all the reviewer's questions and issues on a point-by-point basis.

1. Cellular Differentiation and Maturation Phenotype:

• What is the cellular differentiation and maturation phenotype following TRIM10 loss? Was enucleation severely disrupted after TRIM10 depletion? Given that the authors have generated a TRIM10-depleted HUDEP-2 cell line (Figure 3G using siRNAs), it is surprising that this phenotype was not included. It would be straightforward to knockout TRIM10 using the CRISPR/Cas9 system.

We thank the reviewer for the valuable comments. Although we presented impairments in both pyrenocyte engulfment by macrophages and hemoglobin maturation in TRIM10 α -depleted HUDEP-2 cells using a TRIM10 α -targeted shRNA system in the previous version (Figs. 4 and 6), we agree with the reviewer's point regarding whether erythroid maturation and enucleation are affected by TRIM10 α depletion. To address this, we additionally employed a CRISPR/Cas9-based sgRNA approach to deplete endogenous TRIM10 α expression, and observed a marked reduction in both cellular redness and total erythroblast number (New Figs. EV3F–H), suggesting that TRIM10 α contributes to erythroid maturation and may support erythroid cell survival by facilitating proper hemoglobin maturation. Moreover, since we demonstrated that TRIM10 α plays a critical role in binding extracellular

C1q and promoting pyrenocyte encapsulation for efficient engulfment, we also showed a loss of C1q expression in TRIM10 α -depleted cells (**Fig. 3H**). In line with this, we further examined the functional contribution of C1q under endogenous C1q-depleted conditions, and observed impairments in (1) pyrenocyte engulfment by macrophages (**Fig. 4**); (2) EpoR endocytosis and lysosomal degradation (**New Figs. 5D and 5E**); and (3) EpoR-mediated inhibition of protein synthesis (**Fig. 5B**). We have therefore incorporated these additional datasets and appropriate statements in the revised manuscript, and we hope this clarification addresses the reviewer's concern.

2. Primary Hematopoietic Stem or Progenitor Cells:

• More convincingly, what is the cellular phenotype when TRIM10 is depleted in mouse or human primary hematopoietic stem or progenitor cells? This would be a standard system to test erythroid differentiation or even enucleation. In Figure 1D, the authors showed TRIM10 expression data in DBA or healthy donors. It is necessary to determine whether TRIM10 is functionally relevant, rather than merely correlated.

We agree with the reviewer's comment on the importance of evaluating the functional relevance of TRIM10 α in vivo or ex vivo. However, as TRIM10 α knockout mice are currently unavailable, we instead attempted to purify hematopoietic cells from mouse bone marrow and differentiate them into CFU-E cells. Consistent with our findings in HUDEP-2 cells, we observed an increase in Trim10 α expression in mouse erythroblasts beginning in the mid to late stages of erythroid maturation (**New Fig. EV1C**). Despite multiple attempts over the course of more than four months during the revision period to perform siRNA-mediated knockdown, we were unfortunately unable to proceed further due to a substantial reduction in cell yield following TRIM10 α -targeted siRNA transfection, which made downstream analyses unfeasible.

Nonetheless, these findings suggest that TRIM10 α expression is regulated in a stage-specific manner during erythroid differentiation in both mouse and human systems (**New Fig. EV1C**). In particular, the observed decrease in viability and cellular redness in TRIM10 α -depleted HUDEP-2 cells further supports the possibility that TRIM10 α contributes to erythroid maturation and cell survival (**New Figs. EV3F–H**). To directly address this hypothesis and to further explore whether TRIM10 α dysregulation is functionally involved in the pathogenesis of DBA, we plan to generate *TRIM10a* knockout mice and investigate its physiological relevance and disease-associated mechanisms in vivo. Moreover, the limitations of the current study in demonstrating the physiological relevance of TRIM10 α , as well as the importance of generating *TRIM10a* knockout mice to address this issue, have been discussed in the revised **Discussion** section.

3. Mutant Forms of TRIM10:

• The authors presented several TRIM10 mutant forms in Figure 2. Do these mutants exhibit normal biological function? In other words, if they have a TRIM10-depleted HUDEP-2 cell line, they could demonstrate cellular function by reintroducing these mutant forms and comparing them to the wild-type form.

As described in the main text, the localization of TRIM10 α to the cell surface is critical for its ability to bind extracellular C1q, which in turn facilitates the encapsulation and clearance of pyrenocytes. To support the AlphaFold-predicted structural model suggesting that homodimerization via the coiled-coil (CC) domain and phospholipid binding via the PRY/SPRY domain may be important for TRIM10 α translocation to the cell surface, we performed domain-deletion experiments and confirmed that these regions are indeed required for surface localization (**New Figs. EV2E–G**).

As the reviewer pointed out, we would originally like to assess the functional impact of these domain deletions. However, since TRIM10 α expression is dynamically regulated during erythroblast differentiation and is only expressed at specific stages, we believe that results obtained from constitutive expression of the mutants throughout the entire differentiation process may not accurately reflect their true physiological function, thereby making it difficult to properly evaluate the functional consequences of each domain deletion. Nonetheless, we agree with the reviewer's concern and revised the manuscript to clearly state that the observed results from the deletion mutants are limited to effects on subcellular localization, and should not be interpreted as direct evidence of functional consequences, thereby avoiding overstatement of our findings.

4. Direct Ubiquitinated Targets of TRIM10:

• In Figure 7, the authors reported that TRIM10 self-ubiquitination and its binding to p62 lead to TRIM10 degradation, which promotes the removal of Hb aggregates via autophagy. This is an interesting finding. However, as an E3 ubiquitin ligase, what are the direct ubiquitinated targets of TRIM10? Is Hb aggregate a candidate? Does TRIM10 α directly ubiquitinate HBA and degrade HBA dimers? In this regard, the authors have not fully explored the protein targets of TRIM10. Given that they used IP-mass spectrometry for the protein interactome, they could ideally compare the interactome between wild-type TRIM10 and a TRIM10 enzymatic mutant form. Alternatively, they could profile the total proteome after TRIM10 depletion and analyze potential substrates that might accumulate in the absence of TRIM10 (see the E3 ligase example in the reference method PMID:PMC7820877).

We thank the reviewer for the valuable and insightful comments. As the reviewer

suggested, we also hypothesized that TRIM10 α may function as an E3 ligase involved in the ubiquitination and degradation of HbA, based on the following observations: (1) TRIM10 α physically interacts with both HbA and HbB (**Fig. 6G**); and (2) HbA appears to form high-molecular-weight, ubiquitin-like species that exhibit strong binding affinity to TRIM10 α (**Fig. 6G**). However, if TRIM10 α functioned as an E3 ligase promoting HbA ubiquitination, one would expect an increase in HbA levels upon TRIM10 α knockdown. Contrary to this expectation, we instead observed a decrease in HbA levels (**Fig. 6D**).

To further clarify this result, we additionally examined whether the oligomer-like HbA band pattern observed at day 12 (D12) represents a ubiquitinated form, and whether it accumulates further under proteasome inhibition. When HUDEP-2 cell lysates from D0 and D12 were IP with an anti-Ubiquitin antibody and subsequently immunoblotted with an anti-HbA antibody, there was no detectable increase in HbA ubiquitination at Day 12 compared with Day 0 (**New Fig. 6E**). This suggests that high-molecular-weight HbA species are more likely to represent oligomeric forms of HbA rather than ubiquitinated HbA and that TRIM10 α is unlikely to function as an E3 ligase that directly ubiquitinates HbA.

In line with this point, our current findings strongly indicate that TRIM10 α primarily undergoes auto-ubiquitination, as deletion of its RING domain markedly reduced ubiquitination levels (**Fig. 6I**). These results suggest that TRIM10 α may act predominantly on itself as a substrate under the conditions tested, although the possibility of additional substrates cannot be excluded. Together with the observed TRIM10 α -p62 interaction, we considered the possibility that TRIM10 α may play a role in hemoglobin aggregate regulation. To explore this possibility further, several studies have reported that ‘Heinz bodies’—insoluble proteinaceous inclusion bodies formed by aggregated hemoglobin—are among the most representative inclusion bodies found in reticulocytes (Weiss *et al.*, *Blood*, 2008). During reticulocyte maturation, mitochondria, ribosomes, and other superfluous components are eliminated, while oxidized or misfolded globins transiently accumulate in Heinz bodies before being efficiently cleared (Weiss *et al.*, *Blood*, 2008). These bodies are known to recruit chaperones such as p62/SQSTM1 and HSP proteins (Weiss *et al.*, *Blood*, 2008), all of which, notably, were also observed to bind to TRIM10 α (**New Figs. 3A and EV3B**). Although further studies are needed to elucidate the precise mechanisms, our findings support the idea that TRIM10 α may be involved in the formation or regulation of these hemoglobin-containing protein aggregates, as well as in recruiting degradation machinery such as autophagy. In response to the reviewer’s insightful suggestion, we have incorporated this alternative interpretation into the revised **Discussion** section, clarifying its potential implications.

As the reviewer also suggested, proteomic analysis using cells expressing wild-type TRIM10 α and a catalytic-dead mutant could potentially help identify candidate substrates of TRIM10 α . While such an approach is beyond the primary scope of the current study, we agree that it represents a valuable direction for future investigation and may provide important mechanistic insights. Accordingly, we have revised the manuscript to include additional statements in the **Discussion** section that reflect the reviewer's suggestion.

5. Mechanism of C1q Entry Mediated by TRIM10 α :

• *The authors propose that C1q entry into cells is mediated by TRIM10 α , but TRIM10 α expression in HUDEP2-D0 cells and C1q synthesis do not occur simultaneously. Therefore, why do undifferentiated HUDEP2 cells not absorb C1q? The authors' research found that TRIM10 α suppression prevents C1q from entering cells. So, how does TRIM10 α mediate the entry of C1q into cells? Do the proteins interacting with TRIM10 α bind to any channel proteins? As an intermediate, does it facilitate C1q entry into the cytoplasm?*

The reviewer raised a valuable point by asking about the apparent discrepancy between the TRIM10 α expression and C1q availability, and how TRIM10 α facilitates C1q internalization. As previously noted in the manuscript, erythroblasts do not exhibit endogenous C1q transcription at any stage of differentiation and naturally lack intracellular C1q (Fig. EV3C). Therefore, the presence of C1q observed in erythroblasts is largely dependent on TRIM10 α , which is markedly upregulated during the mid-to-late stages of differentiation. This suggests that TRIM10 α facilitates the binding of extracellular C1q to the cell surface and its subsequent internalization, thereby promoting the selective clearance of C1q-coated pyrenocytes by macrophages via engagement with their abundant C1q receptors.

In particular, EpoR, which we identified in this study as a TRIM10 α -interacting protein, has been reported to undergo internalization via a clathrin-dependent endocytic pathway (R Sulahian *et al.*, *Blood*, 2009). To address the reviewer's question regarding the mechanism of TRIM10 α -mediated C1q internalization, we treated cells with the endocytosis inhibitor Dynasore and observed that the C1q-induced reduction in EpoR levels was rescued upon inhibition (**New Fig. 5E**). This result supports the notion that extracellular C1q bound to TRIM10 α is internalized through clathrin-mediated endocytosis. These findings have been incorporated into the revised manuscript as part of Fig. 5E, and additional statements have been included in the **Results** sections.

6. Inclusion of Proteome Dataset:

• *It would be more comprehensive to include the proteome dataset in Figure 3. A total interactome network or Gene Ontology (GO) analysis is strongly recommended for TRIM10*

binding partners in HUDEP-2 cells from days 0,8,and 12,corresponding to the Pro-E, Poly, and Ortho/reticulocyte stages. Previous proteome studies could be referenced for comparison and discussion (e.g. HUDEP-2 PMID:PMC9718529 and primary CD34+proteome study PMID:PMC7706838).

We thank the reviewer for this valuable suggestion and therefore repeated the proteomics experiments to improve the robustness and clarity of the data. For analysis, we selected only proteins that were consistently identified as TRIM10 α -interacting across replicates. A total of 136, 134, and 130 common TRIM10 α -interacting proteins were identified at D0, D8, and D12, respectively. Interaction strength was quantified by spectral coverage, and the data were visualized with refined color coding to distinguish interaction categories (**New Fig. EV3B**). Consistent with the previous results, proteomics analysis revealed that TRIM10 α binds to cytoskeletal and erythroid-related proteins, indicating roles in enucleation and other cellular functions during erythroid maturation (**New Fig. EV3B**).

To minimize non-specific interactions, we excluded proteins detected at D0, when TRIM10 α expression was barely detectable, and subsequently identified 62 proteins at D8 and 45 at D12. Consistent with our previous results, C1q was the most prominent TRIM10 α -interacting protein at both D8 and D12, while no detectable interaction was observed at D0. Moreover, gene ontology analysis of the 17 proteins uniquely detected at both D8 and D12 revealed enrichment for the "complement C1q complex" within the cellular component category (**New Fig. 3A**). To further validate this finding, we performed immunoprecipitation using an endogenous anti-TRIM10 α antibody followed by immunoblotting with an anti-C1q antibody. This confirmed a clear interaction between endogenous TRIM10 α and C1q. These revised data, along with quantification and improved visualization, are now included in **new Figs. 3A and EV3B**, and the details are described in the revised **Figure legends and Results** section.

Comments from the reviewers that required a response are in bold italics, with each reply appearing in normal font just below the comment.

Referee #3:

The functions of tripartite motif (TRIM) proteins, the largest superfamily of RING-type E3 ligases, in erythroid cells remain poorly understood. Here, Heesoo Kim et al. found that TRIM10a is an erythroid- and stage-specific E3 ligase that plays a crucial role in stepwise erythroid maturation.

First, using published erythroid transcriptome data, clinical datasets, and in-house bulk RNA sequencing data, they found that aberrant TRIM10a expression is closely related to anemia. Then, they explored TRIM10a expression changes via immunofluorescence staining during human terminal erythroid differentiation and found that the localization of TRIM10a includes the erythroblast surface and cytoplasm. TRIM10a self-associates to localize on erythroblast surfaces and interacts with the extracellular complement C1q protein, which promotes pyrenocyte encapsulation.

Furthermore, the authors found that C1q interacts with EpoR and promotes its lysosomal degradation, while C1q depletion prolongs Epo signaling. In addition, cytosolic TRIM10a enhances hemoglobin maturation and sequesters its aggregates via p62-mediated autophagy under oxidative conditions. Finally, they also showed that TRIM10b, an alternatively spliced variant of TRIM10a, is barely expressed in human cells and suppresses erythropoiesis by forming deleterious aggregates.

The authors concluded that aberrant TRIM10 could drive erythroid-related disorders and serve as a potential biomarker or therapeutic target.

The findings in this manuscript are not entirely convincing, as many key results are derived from subjective assays. The following concerns should be addressed to strengthen the conclusions.

We thank the reviewer for giving us valuable and important comments. We have carefully addressed all the reviewer's questions and issues on a point-by-point basis.

Major points:

1. Given that the TRIM10a antibody can recognize several non-specific bands shown in Figure 3B, 3G and 5F, making the results of IP-Mass Spectrometry using this antibody unconvincing, and that ectopic GFP-TRIM10a shows similar colocalization with endogenous TRIM10a, could the authors perform IP-Mass Spectrometry with ectopic GFP-

TRIM10 α to strengthen the results?

We thank the reviewer for pointing out the issue regarding the antibody specificity for TRIM10 α . As the reviewer suggested, it would be informative to analyze HUDEP-2 cells expressing GFP- TRIM10 α . However, since TRIM10 α expression is dynamically regulated during erythroblast differentiation and is only expressed at specific stages, we believe that results obtained from a constitutively GFP-TRIM10 α -expressing stable cell system may not accurately reflect the physiological interactome of TRIM10 α .

To minimize the inclusion of non-specific proteins, we had performed the co-IP assay with anti-TRIM10 α antibody using undifferentiated HUDEP-2 cells (D0), in which TRIM10 α is barely detectable. Therefore, all proteins detected under this condition were excluded from the final dataset to eliminate non-specific binding proteins. Additionally, since TRIM10 α is found to undergo self-oligomerization and self-ubiquitination, multiple TRIM10 α species may appear upon IP. This was confirmed by our mass spectrometry data, where TRIM10 α was consistently detected across a range of molecular weights.

Furthermore, we also repeated the proteomics experiments to improve the robustness and clarity of the data during the revision period. For analysis, we selected only proteins that were consistently identified as TRIM10 α -interacting across replicates, while excluding all proteins detected in the D0 samples, where endogenous TRIM10 α expression is negligible, to eliminate potential non-specific binding.

Based on the mass spectrometry analysis, we specifically focused on C1q, which was identified as a TRIM10 α -binding protein uniquely present at D8 and D12 stages. To further validate these interactions, we performed IP using an endogenous anti-TRIM10 α antibody followed by immunoblotting with an anti-C1q antibody. This confirmed a clear interaction between endogenous TRIM10 α and C1q. These revised data, along with quantification and improved visualization, are now included in **New Figs. 3A and EV3B**, and the details are described in the revised figure legends and **Results** section.

We have also revised the manuscript to include additional clarification in response to the reviewer's concerns to ensure better understanding.

2. Figure 4B and 4C are too subjective to be convincing. The authors should assess erythroid differentiation, enucleation, and pyrenocyte percentage changes using flow cytometry after co-culture, rather than relying solely on counting attached erythroid cells based on selectively chosen representative images.

we agree with the reviewer's point regarding whether erythroid maturation and enucleation are affected by TRIM10 α or C1q depletion. To address this, we employed a CRISPR/Cas9-based sgRNA approach to deplete endogenous TRIM10 α expression, and observed a marked reduction in both cellular redness and total erythroblast number (**New Figs. EV3F–H**), suggesting that TRIM10 α contributes to erythroid maturation and may support erythroid cell survival by facilitating proper hemoglobin maturation. Moreover, since we demonstrated that TRIM10 α plays a critical role in binding extracellular C1q and promoting pyrenocyte encapsulation for efficient engulfment, we also showed a loss of C1q expression in TRIM10 α -depleted cells (**Fig. 4**).

In line with this, we further examined the functional contribution of C1q under endogenous C1q-depleted conditions, and observed impairments in (1) pyrenocyte engulfment by macrophages (**Fig. 4**); (2) EpoR endocytosis and lysosomal degradation (**New Figs. 5D and 5E**); and (3) EpoR-mediated inhibition of protein synthesis (**Fig. 5B**). We have therefore incorporated these additional datasets and appropriate statements in the revised manuscript, and we hope this clarification addresses the reviewer's concern.

3. In Figure 3D, could the authors quantify the immunofluorescence signals? Since both TRIM10 α and C1qC signals consist of two forms-aggregates (bright spots) and smears (faint smears)-it is difficult to compare signal differences across different time points. Additionally, the colocalization between TRIM10 α and C1qC does not appear to be significant. Could the authors provide comments on this observation?

We agree with the reviewer's comment and have therefore quantified colocalization in the majority of the immunofluorescence images using the ellipsoid formula in ZEN core software. We confirmed the clear colocalization of TRIM10 α and C1q, which was further validated by co-IP analysis (**Fig. 3B**). These results are now presented as graphs in **New Figs. 3F and 3G** and the corresponding descriptions have been incorporated into the **Results** section of the revised manuscript.

4. Could the authors provide some in vivo results? What are the phenotypes of TRIM10 α knockout mice and C1q knockout mice? Are there any defects in erythropoiesis in these mice?

We agree with the reviewer's comment on the importance of evaluating the in vivo functional relevance of TRIM10 α and C1q. However, as TRIM10 α and C1q knockout mice are not currently available, we were unable to directly assess their physiological roles in erythropoiesis within the limited three-month revision period. Nonetheless, we fully acknowledge the importance of this question and plan to address it in future studies by

generating TRIM10 α and C1q knockout mouse models to investigate their physiological relevance and potential roles in erythropoiesis. A corresponding statement has been included in the revised **Discussion** section.

5. The statement 'the TRIM10 α Δ CC and TRIM10 α Δ PRY/SPRY mutants mainly localized to the cytosol, losing their ability to target the plasma membrane' is not supported by the IF staining in Figure 2E, as these mutants also localize to the plasma membrane, especially the TRIM10 α Δ CC mutant. It would be better to perform flow cytometry without cell permeabilization to further confirm it.

We thank the reviewer for pointing out the lack of detailed experimental explanation regarding the flow cytometry data in Figure 2E (now presented as **Fig. 2G**). To assess surface expression of both Δ CC and Δ PRY/SPRY mutants, we performed flow cytometry with non-permeabilized conditions. As a result, the Δ PRY/SPRY mutant exhibited a marked reduction in cell surface expression of TRIM10 α , while the Δ CC mutant also showed decreased surface localization compared to the wild-type, albeit to a lesser extent than the Δ PRY/SPRY mutant. Consistent with our previous findings, these results suggest that both the CC and PRY/SPRY domains contribute to the cell surface localization of TRIM10 α , with the PRY/SPRY domain playing a particularly critical role. These results are also included in **New Figs. EV2E–G**, and the relevant descriptions have been added to the **Results** section of the revised manuscript.

6. Also, in Figure 2E, the nucleus size in the WT group appears smaller than in the other groups. Does this indicate that TRIM10 α mutants affect nuclear condensation and terminal erythroid differentiation? If so, please quantify this figure and provide additional assays, such as flow cytometry, to further confirm the findings.

We thank the reviewer for this valuable suggestion. In response, we re-examined all images of HUDEP-2 and HEK-293T cells expressing either wild-type TRIM10 α or the domain deletion mutants and quantified nuclear size. However, no substantial differences were observed between the wild-type and mutant-expressing cells (Please see the Additional Fig. 1 below).

[Additional Figure 1] Nuclear size is not affected by TRIM10 α . Nuclear area of HUDEP-2 cells expressing control vectors or Myc-tagged WT-TRIM10 α and TRIM10 α deletion mutants was quantified using ImageJ

Additionally, we would originally like to assess the functional impact of these domain deletions. However, since TRIM10 α expression is dynamically regulated during erythroblast differentiation and is only expressed at specific stages, we believe that results obtained from constitutive expression of the mutants throughout the entire differentiation process may not accurately reflect their true physiological function, thereby making it difficult to properly evaluate the functional consequences of each domain deletion. Nonetheless, we agree with the reviewer's point and revised the manuscript to clearly state that the observed results from the deletion mutants are limited to effects on subcellular localization, and should not be interpreted as direct evidence of functional consequences, thereby avoiding overstatement of our findings.

7. In Figure 3B, could the authors also detect C1qA and C1qB, since the secreted C1Q protein is a complex of C1qA, C1qB and C1qC in a constant ratio?

We thank the reviewer for these insightful comments and have therefore revisited the mass spectrometry datasets to determine whether C1qA or C1qB were also identified, as was C1qC. We observed the C1qB in both D8 and D12, but no clear peptide signatures corresponding to C1qA was detected above the threshold.

As the reviewer has pointed out, although C1q is composed of three distinct polypeptides (C1qA, B, and C) that assemble into a functional heterotrimeric complex, only C1qC and C1qB was robustly detected in our mass spectrometry dataset. The absence of detectable C1qA may be due to poor tryptic peptide generation, limited ionization efficiency, or selective interaction/enrichment of specific subunits by TRIM10 α . These findings do not exclude the presence of the full C1q complex but rather reflect technical and biochemical limitations commonly encountered in proteomic workflows. We have accordingly included additional statements in the **Result** section of the revised manuscript to reflect these points.

8. In Figure 3F and Supplemental Figure 3D, it is not appropriate to compare cell counts, as gated cell counts are positively correlated with the total recorded cell counts. Percentages would be a better measure to show the differences.

We agree with the reviewer's comment and have therefore revised the data presentation to show percentages instead of absolute cell counts, as suggested. The updated figures and corresponding legends have been modified accordingly in the revised manuscript.

9. In Figure 5C, is HSP70 the loading control? If so, these HSP70 blots should be repeated, as the loading is unequal and the molecular weights of HSP70 vary between the different groups.

We thank the reviewer for this important comment. It has been reported that during erythroid differentiation, commonly used loading controls, including actin, tubulin, and vinculin, undergo extensive proteome remodeling (Mathangasinghe *et al.*, *Haematologica*, 2021; see Ref. 2A–B below). In that study, HSP70 was reported to maintain relatively stable protein levels throughout erythroid maturation, which is why we selected it as the loading control in our study. As the reviewer has pointed out, although slight band shifts in HSP70 have been observed under oxidative conditions during differentiation, the overall protein levels of HSP70 remained consistent from D0 to D12, with only a marginally lower molecular weight band detected at D0. Interestingly, this difference in band mobility disappeared under non-reducing conditions, suggesting that the observed shift may be due to redox-dependent post-translational modifications rather than unequal loading (Please see the Additional Figures at the end of this response). Therefore, we have accordingly included additional statements in the **Result** section of the revised manuscript to reflect these points.

[Additional Figure 2] Difference in band motility of HSP70 disappeared under non-reducing conditions. Lysate from HUDEP-2 cells differentiated for the indicated times were separated by non-reducing or reducing SDS-PAGE, followed by immunoblotting with anti-HSP70 antibody.

10. In Figures 3B, 3G, and 5F, there are several bands on the TRIM10α blot; however, in Figure 5D, there is only one band. This discrepancy is confusing and not convincing, and these assays should be repeated.

We appreciate the reviewer's careful observation regarding the discrepancies in the TRIM10α blot patterns. As shown in our earlier results (**Fig. 1**), TRIM10α expression is barely detectable at Day 0, becomes progressively induced from Day 4 to Day 8, and is ultimately decreased by Day 12. Therefore, the prominent band observed near 50 kDa at Day 0 is most likely a non-specific signal, given the negligible TRIM10α expression at this stage.

Consistently, this band appears just below the specific TRIM10α signal in all uncropped

immunoblots, further supporting the interpretation that it is a non-specific artifact rather than a true TRIM10 α band (Please see the Additional Fig. 3 at the end of this response). However, as the reviewer pointed out, to avoid confusion, we have revised the **Figure legends** and **Results** section to more clearly distinguish specific and non-specific bands.

11. In Figure 3B, what cells were used in the IgG group for the IP assay? Additionally, why does the IgG control also bind a significant amount of TRIM10 α proteins, including both the monomer and polymer?

We agree with the reviewer's comment. HUDEP-2 cells at Day 0 were used for the IgG group in the IP assay shown in Fig. 3B. Although the presence of the IgG heavy chain makes it difficult to clearly distinguish the TRIM10 α monomer band, the band observed in the IgG lane appears at a lower molecular weight and does not correspond to the size of TRIM10 α oligomers. This distinction is more clearly illustrated in **Fig. 6G** of the manuscript; for the reviewer's convenience, we have re-inserted the figure below, with the relevant bands highlighted by red arrows (Please see the Additional Fig. 4 at the end of this response).

12. Since phosphorylation of AKT, mTOR, and S6K, as well as EPOR protein levels, were increased under C1q-depleted conditions in Figure 5C, and these proteins are closely

related to cell proliferation, yet no difference in cell number was observed between the normal and C1q-depleted conditions in Supplemental Figure 4A, could the authors comment on this discrepancy?

We thank the reviewer for these insightful comments. While AKT, mTOR, and S6K signaling pathways are indeed well known for their roles in promoting cell proliferation, erythroid differentiation is a unique process that involves extensive proteome remodeling, organelle clearance, nuclear condensation, and nuclear extrusion (Vinchi, *Hemasphere*, 2018). Moreover, erythroid maturation is tightly linked to cell cycle exit, with well-characterized regulators such as p18 and p27 playing key roles in this transition (Gnanapragasam *et al.*, *Blood*, 2016). Therefore, even under conditions of enhanced proliferative signaling due to C1q depletion, these signals may be insufficient to overcome the intrinsic cell cycle exit program of erythroid cells. This may explain why the cell number did not significantly increase and instead showed a slight decrease despite elevated levels of AKT/mTOR/S6K phosphorylation.

13. Also, in Supplemental Figure 4A, the cell number decreased with differentiation under both normal serum and C1q-depleted serum, suggesting that many cells died. Could the authors comment on this?

We appreciate the reviewer's observation. As addressed in the previous response, erythroid differentiation involves a gradual decline in cell cycle activity, particularly during the mid-to-late stages, resulting in decreased proliferation. Consequently, it is well established that total cell numbers tend to decline as differentiation proceeds (Hsieh *et al.*, *Blood*, 2000). Consistent with this, we also observed a gradual reduction in cell number under both normal and C1q-depleted serum conditions, which likely reflects this physiological transition rather than increased cell death. This clarification has been incorporated into the revised Discussion section of the revised manuscript.

Minor points:

1. Since the authors identified TRIM10 α oligomerization in Supplemental Figure 2B, could they provide evidence of similar oligomerization bands by displaying the full blot in Supplemental Figure 2C?

We thank the reviewer for the comment and confirm that TRIM10 α oligomerization bands are clearly visible in the full blot shown in Additional Fig. 5 at the end of this response, consistent with the results in Supplemental Fig. 2B (now presented as **Fig. EV2C**). All uncropped gels, including this blot, have been provided in the Source Data.

2. In Supplemental Figure 2D, the authors should include images of the WT group with the same scale as the other groups to facilitate comparison.

We agree that including images of the WT group at the same scale would enhance visual comparison. Accordingly, we have updated Supplemental Figure 2D (now presented as **Fig. EV2H**) to include WT images captured and displayed at the same scale as the other groups.

3. There are no indications provided in Figure 3A and 5A.

We thank the reviewer for pointing out the lack of clear indications and have thus revised Fig. 3A to be more representative and added appropriate indications (labels and annotations). The former Figure 5A was removed and its content integrated into the revised Figs. 3A and EV3B.

4. In Figure 5D, there is no IgG control for the IP assay.

We agree with the reviewer's point and have repeated the IP experiment with the inclusion of an IgG control. The revised data are now presented in the updated figure in the manuscript.

5. In Supplemental Figure 6, the western blot results for the nuclear fraction should be shown.

We thank the reviewer's point and have repeated the immunoblot analysis with the inclusion of the nuclear fraction. Interestingly, TRIM10 α was also detected in the nuclear fraction, and upon re-examination of multiple confocal images, we noted a consistent nuclear localization pattern. These observations raise the possibility that TRIM10 α may have a functional role within the nucleus, providing insights for future studies. We have included this result and a brief related statement in the revised manuscript.

Dear Prof. Park,

Thank you for the submission of your revised manuscript to our editorial offices. I have now received the reports from the three referees that I asked to re-evaluate the study, you will find below. As you will see, the referees now fully support the publication of your study in EMBO reports.

Before we can proceed with formal acceptance, I have these editorial requests I ask you to address in a final revised manuscript:

- We now use CRediT to specify the contributions of each author in the journal submission system. CRediT replaces the author contribution section. Please use the free text box to provide more detailed descriptions and do NOT provide your final manuscript text file with an author contributions section. See also our guide to authors: <https://www.embopress.org/page/journal/14693178/authorguide#authorshipguidelines>

- Please order the manuscript sections like this, using only these names:
Title page - Abstract - Keywords - Introduction - Results - Discussion - Methods - Data availability section - Acknowledgements (please put here all the funding information) - Disclosure and Competing Interests Statement - References - Figure legends - Expanded View Figure legends

- Please use our reference format:
<http://www.embopress.org/page/journal/14693178/authorguide#referencesformat>

- I would suggest moving the data shown in the two Appendix figures to the EV figures. Some of these (e.g. EV4) are rather small and could accommodate the data. Like this, we do not need an Appendix. Please update all the respective callouts and the EV figure legends and add the source data to the respective folders of the EV figures source data.

- Please check again that the number "n" for how many independent experiments were performed, their nature (biological versus technical replicates), the bars and error bars (e.g. SEM, SD) and the test used to calculate p-values is indicated in the respective figure legends. Please also check that all the p-values are explained in the legend, and that these fit to those shown in the figure. Please provide statistical testing where applicable. Please avoid the phrase 'independent experiment' but clearly state if these were biological or technical replicates. Please also indicate (e.g. with n.s.) if testing was performed, but the differences are not significant. In case n=2, please show the data as separate datapoints without error bars and statistics. See also:
<http://www.embopress.org/page/journal/14693178/authorguide#statisticalanalysis>

If n<5, please show single datapoints for diagrams. Presently, some diagrams seem to lack statistics, lack n.s. or show incomplete statistics (e.g. 2C, 2G, 7A, 7F, EV1G or EV3E). Please check. Moreover:

- Please indicate the statistical test used for data analysis in the legends of figures 1C, G; 2B, 5A.

- Please note that the box plots need to be defined in terms of minima, maxima, centre, bounds of box and whiskers, and percentile in the legend of figure 1A

- Please note that information related to n is missing in the legends of figures 1A, B, C, D, G, 2D, 5A, G, H; 7E, F, EV2 F.

- Please note that the error bars are not defined in the legends of figures 2C, D; 5A, 7F.

- Please add to each legend (main and EV figures, where applicable) a 'Data Information' section explaining the statistics used or providing information regarding replicates and scales. See:

- There are 3 EV tables called out (EV1 and EV2 in the manuscript text file, and EV3 in the Reagents & Tools Table) that have not been provided and seem to contain information about constructs and primers. Please add this information directly to the Reagents & Tools Table (do not upload separate tables) and adjust the callouts in the main text file.

- Please remove now the referee token from the Data Availability Section (DAS) and make sure that the datasets are public latest upon online publication of the manuscript.

- Please remove mention and links to datasets from the DAS that have not been generated during this study. Publicly available datasets by others that have been re-analyzed in this study need to be added as data citations to the reference list and appropriately called out. See (section 'data citation'):

<https://www.embopress.org/page/journal/14693178/authorguide#referencesformat>

- Please make sure that all the funding information is also entered into the online submission system and that it is complete and similar to the one in the acknowledgement section of the manuscript text file. Presently, the grants RS-2025-00521704, 2018R1A6A1A03025607, Brain Korea (BK21) FOUR Program, Yonsei University Research Fund of 2024-22-0044 are missing

in the submission system. Please check.

- Thanks for providing the source data. Please upload this as one folder per main figure, grouping together all the files for this figure (and ZIPed together), and as one folder for the EV figures containing separate folders for each EV figure).

- Please provide the schematic summary figure in jpeg or tiff format and with the exact width of 550 pixels and a height of not more than 400 pixels. Please make sure that all text in the image can be well read (use appropriate font size).

Best,

Referee #1:

The revised manuscript is significantly improved, with major concerns from my initial review addressed through new data, clearer methodology, and refined interpretations. Imaging quantification, HUDEP-2 characterization, and proteomics validation strengthen the study, while additional experiments convincingly demonstrate C1q-dependent EpoR degradation. Evidence for hemoglobin maturation and aggregate clearance has been clarified, and TRIM10 α/β functional comparisons add further insight.

Although some mechanistic details remain unresolved, the work is now compelling and well supported. I recommend acceptance with only minor editorial revisions if needed.

Referee #2:

The authors have fully addressed my major concerns: they have added extensive functional experiments across multiple erythroid models to establish the causal impact of TRIM10 loss.

Referee #3:

The authors have adequately addressed my comments.

All editorial and formatting issues were resolved by the authors.

Prof. Boyoun Park
Yonsei University
Systems Biology
50 Yonsei-ro, Seodaemun-gu
Seoul 120-749
Korea, Republic of

Dear Prof. Park,

I am very pleased to accept your manuscript for publication in the next available issue of EMBO reports. Thank you for your contribution to our journal.

Yours sincerely,
